# Particles Don't Care About Z: Towards Scaling Entropy Estimation of Unnormalized Densities

## Abstract

Computing the differential entropy of distributions known only up to a normalization constant is a long-standing challenge with broad theoretical and practical significance. While variational inference is the most scalable approach for density approximation *from samples*, its potential in settings where *only the unnormalized density* is available remains largely under-explored. The central difficulty lies in constructing variational distributions that simultaneously ($i$) exploit the structure of the unnormalized density, ($ii$) are expressive enough to capture complex target distributions, ($iii$) remain computationally tractable, and ($iv$) support efficient sampling. Recently, Messaoud et al. (2024) introduced *P-SVGD*, a particle-based variational method that leverages Stein Variational Gradient Descent dynamics, satisfies all of these constraints and demonstrates promising results in low-dimensional setups. We show, however, that *P-SVGD* does not scale to high dimensions due to *fundamental algorithmic flaws*: ($i$) misdiagnosed sensitivity to *SVGD* hyperparameters, ($ii$) violation of the global invertibility assumption in the entropy derivation, ($iii$) omission of a critical trace-of-Hessian term, ($iv$) along with suboptimal heuristics, including a divergence-based sampling check that induces mode collapse and loose informal bounds with no practical value. These issues severely limit both the correctness and the scalability of the approach. We propose *MET-SVGD*, a principled extension of *P-SVGD* that addresses these flaws by providing a general framework for *SVGD* hyperparameters selection with global invertibility and convergence guarantees. This enabled more accurate and scalable entropy estimation in high-dimensional settings. Empirically, in entropy estimation benchmarks, *MET-SVGD* achieves accuracy improvements of up to $12\times$ and $16\times$ over *P-SVGD* and baselines from the *SVGD* literature, respectively. On CIFAR-10 Energy-Based image generation, it improves FID by $80.4\%$ compared to *P-SVGD* and achieves $64\times$ higher training stability. In Maximum-Entropy reinforcement learning, *MET-SVGD* yields up to $16\%$ better returns than *P-SVGD*. We will make our code publicly available at `https://tinyurl.com/2esyfx8j`.

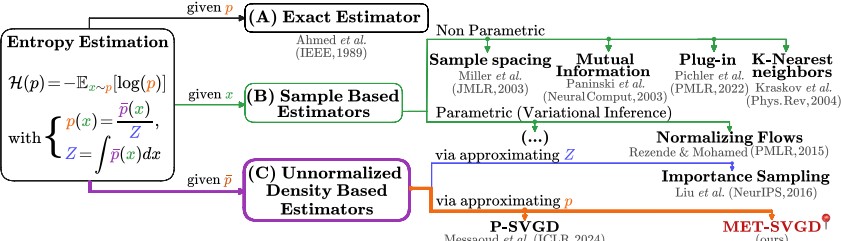

Figure 1: *MET-SVGD* is a new variational inference approach for entropy estimation of distributions known only up to a normalization constant (given $\bar{p}$), extending the *P-SVGD* (Messaoud et al., 2024) lineage with principled corrections enabling scalability to high-dimensional targets.

## 1 Introduction

The differential entropy (Cover, 1999; Shannon, 1948) of a $d$-dimensional random variable $X$ with a probability density function $p(x) = \bar{p}(x)/Z$ is $\mathcal{H}(p) = -\mathbb{E}_{x \sim p(x)}[\log p(x)] = -\int p(x) \log p_X(x) dx$, with $Z = \int \bar{p}(x) dx$ being the normalization constant. The differential entropy plays a central role in information theory, signal processing, and machine learning (Tarasenko, 1968; Learned-Miller & III, 2003; Wulfmeier et al., 2015; Liu et al., 2022a; Hino & Murata, 2010; Rubinstein & Kroese, 2004; Mannor et al., 2005). However, estimating it is challenging, since a closed-form expression is only available for a limited class of distributions (*e.g.*, Gaussians). In practice, only samples $x \sim p$ or the unnormalized density $\bar{p}$ are available (Fig. 1).

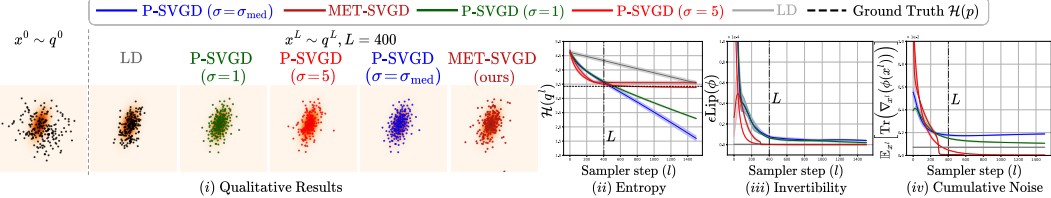

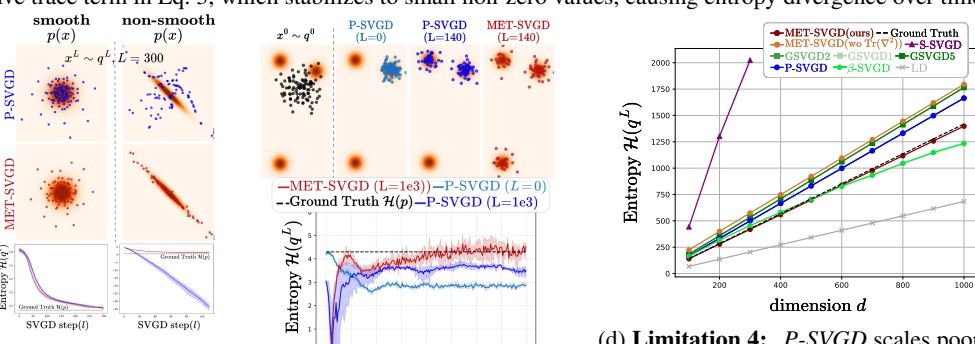

(a) **Limitation 1:** *P-SVGD* suffers from high sensitivity to the RBF kernel bandwidth $\sigma$. *(i) Qualitatively*, particles converge under all sampling schemes, including Langevin Dynamics (*LD*). *(ii)* However, *quantitatively*, entropy only matches the ground-truth for some $\sigma$ values. *(iii)* Messaoud et al. (2024) misdiagnose this issue as violation of the *SVGD* update invertibility; yet, we show that our global invertibility condition (Prop. 3.2) is always satisfied. *(iv)* Instead, entropy divergence in *LD* and *P-SVGD* with $\sigma \in \{\sigma_{\text{med}}, 1\}$ arises from the cumulative trace term in Eq. 3, which stabilizes to small non-zero values, causing entropy divergence over time.

(b) **Limitation 2:** *P-SVGD* suffers from poor convergence to non-smooth targets.

(c) **Limitation 3:** *P-SVGD* suffers from mode collapse. $L$ is the number of *SVGD* steps.

(d) **Limitation 4:** *P-SVGD* scales poorly to high dimensional spaces. Benchmark from Liu et al. (2022b) (App. H). *MET-SVGD(wo. $Tr(\nabla^2)$)* refers to the variant without the correction term.

Figure 2: *P-SVGD* limitations: (a) high sensitivity to kernel variance ($\sigma$), (b) poor convergence to non-smooth targets, (c) sampling mode collapse and (d) limited scalability to high-dimensional spaces. *MET-SVGD* addresses these shortcomings and achieves improved accuracy and scalability in entropy estimation. We provide implementation details in App. H.

While entropy estimation from samples has been extensively studied (Beirlant et al., 1997; Paninski, 2003), the setting where only the unnormalized density is available remains underexplored. This setup is, however, central to many machine learning applications. Notably, in Energy-Based Models (EBMs) (LeCun et al., 2006), entropy is critical for promoting sample diversity and avoiding mode collapse Liu et al. (2023). Another example is Maximum Entropy Reinforcement Learning (MaxEnt RL) (Haarnoja et al., 2018) where the entropy directly drives exploration and robust policy learning.

A common approach to entropy estimation in *unnormalized density settings* is to approximate the normalization constant $Z$ using sampling-based techniques, *e.g.*, importance sampling (Cantwell, 2022). However, such estimates often suffer from high variance in high-dimensional spaces. Another approach is to use normalizing flows Rezende & Mohamed (2015). But in this case, the constructed variational distribution does not leverage knowledge about the unnormalized density $\bar{p}$. In contrast, Messaoud et al. (2024) introduced Parametrized Stein Variational Gradient Descent (*P-SVGD*), which builds on Stein Variational Gradient Descent (*SVGD*) sampler (Liu & Wang, 2016) to construct a variational distribution $q^L$ from $\bar{p}$. *SVGD* updates a set of interacting particles $\{x_i\}_{i=0}^{M-1}$ using a deterministic velocity field $\phi(\cdot)$ that balances a gradient force with a repulsive one:

$$\phi(x_i^l) = \mathbb{E}_{x_j^l \sim q^l} \left[ \kappa(x_i^l, x_j^l) \nabla_{x_j^l} \log \bar{p}(x_j^l) + \nabla_{x_j^l} \kappa(x_i^l, x_j^l) \right], \quad (1)$$

following the update rule $x_i^{l+1} = x_i^l + \epsilon \phi(x_i^l)$, where $\epsilon$ is the step-size, $q^l$ is the particles distribution at step $l \in [1, L]$ and $\kappa(\cdot, \cdot)$ is typically an RBF kernel with variance $\sigma^2$, *i.e.*, $\kappa(x_i, x_j) = \exp(-||x_i - x_j||^2 / 2\sigma^2)$. *P-SVGD* derives a closed form expression of $q^l$ at every step $l$, including the final one:

$$\log \hat{q}^L(x_i^L) = \log q^0(x_i^0) - \epsilon \sum_{l=0}^{L-1} \sum_{i \neq j=0}^{M-1} \frac{\kappa(x_j^l, x_i^l)}{M\sigma^2} \left( d - \frac{||x_i^l - x_j^l||^2}{\sigma^2} - (x_i^l - x_j^l)^\top \nabla_{x_j^l} \log \bar{p}(x_j^l) \right) + \mathcal{O}(\epsilon^2). \quad (2)$$

$\hat{q}^L$, the empirical estimate of $q^L$, is efficient to compute as it only depends on first-order derivatives and vector dot products. Under mild assumptions (Villani et al., 2009), it can approximate a broad

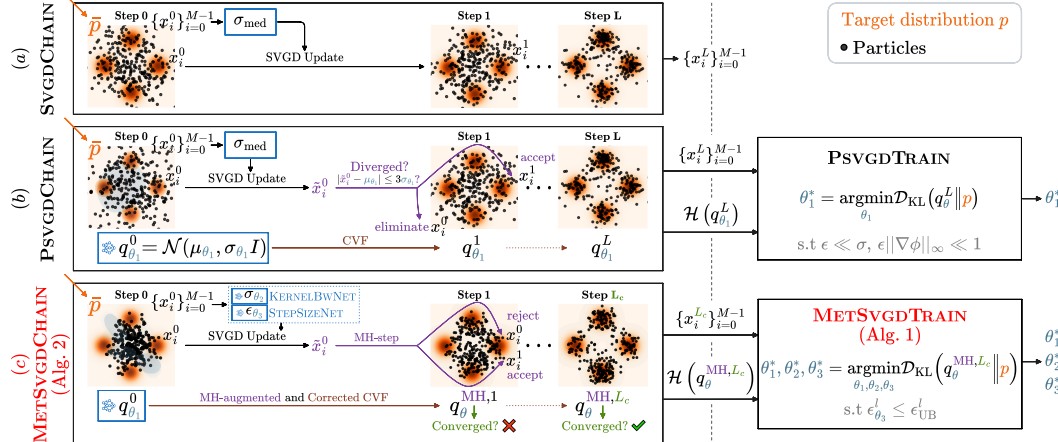

Figure 3: **Novelty over P-SVGD.** (b) P-SVGD and (c) MET-SVGD transform (a) SVGD from a sampling algorithm to a variational inference one. In both methods, the parameters of the variational distribution are learnt end-to-end via reverse KL-divergence minimization (PSVGDTRAIN, METSVGDTRAIN). *MET-SVGD*'s contributions are: $(C_1)$ A single sufficient condition on the step-size for both global invertibility of the *SVGD* update and $\log\text{-det}$ approximation in the entropy derivation, replacing two informal constraints in PSVGDTRAIN (Corr. 3.3). $(C_2)$ End-to-end learning of kernel bandwidth $\sigma_{\theta_2}$ and step-size $\epsilon_{\theta_2}$, mitigating hyperparameter sensitivity (Sec. 3.2). $(C_3)$ Adaptive number of sampling steps $L_c$ instead of a fixed one, with Stein's Identity-based convergence monitoring (Eq. 6). $(C_4)$ Corrected application of the Change-of-Variables Formula (CVF) via efficient restoration of the missing trace-of-Hessian (Sec. 3.3). $(C_5)$ Replacement of the divergence control heuristic with a principled Metropolis-Hastings (MH) step (Sec . 3.4).

class of densities, enabling accurate $\mathcal{H}(p)$ estimation with compelling results in MaxEntr RL. Nevertheless, we show that *P-SVGD* exhibits several limitations, including sensitivity to hyperparameters, mode collapse, poor convergence to non-smooth targets, and lack of scalability (Fig. 2).

In this paper, we trace the above-listed *P-SVGD* limitations to fundamental algorithmic flaws and introduce principled fixes that yield a scalable framework with provable correctness and asymptotic convergence guarantees, which we call *MET*ropolis–Hastings augmented *SVGD* (*MET-SVGD*). First, we refute the claim in *P-SVGD* that the sensitivity to the RBF kernel bandwidth arises from violations of the invertibility assumption. We prove that global invertibility is, in fact, satisfied, and show that divergence instead stems from accumulating small noise over time. To address this, *MET-SVGD* learns adaptive step-wise kernel bandwidths and step sizes. Second, we derive a new sufficient condition for *global* invertibility based on Banach's fixed-point theorem (Banach, 1932), replacing *P-SVGD*'s reliance on a weaker *local* condition. Third, we consolidate two informal and independent step-size conditions for invertibility and $\log\text{-det}$ approximation (Proposition 3.2 and Theorem 3.1 in *P-SVGD*), into a single principled one. Besides, we restore the trace-of-Hessian term omitted in *P-SVGD* for computational efficiency, using Hutchinson's estimator Hutchinson (1989) and show that this simplification is a significant cause behind poor scalability, as it is only valid asymptotically and fails in the finite-particle regime. Additionally, for divergence control, we replace *P-SVGD*'s particle truncation heuristic, which we show encourages mode collapse, with a Metropolis–Hastings (MH) correction Robert et al. (2004) that induces a more expressive variational distribution $q^L$ and guarantees asymptotic convergence. Finally, we use Stein's identity (Kattumannil, 2009) violation as a metric to adaptively determine convergence instead of fixing the number of sampling steps in advance as in *P-SVGD*. This enables adaptation to the complexity of target distributions. This is, for instance, critical in reinforcement learning, where each state induces a different action distribution. All of the proposed contributions introduce minimal overhead, and together elevate particle-based variational inference from a sampling method to a principled and scalable framework for entropy estimation for unnormalized densities.

*MET-SVGD* achieves SOTA performance on *SVGD* scalability benchmarks with up to $16\times$ higher accuracy, thus providing a solution to the long-standing challenge of automatically setting samplers' hyperparameters. On entropy estimation, *MET-SVGD* achieves up to $12\times$ higher accuracy than *P-SVGD* on Gaussian and Gaussian Mixture Models targets with up to 1000 dimensions. On CIFAR10 image generation using EBMs, it results in $77.27\%$ improvement in FID and $64\times$ better training stability compared to *P-SVGD*. In MaxEntr RL, it yields up to $16\times$ higher returns than *P-SVGD*.

## 2 RELATED WORK

In the following, we provide a brief review of variational inference, *SVGD* and MH.

**Variational Inference (VI)** (Fox & Roberts, 2012) approximates the target $p$, via a simpler-to-sample-from distribution $q^*$ from a predefined family $\mathcal{Q}$ by minimizing the reverse KL-divergence *i.e.*, $q^* = \arg\min_{q \in \mathcal{Q}} D_{\mathrm{KL}}(q\|p)$. More expressive $\mathcal{Q}$ families yield better approximations.

**Stein Variational Gradient Descent (SVGD)** (Liu & Wang, 2016) is a sampling algorithm with update rule given in Eq. 1. Traditionally, an RBF kernel is used, with its bandwidth $\sigma$ set via the median heuristic: $\sigma_{\mathrm{med}} = \mathrm{median}\{\|x_i^l - x_j^l\|\}_{i,j=0}^{M-1}/\log M$. The bandwidth $\sigma$ controls the influence of neighboring particles $\{x_j^l\}$ on the update of particle $x_i^l$: larger $\sigma$ values induce broader neighborhoods, while $\sigma \to 0$ decouples the particles (Fig. 12 in App. B.4). *SVGD* has several advantages over existing *approximate inference* methods. Unlike variational inference (VI), *SVGD* can sample from arbitrary complex distributions under smoothness assumptions (Villani et al., 2009). Compared to Markov Chain Monte Carlo (MCMC) methods (Chib, 2001), *SVGD* is more particle-efficient and its convergence can be easily checked using Stein's Identity (Kattumannil, 2009). However, *SVGD* convergence is only proved under certain conditions, such as sub-Gaussian targets (Shi & Mackey, 2022) or infinite particles (Salim et al., 2022; Liu & Wang, 2018). Additionally, *SVGD* suffers from poor scalability to high-dimensional spaces due to diminishing repulsive forces (Zhuo et al., 2018). To address this, existing solutions are based on dimensionality reduction via projections into low-dimensional manifolds (Gong et al., 2021; Liu et al., 2022b), which, however, either lead to inflated variance (*e.g.*, *S-SVGD* (Gong et al., 2021)) or are inefficient (*e.g.*, *GSVGD* (Liu et al., 2022b)).

**Parametrized SVGD (P-SVGD)** Messaoud et al. (2024)) is a **VI** method for entropy estimation from unnormalized densities. Under *invertibility assumption* of the *SVGD* update rule (Eq. 1), it computes the density of the particles $q^L(x^L)$ by sequentially applying the change-of-variables formula (CVF) (Devore et al., 2012) over $L$ steps, under an invertibility condition derived from the implicit function theorem (App. A.3): $\log q^{l+1}(x^{l+1}) = \log q^l(x^l) - \log|\det(I + \epsilon\nabla_{x^l}\phi(x^l))|$. To avoid computing the full Jacobian, two approximations are introduced: (*i*) If $\epsilon\|\nabla_{x^l}\phi(x^l)\|_\infty \ll 1$, the determinant is reduced to its trace via Jacobi's formula (App. A.4), leading to

$$\log q^L(x^L) = \log q^0(x^0) - \epsilon \sum_{l=0}^{L-1} \mathbb{E}_{x_j^l \sim q^l}\Big[\mathrm{Tr}\Big(\partial\bar{\phi}(x^l, x_j^l)/\partial x^l\Big)\Big] + \mathcal{O}(\epsilon^2), \qquad (3)$$

where $\bar{\phi}(x^l, x_j^l)$ is the contribution of particle $x_j^l$ to the update of particle $x^l$ and $\phi(x^l) = \mathbb{E}_{x_j^l}[\bar{\phi}(x^l, x_j^l)]$ is the velocity defined in Eq. 1. (*ii*) The trace term is further approximated using only first-order derivatives and vector dot products under the RBF kernel $\kappa(x^l, x_j^l) = \exp(-\|x^l - x_j^l\|^2/2\sigma^2)$, leading to Eq. 2. For computational efficiency, *P-SVGD omits the trace-of-Hessian term* $\mathrm{Tr}(\nabla_{x^l}^2 \log p(x^l))$ by sampling $x_j^l \neq x^l$ to approximate the expectation in $\phi$ (Eq. 2). The authors also demonstrate that *learning the initial distribution* parameters $q_\theta^0 = \mathcal{N}(\mu_\theta, \sigma_\theta I)$ via minimizing the reverse KL-divergence $D_{KL}(q_\theta^L\|p)$ and *preventing samples divergence* by constraining particles to stay within a few standard deviations of $q_\theta^0$'s mean, are essential for scaling (Fig. 3). In this work, we advance *P-SVGD* lineage by identifying and fixing several algorithmic limitations.

**Metropolis-Hastings** (MH, Robert et al. (2004)) is an MCMC method for sampling from a probability distribution when direct sampling is infeasible. The process involves two steps: (*i*) propose a new state $\tilde{x}^l$ from a proposal distribution $q^l(\tilde{x}^l|x^{l-1})$, and (*ii*) accept the proposal with probability

$$\alpha^l = \alpha(x^{l-1}, \tilde{x}^l) = \min\Big[1, \Big(p(\tilde{x}^l)/p(x^{l-1})\Big) \cdot \Big(q^l(x^{l-1}|\tilde{x}^l)/q^l(\tilde{x}^l|x^{l-1})\Big)\Big]. \qquad (4)$$

If accepted, the the new state is set to $\tilde{x}^l$ ; $x^l = \tilde{x}^l$, else the current state is retained $\tilde{x}^l = x^{l-1}$. This ensures asymptotic convergence to the target distribution given sufficient steps (Roberts & Rosenthal, 2004). Additional related work is covered in App. B.

## 3 APPROACH

*MET-SVGD* is a variational-inference method for computing the entropy of densities $p$ known up to a normalization constant, *i.e.*, it approximates $p$ with a tractable, simple to sample from distribution and estimates $\mathcal{H}(p)$ via the entropy of this distribution. To achieve this *MET-SVGD* introduces a series of optimizations to address *P-SVGD*'s key limitations as illustrated in Fig. 3: ($C_1$) *P-SVGD* introduces two informal independant constraints on the step-size including a *local* invertibility one, although the entropy derivation requires *global* invertibility; *MET-SVGD* unifies these constraints

---

**Algorithm 1** MET-SVGD-TRAIN (Training)

---

**input** Unnormalized target density $\bar{p}$; initial particle distribution $q_{\theta_1}^0$; number of particles $M$; maximum number of inner SVGD steps $L$; number of training iterations $T$; maximum number of iterations $B$ for the reverse computation; deepnets for the initial distribution, step size, and kernel bandwidth parameterized by $\theta_1$, $\theta_2$, and $\theta_3$, respectively.

**output** Trained parameters $\{\theta_1^*, \theta_2^*, \theta_3^*\}$.

1: Initialize $\theta_1, \theta_2, \theta_3$.
2: **for** $t = 1, \ldots, T$ **do**
3:    $\{x_i^{L_c}\}_{i=0}^{M-1}, \widehat{\mathcal{H}}\left(q_\theta^{\mathrm{MH}, L_c}\right) \leftarrow$ MetSvgdChain $\left(\bar{p}, q_{\theta_1}^0, M, L, B; \theta\right)$.          # cf. Alg. 2
4:    Compute training loss $D_{\mathrm{KL}}\left(q_\theta^{\mathrm{MH}, L_c} \middle\| p; \theta\right)$ using $\widehat{\mathcal{H}}$ and $\bar{p}$.          # cf. Eq. 5
5:    Update parameters $\theta$ using Adam optimizer Kingma & Ba (2014).
6: **end for**
7: **return** $\{\theta_1^*, \theta_2^*, \theta_3^*\}$.

---

into a single principled one satisfying *global* invertibility (Sec. 3.1). $(C_2)$ *P-SVGD* is highly sensitive to hyperparameters (Fig. 2a-*ii*) and provides no tuning guidelines. We show that this sensitivity arises from accumulation of noise in the trace term of Eq. 3, which leads to entropy divergence (Fig. 2a-*iv*); *MET-SVGD* learns *SVGD*'s hyperparameters end-to-end via reverse KL-divergence minimization, eliminating the need for *SVGD* hyperparameter tuning (Sec. 3.2). $(C_3)$ *P-SVGD* uses a fixed number of *SVGD* steps $L$, which may be insufficient for convergence; *MET-SVGD* adaptively determines the number of steps $L_c$ using a new efficient formulation of Stein's Identity as a convergence criterion (Sec. 3.2). $(C_4)$ *P-SVGD* omits the trace-of-Hessian term, which limits scalability in high-dimensional spaces (Fig. 2d); *MET-SVGD* efficiently restores this term as explained in Sec. 3.3. $(C_5)$ *P-SVGD* suffers from poor convergence to non-smooth targets and sampling mode collapse (Fig. 2b and Fig. 2c), due to its divergence control heuristic and the absence of convergence guarantees in the finite particle regime; *MET-SVGD* replaces this heuristic with an MH step, guaranteeing asymptotic convergence independently from the number of particles (Sec. 3.4). *MET-SVGD*'s training and inference algorithms are summarized in Alg. 1 and Alg. 2, respectively.

### 3.1 Conditions on the Step-Size for Invertibility and $|\log\text{-det}|$ Approximation

In *P-SVGD*, Eq. 2 is derived by $(i)$ using the change-of-variables formula (CVF) (App. A.2) under invertibility assumption of the *SVGD* update and $(ii)$ approximating the $\log$-det term in the CVF with an efficient trace estimator. These steps introduce two conditions on the *SVGD* step-size: $(i)$ $\epsilon \ll \sigma$ and $(ii)$ $\epsilon \|\nabla_{x^l}\phi(x^l)\|_\infty \ll 1$. However, these bounds suffer from two major limitations: $(i)$ Both are informal (use of $\ll$); in practice, $\epsilon$ is simply set to a small value, hoping that both constraints hold, which may not be true and often results in more steps than necessary. $(ii)$ The step-size condition only guarantees *local* invertibility (by the implicit function theorem (Krantz & Parks (2002))), whereas the CVF requires *global* invertibility. To fix this, we extend a sufficient condition for invertible resnets (Behrmann et al. (2019)) to *SVGD* (Prop. 3.1), derive a precise condition on $\epsilon$ for the $\log$-det approximation (Prop. 3.2), and unify both into a single efficient bound (Corr. 3.3).

**Proposition 3.1** (Sufficient condition for *global SVGD* invertibility). *Let $p(x) = \bar{p}(x)/Z$ be a continuously differentiable and strictly positive density on $\mathcal{X} \subset \mathbb{R}^d$, and let $f(x) = x + \epsilon\,\phi(x)$ be the SVGD update map with step size $\epsilon > 0$, where $\phi$ is the velocity field defined in Eq. 1 and computed with an RBF kernel. If $\epsilon \max_x \|\nabla\phi(x)\|_2 < 1$, then $f$ is globally invertible.*

*Proof Sketch:* We prove that the *SVGD* velocity is Lipschitz continuous under the assumptions of the proposition and leverage the Banach fixed point theorem for contractive mappings (App. D.2).

**Proposition 3.2** (Sufficient condition for $\log$-det Approximation). *Let $\phi : \mathbb{R}^d \to \mathbb{R}^d$ be a differentiable vector-valued map. $\log|\det(I + \epsilon\nabla_x\phi(x))| = \epsilon\,\mathrm{Tr}(\nabla_x\phi(x)) + \mathcal{O}\left(\epsilon^2\lambda_{max}^2(\nabla_x\phi(x))\right)$, if $\epsilon\,|\lambda_{max}(\nabla_x\phi(x))| < 1$, with $\lambda_{max}$ being the largest eigenvalue value in magnitude.*

Proof is in App. D.3. Note, *P-SVGD* chooses a very small $\epsilon$, hence the $\mathcal{O}(\epsilon^2)$ error term in Eq. 2.

**Corollary 3.3.** *Under the setup of Prop. 3.1, the $L$-step composite SVGD map defines the distribution in Eq. 2 if $\epsilon < \min_l \epsilon_{UB}^l$ with $\epsilon_{UB}^l = 1/\sup_{x^l} \sqrt{\mathrm{Tr}(\nabla_{x^l}\phi(x^l)\nabla_{x^l}\phi^T(x^l))}$ for all $l \in [0, L-1]$.*

*Proof Sketch:* We leverage: $|\lambda_{\max}(A)| \le \|A\|_2 \le \sqrt{\mathrm{Tr}(AA^T)}, \forall A \in \mathbb{R}^{d \times d}$ (App. D.4).

---

**Algorithm 2** METSVGDCHAIN (Inference)

---

**input** Unnormalized target $\bar{p}$; number of particles $M$; maximum number of steps $L$; deepnets for the initial distribution, maximum number of iterations $B$ for the reverse computation; step size, and kernel bandwidth parameterized by $\theta_1$, $\theta_2$, and $\theta_3$, respectively.

**output** Final particles $\{x_i^{L_c}\}_{i=0}^{M-1}$ and entropy estimate $\widehat{\mathcal{H}}(q_\theta^{\mathrm{MH},L_c})$.

1: $\{x_i^0\}_{i=0}^{M-1} \sim q_{\theta_1}^0$.                           # sample initial particles
2: $q_{\mathrm{MH}}^0(x_i^0) \leftarrow q_{\theta_1}^0(x_i^0)$.
3: $l \leftarrow 0$.
4: **repeat**
5:   $\sigma_{\theta_2}^l \leftarrow \text{KERNELBWNET}(\{x_i^l\}_{i=0}^{M-1}; \theta_2)$.      # kernel bandwidth   # cf. Eq. 1
6:   $\epsilon_{\theta_3}^l \leftarrow \text{STEPSIZENET}(\{x_i^l\}_{i=0}^{M-1}; \theta_3)$.                         # step size
7:   $\epsilon_{\theta_3}^l \leftarrow \min\left(\epsilon_{\theta_3}^l, \epsilon_{\mathrm{UB}}^l(\bar{p}, \{x_i^l\}_{i=0}^{M-1}, \sigma_{\theta_2}^l)\right)$.                      # cf. Corr. 3.3
8:   **for** $i = 0, \ldots, M-1$ **do**
9:     $r_i^l = \mathcal{U}\{-1, 1\}$                              # SVGD update direction
10:    $\tilde{x}_i^{l+1} \leftarrow \text{SVGDUPDATE}\left(r_i^l, \{x_j^l\}_{j=0}^{M-1}, \bar{p}, \sigma_{\theta_2}^l, \epsilon_{\theta_3}^l\right)$
11:    Compute acceptance probability $\alpha_{i,\theta}^l$ for proposal $\tilde{x}_i^{l+1}$.        # cf. Prop. 3.5
12:    Draw $u_i^l \sim \mathcal{U}(0, 1)$.
13:    **if** $u_i^l \leq \alpha_{i,\theta}^l$ **then** $x_i^{l+1} \leftarrow \tilde{x}_i^{l+1}$ **else** $x_i^{l+1} \leftarrow x_i^l$        # accept/reject step
14:    Compute $\hat{q}_\theta^{\mathrm{MH},l+1}(x_i^{l+1})$                          # cf. Prop. 3.6
15:   **end for**
16:   $l \leftarrow l+1$.
17: **until** $l \geq L$ **or** $\left(\text{SI}(\hat{q}_\theta^{\mathrm{MH},l+1}, p) - \text{SI}(q_\theta^{\mathrm{MH},l}, p) > 0\right)$              # cf. Eq. 6
18: $L_c \leftarrow l$.
19: $\widehat{\mathcal{H}}(q_\theta^{\mathrm{MH},L_c}) = \frac{-1}{M} \sum_{i=0}^{M-1} \log q_\theta^{\mathrm{MH},L_c}(x_i^{L_c})$              # Entropy estimate
20: **return** $\{x_i^{L_c}\}_{i=0}^{M-1}, \widehat{\mathcal{H}}(q_\theta^{\mathrm{MH},L_c})$.

**Subroutine:** SVGDUPDATE$\left(r_i^l, \{x_j^l\}_{j=0}^{M-1}, \bar{p}, \sigma_{\theta_2}^l, \epsilon_{\theta_3}^l\right)$

1: **if** $r_i^l = 1$ **then** $x_i^{l+1} \leftarrow x_i^l + \epsilon_{\theta_3}^l \phi_\theta(x_i^l)$                # forward SVGD
2: **else** $x_i^{l+1} \leftarrow \text{BANACHFIXEDPOINT}\left(\{x_j^l\}_{j=0}^{M-1}, B, \bar{p}, \sigma_{\theta_2}^l, \epsilon_{\theta_3}^l\right)$   # inverse SVGD (Alg. 3)
3: **return** $x_i^{l+1}$

---

**Complexity.** In App. D.5, we show that $\epsilon_{\mathrm{UB}}^l$ can be efficiently computed using only first-order derivatives and vector dot products by leveraging $\text{Tr}(\nabla\phi)$, making this condition practical.

## 3.2 OPTIMIZED SVGD PARAMETERS

A key drawback of *P-SVGD* is its high sensitivity to the RBF kernel bandwidth $\sigma$, which Messaoud et al. (2024) attribute to violation of the invertibility of the *SVGD* update rule (Eq. 1). In a 2d Gaussian experiment (reproduced in Fig. 2a), they show that, paradoxically, while both *SVGD* and Langevin Dynamics (*LD*) (update rule in App. A.1) *qualitatively* converge to the target, *i.e.*, the particles reach high-density regions (Fig. 2a-*i*), the entropy estimate converges only for specific $\sigma$ values, *e.g.*, $\sigma = 5$ (Fig. 2a-*ii*). The authors hypothesize that this behavior arises from *LD* being non-invertible and *SVGD* being invertible only for some $\sigma$ values. This is incorrect. As shown in Fig. 2a-*iii*, the condition from Prop. 3.1 is always satisfied. Instead, we show that the poor quantitative convergence to $\mathcal{H}(p)$ arises from the cumulative residual noise in the trace term of Eq. 3, *i.e.*, $\mathbb{E}_{x^l \sim q^l}[\text{Tr}(\nabla_{x^l}\phi(x^l))] \not\to 0$ as $l \to \infty$ (Fig. 2a-*iv*), yielding a quasi-linear growth in the entropy with the number of steps. To address this, we propose a principled procedure for *SVGD* hyperparameter selection: using the closed-form expression of $q_\theta^L$, *MET-SVGD* learns a step-wise kernel bandwidth $\sigma_{\theta_2}^l$ and step-size $\epsilon_{\theta_3}^l$ alongside the initial distribution $q_{\theta_1}^0$ by minimizing the reverse $D_{KL}$:

$$\theta^* = \arg\min_\theta -\mathcal{H}(q_\theta^L) - \mathbb{E}_{x^L \sim q_\theta^L}\left[\log \bar{p}(x^L)\right], \quad \text{s.t.} \quad \epsilon_{\theta_3}^l \leq \epsilon_{\mathrm{UB}}^l \quad \forall l \in [0, L-1], \qquad (5)$$

with $\epsilon_{\mathrm{UB}}^l$ being the upper-bound from Corr. 3.3 and $\theta = \{\theta_i\}_{i=1}^3$. Besides, we propose an efficient convergence check enabling an adaptive number of steps $L_c$, rather than fixing it a priori.

**Kernel and Parameters.** As the kernel bandwidth defines the neighborhood of influence for each particle, *i.e.*, which other particles contribute to its update, it can vary across iterations and dimensions. To capture this, we learn step- and dimension-wise kernel bandwidths $\sigma^l_{\theta_2} \in \mathbb{R}^d$. Fig. 4 shows, this improves convergence and exhibits behavior distinct

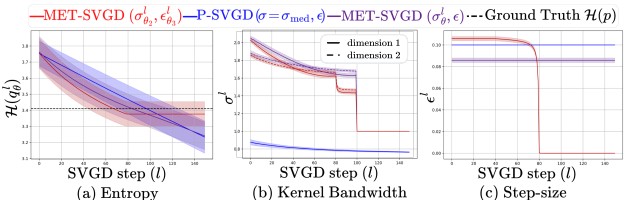

Figure 4: Learning step-/dimension- wise kernel-bandwidth and step-wise step-size improves the entropy estimate accuracy. Gaussian target from Fig. 2a.

from the commonly used $\sigma_{\mathrm{med}}$. We also evaluated alternative kernels such as the bilinear and DKEF kernels (App. C.3- C.4), but found the RBF kernel to provide the most favorable trade-off between flexibility and computational efficiency.

**Step-Size.** Learning the kernel bandwidth alone is insufficient to ensure convergence to ground-truth entropy, *i.e.*, the cumulative trace term $\mathbb{E}_{x^l \sim q^l}[\epsilon \operatorname{Tr}(\nabla_{x^l}\phi(x^l))]$ does not necessarily vanish as $l \to \infty$. In App. E, we show, via Taylor expansion, that this term corresponds to a 8$^{\mathrm{th}}$-degree polynomial which convergence to zero requires the existence of at least one real root. This is a nontrivial, fragile condition, as the polynomial's coefficients depend on the particle positions during training. To address this, we propose learning a step-wise step-size $\epsilon^l_{\theta_3}$ jointly with $\sigma^l_{\theta_2}$. This design enables the model to modulate the learning rate at each iteration, promoting a more stable convergence of the *SVGD* dynamics. For instance, notice how $\epsilon^l_{\theta_3}$ converges to 0 in the setup of Fig. 4. Note that we don't learn a dimension-wise step-size as $\phi$ is by design the direction of maximum decrease for reverse $D_{\mathrm{KL}}$ Liu & Wang (2016). To satisfy the bound from Corr. 3.3, we truncate the learned step size at every step $l \in [0, L-1]$, *i.e.*, $\epsilon^l_{\theta_3} \leftarrow \min(\epsilon^l_{\theta_3}, \epsilon^l_{\mathrm{UB}})$. Finally, note that $\epsilon^l_{\theta_3} \to 0$ is a necessary but not a sufficient condition for $q^L_\theta$ converging to $p$. By Theorem 3.1 of Liu & Wang (2016), convergence further requires the gradient of the KL-divergence with respect to $\epsilon_{\theta_3}$ to vanish, $\nabla_{\epsilon_{\theta_3}} \mathrm{KL}(q^L_{\theta_3}\|p) = 0$, as this is equivalent to satisfying Stein's identity (see App. B.4.1).

**Number of Steps** $L$ is fixed in *P-SVGD*, which does not guarantee convergence to the target. To address this, *MET-SVGD* employs an adaptive number of steps $L_c$ determined dynamically by measuring violation of Stein's identity (App. A.6-A.8) for $\phi$ in Eq. 1. We evaluate this violation efficiently using the precomputed quantities $\nabla_{x^l} \log p(x^l)$ and $\operatorname{Tr}(\nabla_{x^l}\phi(x^l))$ from Eq. 3 (Prop. 3.4). The proof is in App. F. This yields a major advantage over MCMC samplers, which rely on approximate chain diagnostics that are more expensive and less reliable for assessing convergence (Robert, 1999).

**Proposition 3.4.** *Let $p(x) = \bar{p}(x)/Z$ be a continuously differentiable and strictly positive density on $\mathcal{X} \subset \mathbb{R}^d$ with $\lim_{\|x\|\to\infty} p(x)\,\phi(x) = 0$, and $\phi(x)$ the SVGD velocity field with an RBF kernel from Eq. 1 satisfying the step-size bound in Corr. 3.3, Stein's identity violation $SI(q^l, p)$ at every step $l$ is:*

$$SI(q^l, p) = \sqrt{\mathbb{E}_{x^l}\left[\phi(x^l)^T \nabla_{x^l} \log p(x^l) + \operatorname{Tr}(\nabla_{x^l}\phi(x^l))\right]} \tag{6}$$

**Complexity.** Extending *P-SVGD* to learn $\sigma^l_{\theta_2}$ and $\epsilon^l_{\theta_3}$ with lightweight deep nets adds negligible training cost and reduces inference to a single forward pass. In contrast, *SVGD* and *P-SVGD* rely on grid search for $L$ and $\epsilon$, which is computationally expensive and suboptimal. Also, $\sigma_{\mathrm{med}}$ scales quadratically with the number of particles due to pairwise distance computations. For $L_c$, naïve backpropagation through every *SVGD* update leads to linear memory growth with the number of steps. To address this, we backpropagate only every $k$ steps in large-scale experiments, exploiting the fact that particle positions change marginally between consecutive updates under smooth targets.

### 3.3 CORRECTED DERIVATION OF $q^l_\theta$

As explained in Sec. 3, *P-SVGD* approximates the expectation over particles in Eq. 3 by excluding the updated particle itself ($x^l \neq x^l_j$), so that the trace term can be estimated using only first-order derivatives and thereby avoiding explicit Hessian computation, *i.e.*, $\nabla^2_x \log p_\theta(x)$. While this approximation is valid asymptotically, it breaks in the finite-particle regime. As illustrated in Fig. 5, this translates into inconsistent updates across particles, as the crossed term is missing, inducing a different distribution for every particle, and making the entropy ill-defined. Additionally, in *P-SVGD*, the expectation over particles is handled inconsistently: the density estimation in Eq. 2 excludes the updated particle, while the update rule in Eq. 1 includes all particles resulting in a mismatch between the sampling process and its corresponding density computation. This inconsistency is a key source of *P-SVGD*'s poor scalability as shown in Fig. 2d (orange vs red). To address this, *MET-SVGD* corrects the approximation by adding the missing term to the entropy using ①

Figure 5: Correction Term. In *P-SVGD*, excluding the updated particle (crossed) when approximating the expectation in Eq. 2 is incorrect: particles undergo different updates leading them to follow different distributions, which makes the estimation of the target's entropy incorrect. *MET-SVGD* incorporates the missing term in the entropy estimate using Hutchinson's estimator (Sec. 3.3).

Hutchinson's estimator (Hutchinson, 1989) and ② double differentiation trick (Song et al., 2020):
$\text{Tr}\left(\nabla^2_{x^l_i}\log p(x^l_i)\right) \overset{①}{=} \mathbb{E}_{v\sim p_v}\left[v^T\nabla^2_{x^l_i}\log p(x^l_i)v\right] \overset{②}{=} \mathbb{E}_{v\sim p_v}\left[\nabla_{x^l_i}\left(v^T\nabla_{x^l_i}\log p(x^l_i)\right)v\right]$, where $p_v$ is chosen such that $\mathbb{E}[v]=0$ and $\mathbb{E}[vv^T]=I$ (*e.g.*, $p_v$ is the Radamacher distr). Importantly, *SVGD* is less sensitive to trace approximation errors compared to other MCMC methods (*e.g.*, *LD*) as shown in Fig. 2d (red vs gray). Notably, the trace term in *SVGD* is scaled by the number of particles $M$:

$$\log\hat{q}^L_\theta(x^L)=\log q^0_{\theta_1}(x^0)+\epsilon_{\theta_3}\sum_{l=0}^{L-1}\sum_{x^l_j\neq x^l}\text{Tr}\left(\frac{\partial\bar{\phi}_\theta(x^l,x^l_j)}{\partial x^l}\right)+\frac{\epsilon_{\theta_3}}{MV}\sum_{v=0}^{V-1}\nabla_{x^l}\left(v^T\nabla_{x^l}\log\bar{p}(x^l)\right)v+\mathcal{O}(\epsilon^2_{\theta_3}),$$

where $\epsilon_{\theta_3}=\max_l\epsilon^l_{\theta_3}/\epsilon^l_{\text{UB}}$, as we show in App. D.4. In contrast, in *LD*, the log-density

$$\log\hat{q}^L_\theta(x^L)=\log q^0_{\theta_1}(x^0)+\frac{\epsilon^l_{\theta_3}}{V}\sum_{l=0}^{L-1}\sum_{v=0}^{V-1}\nabla_{x^l}(v^T\nabla_{x^l}\log p(x^l))v+\mathcal{O}(\epsilon^2_{\theta_3}),$$

depends only on the trace approximation (proof in App. C.5). Hence, by incorporating this correction, *MET-SVGD* improves scalability, enhances the accuracy of entropy estimation, and ensures consistency between the sampling dynamics and the associated density derivation. **Complexity.** Only one additional first-order derivative and two vector dot products per sample are required. In practice, we find that a single sample $v$ is typically sufficient.

## 3.4 DIVERGENCE CONTROL VIA METROPOLIS HASTINGS

In many applications, $\bar{p}$ is modeled as a deepnet and learnt end-to-end, which can result in regions with abrupt gradients causing divergence during sampling. To prevent this, *P-SVGD* introduces a heuristic that removes particles deviating beyond a fixed number of standard deviations from the mean of the initial Gaussian distribution $q^0_{\theta_1}$ (Fig. 3b). Intuitively, the initial distribution approximates the support of the target, and particles that stray too far are likely to be out-of-distribution. Yet, this heuristic has many limitations: it ($i$) exacerbates mode collapse by discouraging exploration of distant modes (Fig. 2c), ($ii$) is inefficient, as replacing rejected particles requires restarting the sampling chain, and ($iii$) prevents divergence but does not improve convergence to non-smooth targets (Fig. 2b). To address this limitation, we introduce a principled divergence-control mechanism based on Metropolis-Hastings. After each update, the proposed state $\tilde{x}^l$ is accepted with probability $\alpha^l_\theta$, i.e., $x^l=\tilde{x}^l$, and otherwise the chain remains at the previous state, $x^l=x^{l-1}$, with probability $1-\alpha^l_\theta$. To obtain a Markov chain with strong convergence guarantees (App. B.5–B.6), we extend the SVGD state space with a Rademacher auxiliary variable $r\in\{-1,1\}$, analogous to the momentum-flip mechanism in Hamiltonian Monte Carlo Betancourt (2017). At *MET-SVGD* step $l$, we first sample $r^l$. When $r^l=1$, the proposal $\tilde{x}^l$ is generated via the forward SVGD update, *i.e.*, $\tilde{x}^l=x^{l-1}+\epsilon_{\theta_3}\phi_\theta(x^{l-1})$. When $r^l=-1$, we instead apply the corresponding inverse update, computed via the Banach fixed-point operator described in Alg. 3. This construction yields a reversible proposal distribution, and the acceptance probability enforcing detailed balance is computed efficiently by leveraging $\text{Tr}(\nabla_{x^l}\phi_\theta(x^l))$. We provide the proof in App. G.2.

**Proposition 3.5.** *Let $p(x)=\bar{p}(x)/Z$ be a strictly positive and continuously differentiable density on $\mathcal{X}\subset\mathbb{R}^d$, $\phi(x)$ the SVGD velocity field with an RBF kernel from Eq. 1 satisfying the step-size bound in Corr. 3.3 and $r^l\in\{-1,1\}$ the Rademacher auxiliary variable selecting the forward ($r^l=1$) or backward ($r^l=-1$) SVGD update in MET-SVGD. The log-likelihood of the MH acceptance probability for a MET-SVGD update of a particle $x^{l-1}$ is:*

$$\log\alpha^l_\theta=\min\left[0,\log\bar{p}(\tilde{x}^l)-\log\bar{p}(x^{l-1})+r^l\epsilon_{\theta_3}\text{Tr}(\nabla_{x^l}\phi_\theta(x^l))+\mathcal{O}\left((\epsilon^l_{\theta_3}/\epsilon^l_{UB})^2\right)\right].$$

We derive the *MH-augmented density* over particles after incorporating MH step (proof in App. G.2).

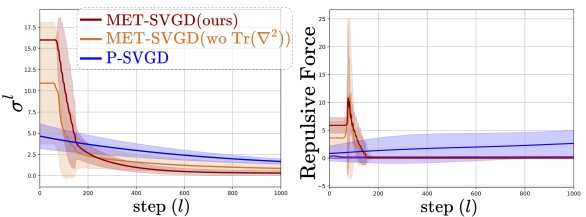
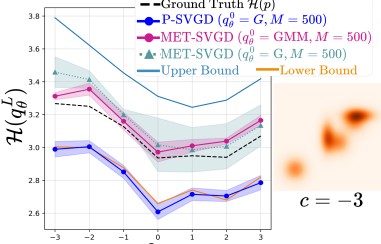

(a) Kernel bandwidth $\sigma^l$ and repulsive force ($2^{\text{nd}}$ term in Eq. 1) across *SVGD* steps $l$ for Gaussian target $p = \mathcal{N}(0, I_d)$ and initial distribution $q^0 = \mathcal{N}(2\mathbb{1}, 2I_d)$ with $d = 100$. This is the same setup of Fig. 2d.

(b) Target is a GMM with five components; the fifth one has a moving mean $\boldsymbol{\mu}_5 = (c, c)$ with $c \in [-3, 3]$ (App. H.2).

Figure 6: Results of entropy estimation on Gaussian and GMM targets

**Proposition 3.6.** *Under the setup of Prop. 3.5, the MH-augmented density at step $l$ is computed as:*
$q_\theta^{MH,l}(x^l) = \frac{1}{2} q_\theta^{MH,l}(x^l | r^l = 1) + \frac{1}{2} q_\theta^{MH,l}(x^l | r^l = -1)$, *where for $c \in \{-1, 1\}$:*

$$q_\theta^{MH,l}(x^l | r^l = c) = \alpha_\theta^l q_\theta^{MH,l-1}(x^{l-1}) \left| \det\left( I + \epsilon_{\theta_3} \nabla_{x^{l-1}} \phi_\theta(x^{l-1}) \right) \right|^{-c} + (1 - \alpha_\theta^l) q_\theta^{MH,l-1}(x^{l-1}). \quad (7)$$

Note that $q_\theta^{\text{MH},l}$ is a mixture of *SVGD*-based distributions over different paths, significantly enriching the expressiveness of the variational family. Also, in setups where $\bar{p}$ is learned end-to-end, $\alpha_\theta^l$ naturally down-weighs poor updates in non-smooth regions.

**Convergence guarantees.** *MET-SVGD* is an MH algorithm with an efficient *SVGD*-based proposal distribution. It therefore inherits asymptotic *convergence guarantees* from MH, *i.e.*, for $L \to \infty$, $q_\theta^{\text{MH},L}$ *converges strongly* to the target $p$ independently from the number of particles $M$ (Mengersen & Tweedie, 1996). Differently, *SVGD* requires $L, M \to \infty$ for *weak convergence* (Sun et al., 2023). The impact of MH on the bounds for *SVGD* convergence rate in the finite particles setup is not trivial and is the subject of future work.

**Complexity.** The MH step introduces an additional $\mathcal{O}(L_c B/2)$ back-propagations in expectation, stemming from the reverse SVGD updates computed via the Banach fixed-point theorem (Alg. 3). In practice, $L$ remains small in our experiments due to learning an expressive initial distribution.

We summarize the *MET-SVGD*'s significance for approximate inference in Tab. 1, Figs. 9 and 10, and its novelty over *P-SVGD* in Tab. 2.

## 4 EXPERIMENT

### 4.1 ENTROPY ESTIMATION ON GAUSSIAN AND GMM TARGETS

*MET-SVGD* consistently outperforms *P-SVGD* across Gaussian (Figs. 2a, 2d and 17) and GMM (Figs. 6b and 23) setups. Notably, Figs. 2d and 23 show that, while *P-SVGD* and projection-based variants such as *S-SVGD* (Gong et al., 2021) and *GSVGD* (Liu et al., 2022b) struggle to scale beyond 20-$d$, $\beta$-*SVGD* (Sun & Richtárik, 2022) shows improved behavior, yet still underperforms *MET-SVGD* in high dimensions. *MET-SVGD* achieves high accuracy in up to 1000-$d$. Notice that *MET-SVGD* mitigates the vanishing repulsive force (Fig. 6a) previously identified as the root cause of *SVGD*'s poor scalability Ba et al. (2022). These gains are enabled by learning $\sigma_{\theta_2}^l$ ($C_2$), as indicated by the trend difference compared to $\sigma_{\text{med}}^l$ (Fig. 6a), and restoring the correction term ($C_4$) as depicted in Fig. 2d. Also, as shown in Fig. 2b, MH helped with non-smooth targets ($C_5$).

### 4.2 LEARNING ENERGY-BASED MODELS

Training EBMs $p_\phi(x) = \bar{p}_\phi(x)/Z$ via maximum likelihood is, in general, intractable due to the partition function $Z$. When the sampler has a tractable distribution $q_\theta$, a tight lower bound can be computed: $\mathcal{L}_{\text{ELBO}}(\phi, \theta) = \mathbb{E}_{x \sim q_\theta}[\log \bar{p}_\phi(x)] - \mathbb{E}_{x \sim p_d}[\log \bar{p}_\phi(x)] + \mathcal{H}(q_\theta)$, as detailed in App. I. The entropy is often omitted as it's not trivial to compute for popular samplers (*e.g.*, *LD*), the loss is then reduced to the commonly used contrastive divergence loss $\mathcal{L}_{\text{CD}}(\phi)$. We optimize $\mathcal{L}_{\text{ELBO}}(\phi, \theta)$ using *P-SVGD*, *MET-SVGD* and Glow-NF (Kingma & Dhariwal (2018)), and *LD* trained with $\mathcal{L}_{\text{CD}}(\phi)$. We evaluate on the *Moon dataset* (Rezende & Mohamed, 2015) (Fig. 24), where we show that incorporating MH ($C_5$) significantly improves accuracy as smoothness of the target distribution decreases. For CIFAR-10, we report the Fréchet Inception Distance (FID) averaged over 5 random seeds, as well as a stability score, measured by the standard deviation of FID from its best achieved value until the end of training (Tab. 16 in App. I).

In Fig. 7, we show that not including the trace of Hessian in *MET-SVGD* $(C_4)$ or not incorporating the step-size bound $(C_1)$ leads to divergence (violet, gray). Fig. 27 shows that runs with the trace term learn smoother landscapes which are more convenient for sampling. Without the step-size bound, the entropy derivation is not correct. Replacing $\sigma_{\text{med}}$ with the learnable one $(C_2)$ improves stability and yields significantly better FID scores relative to *P-SVGD* (green vs orange). In Fig. 29b, we show that $\sigma_{\text{med}}$ is in average more than an order of magnitude higher than $\sigma_{\theta_2}$ in the working configurations, leading to spuriously correlating several particles. Additionally, learning the step-size $(C_2)$ (red) enables faster convergence to the target ($\epsilon^l_{\theta_3} \gg \epsilon$ in Fig. 29) and results in smoother land-

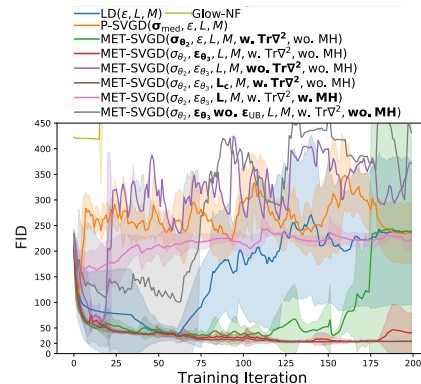

Figure 7: FID on CIFAR-10; bold marks changes between configurations.

scapes (Fig. 27). Using an adaptive number of steps $L_c$ further stabilizes the training $(C_3)$, leading to $64\times$ improvement over *P-SVGD* (brown) as depicted in Tab. 16. Yet, experiments with MH $(C_5)$ diverge (pink). This issue is not inherent to *MET-SVGD* itself; rather, it arises from the combination of MH rejection dynamics and the instability of the contrastive divergence loss variants. Specifically, $\mathcal{L}_{\text{ELBO}}$ is not lower-bounded (can be driven to $-\infty$) and thus well-known for being unstable. Its stability depends critically on the quality of samples used to approximate the first expectation. When the underlying energy landscape is highly complex, the MH acceptance rate can be low during the initial stages of training (Fig. 28). Hence, the particles fail to move toward the high-density regions of the target. This leads to poor samples and eventually divergence. Similarly, Glow-NF struggles to generate good samples early in training and diverges. Normalizing flows are known to be difficult to train and often exhibit instability in practice (Vaitl et al., 2022). Qualitative results and implementation details are in App. I.

### 4.3 MAX-ENTROPY RL

Unlike classical RL, which learns a deterministic policy (Sutton et al., 1999), MaxEnt RL (Ziebart, 2010) learns a stochastic policy $\pi_\theta$ by maximizing the sum of expected rewards and entropies: $\pi_\theta^* = \arg\max_{\pi_\theta} \sum_t \mathbb{E}_{(s_t,a_t)}[r(s_t, a_t) + \alpha\mathcal{H}(\pi_\theta(\cdot|s_t))]$. Following Messaoud et al. (2024), we model the policy as an *SVGD* sampler and estimate the entropy using *MET-SVGD* on Walker2d-v2 and Humanoid-v2

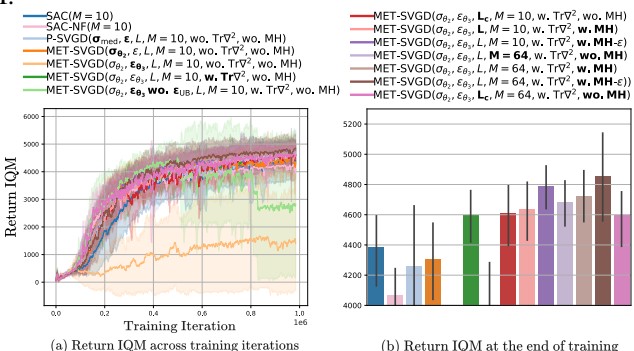

Figure 8: IQM return on Walker2d-v2 environment.

(Brockman et al., 2016). We also compare to SAC (Haarnoja et al., 2018) and SAC-NF, which model the policy as a Gaussian and autoregressive flows (Papamakarios et al., 2017), respectively. We train 5 different instances of each algorithm with different seeds and report the average return on 10 rollouts every 1000 steps in Fig. 31. Ablations show that learning the kernel bandwidth $(C_2)$ and restoring the correction term $(C_4)$ substantially improve performance. Removing the step-size bound $(C_1)$ leads to divergence, as expected. Learning the step size without the correction term underperforms all baselines, due to samples divergence in non-smooth landscapes (see Fig. 41). Using an adaptive number of *SVGD* steps $(C_3)$ slightly reduces returns as it decreases initial exploration. Incorporating an MH step $(C_5)$ with an $\epsilon$-greedy schedule, *i.e.*, high MH probability later and lower early, performs best as it preserves early exploration and improves later exploitation. The gains become more pronounced when increasing particles from 10 to 64 (brown). In Fig. 33b, we show that SAC-NF exhibits strong mode collapse as it saturates early during training leading to limited exploration and low returns. Additional results are in App. J.

### 5 CONCLUSION

*MET-SVGD* advances the *P-SVGD* lineage by diagnosing key limitations and introducing principled fixes that deliver consistent empirical gains. It, thus, addresses two long-standing challenges: (i) estimating the entropy of targets known only up to a normalization constant, and (ii) systematic tuning of sampler parameters. These capabilities are highly desired for scientific applications that operate on unnormalized densities, *e.g.*, molecular design in drug discovery.

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

# Supplementary Material

*MET-SVGD* is a novel variational inference approach for entropy estimation that overcomes key limitations of *P-SVGD* Messaoud et al. (2024), particularly poor convergence and scalability in high-dimensional spaces (Fig. 2). To achieve this, it introduces: (1) A sufficient condition for global invertibility; (2) Optimized parameter search for improved stability (Sec. 3.2); (3) Metropolis-Hastings augmented *SVGD* updates to ensure asymptotic convergence (Sec. 3.4). (4) A correction term to the density estimation in *P-SVGD* (Sec. 3.3). We summarize the novelty over *P-SVGD* in Tab. 2. *MET-SVGD* maintains computational efficiency, requiring no significant additional memory or runtime overhead. Its full workflow is illustrated in Alg. 2 and Alg. 1. Beyond entropy estimation,

- *MET-SVGD* bridges the gap between Metropolis-Hastings algorithms (MH) Robert et al. (2004), particle-based sampling techniques (*SVGD*) Liu & Wang (2016), and parametrized variational inference (P-VI) Fox & Roberts (2012), leveraging the strengths of each (Tab. 1): (1) scalability from P-VI, (2) expressivity, convergence detection, and particle efficiency from *SVGD*, as well as (3) convergence guarantees from MH. See Fig. 7
- *MET-SVGD* is a new approach for unprecedentedly scaling *SVGD* to high-dimensional spaces while being computationally more efficient than all proposed approaches in the literature Gong et al. (2021); Liu et al. (2022b).
- *MET-SVGD* is a new approach for end-to-end learning of sampler parameters. It enables training samplers via KL-divergence minimization, achieving compelling results for both *LD* and *SVGD* (Fig. 10).
- *MET-SVGD* is a new normalizing flow model with (1) an adaptive number of updates controlled by a convergence check and (2) a full-rank Jacobian for improved flexibility and expressivity (Fig. 11).

Our code is available at `https://tinyurl.com/2esyfx8j`.

|  | *P-VI* | *MCMC* | *SVGD* | *P-SVGD* | *MET-SVGD* |
|---|---|---|---|---|---|
| Expressivity | ✗ | ✓ | ✓ | ✓ | ✓✓ |
| Convergence Detection | ✓ | ✗ | ✓ | ✓ | ✓ |
| Convergence Guarantees | ✗ | ✓ | ✗ | ✗ | ✓ |
| Sampling Efficiency | ✓ | ✗ | ✓ | ✓ | ✓ |
| Tractable Entropy | ✓ | ✗ | ✗ | ✓ | ✓ |
| Parameter Efficiency | ✓ | - | - | ✓✓ | ✓✓ |

Table 1: *MET-SVGD* inherits advantages of different approximate inference methods: *P-VI*, *SVGD*, and *MCMC*.

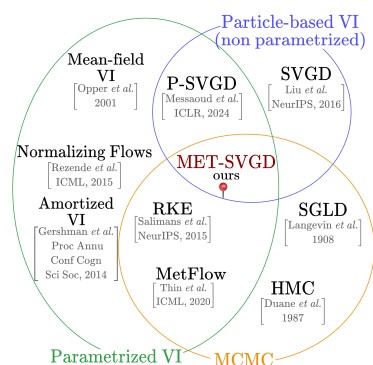

Figure 9: *MET-SVGD* bridges the gap between parametrized variational inference (*P-VI*), particle-based variation inference (*SVGD*), and *MCMC* methods, inheriting the strengths of each.

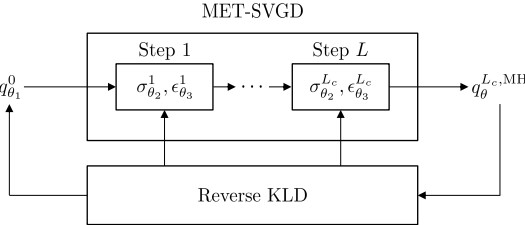

Figure 10: *MET-SVGD* is a principled approach to learn samplers' parameters via estimating the particles' induced density and minimizing the reverse KLD between this density and the unnormalized target.

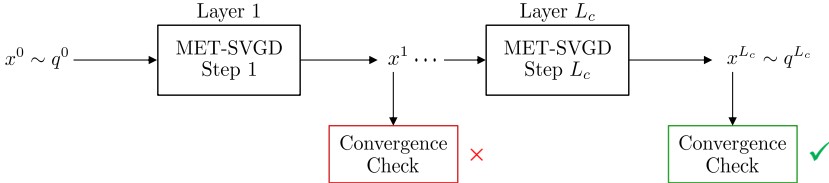

Figure 11: *MET-SVGD* is a normalizing flow model with a full rank Jacobian and an adaptive number of layers.

| | P-SVGD | MET-SVGD |
|---|---|---|
| Invertibility Condition | Local, imprecise: $\epsilon \ll \sigma$ (Implicit Function Theorem; Prop. 3.2 in *P-SVGD* paper) | Global, precise: $\epsilon < \epsilon^{\text{UB}}$ (Banach theorem; Cor. 3.3) |
| Entropy Trace Approximation | Requires $\epsilon \|\nabla\phi^l\|_\infty \ll 1$ (Thm. 3.1 in *P-SVGD* paper) | Automatically satisfied under invertibility constraint (Cor. 3.3) |
| Divergence Control | Heuristic: particle truncation based on distance to $q_0$'s mean | Metropolis–Hastings step (Sec. 3.4) |
| Tr(Hessian) in Entropy | Omitted; invalid approximation in finite particle setup (Thm. 3.3) | Restored via Hutchinson's estimator (Sec. 3.3) |
| Kernel Bandwidth $\sigma$ | Median heuristic $\mathcal{O}(M^2)$ | Learned end-to-end via lightweight deepnets (Sec. 3.2) |
| Step Size $\epsilon$ | Fixed | Learned end-to-end via lightweight deepnets (Sec. 3.2) |
| Number of Steps $L$ | Fixed | Adaptive via Stein Identity (Sec. 3.2) |
| Computational Complexity | Grid search for $\epsilon$; median heuristic for $\sigma$ | Efficient reuse of $\text{Tr}(\nabla\phi^l)$ for invertibility, MH correction, and convergence check; deepnet inference adds minor overhead |
| Memory Complexity | — | Two lightweight deepnets for $\sigma_{\theta_2}$ and $\epsilon_{\theta_3}$ (Sec. 3.2) |
| Convergence Guarantees | Asymptotic: $L, M \to \infty$ | Asymptotic: $L \to \infty$ |
| Empirical Performance | Sensitive to hyperparameters; mode collapse; poor scalability to non-smooth/high-d targets (Figs. 2a–2d) | <ul><li>Gaussian/GMM: $12\times$ improved accuracy over *P-SVGD* (Figs. 4,6b)</li><li>EBM-based image generation: 80.4% better FID compared to *P-SVGD*, and $64\times$ improved training stability (Fig. 7)</li><li>MaxEntr RL: 16% better returns than *P-SVGD* (Fig. **??**)</li></ul> |

Table 2: Comparison: *P-SVGD* vs. *MET-SVGD*.

The rest of the appendix is organized as follows:

- Appendix A: **Preliminaries** including Langevin Dynamics, the Change of Variable formula for probability densities, the implicit function theorem, Jacobi's formula's corollary, the mean value theorem, Stein's identity, Stein discrepancy, and kernelized Stein discrepancy
- Appendix B: **Additional related work and background** on entropy estimation, sampling-based variational inference, Normalizing Flows, *SVGD*, and Metropolis-Hastings
- Appendix C: **Derivation of closed-form density expressions for *LD* and *SVGD* samplers** using RBF, Bilinear, and DKEF kernels
- Appendix D: **Derivation of step-size bounds**
- Appendix E: **Motivation for learning the step-size**
- Appendix F: **Derivation of the convergence check**
- Appendix G **Derivation of the Metropolis-Hastings augmented entropy**
- Appendix H: **Additional results on entropy estimation**
- Appendix I: **Additional results on learning EBMs for image generation**
- Appendix J: **Additional results on MaxEntr RL**

## A PRELIMINARIES

In the following, we review preliminaries about Langevin Dynamics, the Change of Variable formula for pdfs, the Implicit Function theorem, the corollary of Jacobi's formula, the Mean Value theorem, Stein's Identity, Stein discrepancy, and kernelized Stein discrepancy.

### A.1 LANGEVIN DYNAMICS (*LD*)

*LD* (Welling & Teh, 2011) is a popular Markov chain Monte Carlo (MCMC) method for sampling from a distribution. Let $p$ be a differentiable density on $\mathbb{R}^d$. *LD* first initializes a sample $x^0$ from a random initial distribution. Then at every step, it adds the gradient of the proposal distribution $p(x^l)$ to the previous sample $x^l$, together with a Brownian motion $\xi \sim \mathcal{N}(0, I)$. We denote with $\epsilon$ the step size. The iterative update for *LD* is:

$$x^{l+1} = x^l + \epsilon \nabla_{x^l} \log p(x^l) + \sqrt{2\epsilon} \xi. \tag{8}$$

### A.2 CHANGE OF VARIABLE FORMULA (CVF)

Let $X \in \mathcal{X}$ and $Z \in \mathcal{Z}$ be two random variables with densities $p_X$ and $p_Z$. If $f : Z \to X$ is a bijective, differentiable function, and $X = f(Z)$, then

$$p_X(x) = p_Z(z) \left| \det \frac{\partial f^{-1}(x)}{\partial x} \right| = p_Z(z) \left| \det \frac{\partial f(z)}{\partial z} \right|^{-1}$$

### A.3 IMPLICIT FUNCTION THEOREM

Let $f : \mathbb{R}^n \to \mathbb{R}^n$ be continuously differentiable on some open set containing $a$, and suppose $\det (\nabla_x f(x)) \neq 0$. Then, there is some open set $V$ containing $x$ and an open $W$ containing $f(x)$ such that $f : V \to W$ has a continuous inverse $f^{-1} : W \to V$ which is differentiable $\forall y \in W$.

### A.4 COROLLARY OF JACOBI'S FORMULA

Given an invertible matrix $A \in \mathbb{R}^{d \times d}$, the following equality holds:

$$\log(\det A) = \mathrm{Tr}\left(\log A\right) = \mathrm{Tr}\left(\sum_{k=1}^{\infty} (-1)^{k+1} \frac{(A-I)^k}{k}\right). \tag{9}$$

The second equation is obtained by taking the power series of $\log A$. Hence, under the assumption $\|A - I\|_\infty \ll 1$, we obtain: $\log(\det A) = \mathrm{tr}(A - I) + \mathcal{O}(\epsilon^2)$, where $\|\cdot\|_\infty$ is the infinity norm.

### A.5 THE MEAN VALUE THEOREM

Let $f : \mathbb{R}^d \to \mathbb{R}$ be differentiable on $\mathbb{R}^d$ with a Lipschitz continuous gradient $\nabla f$. Then for given $x$ and $\bar{x}$ in $\mathbb{R}^d$, there is $y = x + t(x - \bar{x})$ with $t \in [0, 1]$, such that

$$f(x) - f(\bar{x}) = \nabla_y f(y) \cdot (x - \bar{x}).$$

### A.6 STEIN'S IDENTITY

Let $p(x)$ be a smooth, strictly positive density supported on $\mathcal{X} \subseteq \mathbb{R}^d$, and $\phi(x) = [\phi_1(x), \cdots, \phi_d(x)]^T$ be a smooth vector-valued function that satisfies either $p(x)\phi(x) = 0$ for all $x \in \partial\mathcal{X}$ if $\mathcal{X}$ is compact, or $\lim_{\|x\|\to\infty} p(x)\phi(x) = 0$ if $\mathcal{X} = \mathbb{R}^d$. Stein's identity states that:

$$\mathbb{E}_{x\sim p}[\mathcal{A}_p \phi(x)] = 0,$$

where the Stein operator $\mathcal{A}_p$ is defined as:

$$\mathcal{A}_p \phi(x) = \phi(x)\nabla_x \log p(x)^\top + \nabla_x \phi(x) \tag{10}$$

*Proof.* In the following we assume $\mathcal{X} = [a, b]$:

$$\mathbb{E}_{x \sim p}[\mathcal{A}_p \phi(x)] = \int_a^b \left( p(x)\phi(x)\nabla_x \log p(x)^\top + p(x)\nabla_x \phi(x) \right) dx$$

$$\stackrel{(i)}{=} \int_a^b \left( \phi(x)\nabla_x p(x)^\top + p(x)\nabla_x \phi(x) \right) dx$$

(*i*) Uses the identity $\nabla_x \log p(x) = \frac{\nabla_x p(x)}{p(x)}$.

Since the integral of a matrix is the matrix of integrals, we compute the integral for the element indexed by $i, j$ as

$$\int_a^b \left( \phi_i(x)\nabla_{x_j} p(x)^\top + p(x)\nabla_{x_j} \phi_i(x) \right) dx \stackrel{(ii)}{=} [p(x)\phi_i(x)]_a^b \stackrel{(iii)}{=} 0$$

(*ii*) Applies integration by parts: $\int_a^b f(x)g'(x) + f'(x)g(x)\, dx = [f(x)g(x)]_a^b$

(*iii*) Boundary term vanishes under the aforestated assumptions

$\square$

### A.7 STEIN DISCREPANCY

Given the same setup in Sec. A.6, Stein discrepancy between two distributions $q$ and $p$ is defined as the maximum squared violation of Stein's identity:

$$\mathbb{S}(q, p) = \max_{\phi \in \mathcal{F}} \left\{ \left( \mathbb{E}_{x \sim q} \left[ \text{Tr} \left( \mathcal{A}_p \phi(x) \right) \right] \right)^2 \right\},$$

where $\mathcal{F}$ is a set of functions with bounded Lipschitz norms.

Intuitively, since Stein's identity ensures that $\left( \mathbb{E}_{x \sim p} \left[ \text{Tr} \left( \mathcal{A}_p \phi(x) \right) \right] \right)^2 = 0$, replacing the expectation with respect to $p$ with the expectation with respect to $q$ provides information about how different $q$ is form $p$.

### A.8 KERNELIZED STEIN DISCREPANCY

Given the same setup in Sec. A.7, kernelized Stein discrepancy is a special case of the Stein Discrepancy that results from setting $\mathcal{F} = \mathcal{H}^D = \underbrace{\mathcal{H} \times \cdots \times \mathcal{H}}_{D \text{ times}}$, where $\mathcal{H}$ is a Rreproducing Kernel Hilbert Space (RKHS) with a corresponding kernel $k(x, y)$:

$$\mathbb{S}(q, p) = \max_{\phi \in \mathcal{H}^D} \left\{ \left( \mathbb{E}_{x \sim q} \left[ \text{Tr} \left( \mathcal{A}_p \phi(x) \right) \right] \right)^2 \quad \text{s.t.} \quad ||\phi||_{\mathcal{H}^D} \le 1 \right\},$$

where $||\phi||_{\mathcal{H}^D}$ is defined by the vector dot product $||\phi||_{\mathcal{H}^D}^2 = \langle \phi, \phi \rangle_{\mathcal{H}^D} = \sum_d \langle \phi_d, \phi_d \rangle_{\mathcal{H}}$.

The maximizer of the Kernelized Stein Discrepancy is $\phi_{q,p} = \phi_{q,p}^* / ||\phi_{q,p}^*||_{\mathcal{H}^D}$, where $\phi_{q,p}^*(\cdot) = \mathbb{E}_{x \sim q}[\mathcal{A}_p k(x, \cdot)]$, and $\mathcal{A}_p$ is the stein operator (Eq. 10). Moreover, $\mathbb{S}(q, p) = ||\phi_{q,p}^*||_{\mathcal{H}^D}^2$.

*Proof.* We begin by deriving the maximizer of the kernelized Stein discrepancy, then we show that $\mathbb{S}(q, p) = ||\phi_{q,p}^*||_{\mathcal{H}^D}^2$.

**Derivation of the Maximizer of $\mathbb{S}(q, p)$:**

$$\mathbb{E}_{x \sim q}[\text{Tr} \left( \mathcal{A}_p \phi(x) \right)] \stackrel{(i)}{=} \sum_d \mathbb{E}_{x \sim q} \left[ \phi_d(x)\nabla_{x_d} \log p(x) + \nabla_{x_d} \phi_d(x) \right]$$

$$\stackrel{(ii)}{=} \sum_d \mathbb{E}_{x \sim q} \left[ \langle \phi_d(\cdot), k(x, \cdot) \rangle_{\mathcal{H}} \nabla_{x_d} \log p(x) + \langle \phi_d(\cdot), \nabla_{x_d} k(x, \cdot) \rangle_{\mathcal{H}} \right]$$

$$\stackrel{(iii)}{=} \sum_d \langle \phi_d(\cdot), \mathbb{E}_{x \sim q} \left[ k(x, \cdot) \nabla_{x_d} \log p(x) + \nabla_{x_d} k(x, \cdot) \right] \rangle_{\mathcal{H}}$$

$$\stackrel{(iv)}{=} \langle \phi(\cdot), \mathbb{E}_{x \sim q} \left[ \mathcal{A}_p k(x, \cdot) \right] \rangle_{\mathcal{H}^D}$$

$(i)$ by definition of the Stein operator

$(ii)$ by the reproducing property of the RKHS $\mathcal{H}$

$(iii)$ by linearity of the expectation and the inner product

$(iv)$ by definition of the inner product in the RKHS $\mathcal{H}^D$

The $\phi$ that maximizes this inner product under the constraint $||\phi||_{\mathcal{H}^D} \leq 1$ is the one proportional to $\mathbb{E}_{x \sim q} \left[ \mathcal{A}_p k(x, \cdot) \right]$ with $||\phi||_{\mathcal{H}^D} = 1$, thus $\phi_{q,p} = \phi_{q,p}^* / ||\phi_{q,p}^*||_{\mathcal{H}^D}$, where $\phi_{q,p}^*(\cdot) = \mathbb{E}_{x \sim q} \left[ \mathcal{A}_p k(x, \cdot) \right]$.

This also shows that $\mathbb{E}_{x \sim q}[\text{Tr} \left( \mathcal{A}_p \phi(x) \right)] = \langle \phi(\cdot), \mathbb{E}_{x \sim q} \left[ \mathcal{A}_p k(x, \cdot) \right] \rangle_{\mathcal{H}^D}$ for $\phi \in \mathcal{H}^D$.

**Proof that $\mathbb{S}(q, p) = ||\phi_{q,p}^*||_{\mathcal{H}^D}^2$:**

We simply plug $\phi_{q,p}$ into $\mathbb{S}(q, p)$.

$$\mathbb{S}(q,p) \stackrel{(i)}{=} \left( \mathbb{E}_{x \sim q} \left[ \text{Tr} \left( \mathcal{A}_p \phi_{q,p}(x) \right) \right] \right)^2 \stackrel{(ii)}{=} \langle \phi_{q,p}(\cdot), \mathbb{E}_{x \sim q} \left[ \mathcal{A}_p k(x, \cdot) \right] \rangle_{\mathcal{H}^D}^2 \stackrel{(iii)}{=} \langle \phi_{q,p}(\cdot), \phi_{q,p}^*(\cdot) \rangle_{\mathcal{H}^D}^2$$

$$\stackrel{(iv)}{=} \left\langle \frac{\phi_{q,p}^*(\cdot)}{||\phi_{q,p}^*(\cdot)||_{\mathcal{H}^D}}, \phi_{q,p}^*(\cdot) \right\rangle_{\mathcal{H}^D}^2 = ||\phi_{q,p}^*(\cdot)||_{\mathcal{H}^D}^2$$

$(i)$ by definition of $\mathbb{S}(q, p)$

$(ii)$ using the second result from the previous derivation

$(iii)$ by definition of $\phi_{q,p}^*$

$(iv)$ by definition of $\phi_{q,p}$

$\square$

## B ADDITIONAL RELATED WORK

In the following, we review additional work on the differential entropy, sampling-based variational inference, Normalizing Flows, Stein Variational Gradient Descent (*SVGD*) and Metropolis Hastings (MH) convergence.

### B.1 DIFFERENTIAL ENTROPY

Differential entropy, first introduced by Shannon in his foundational work on information theory Shannon (1948), has been widely studied in statistics Box & Tiao (1992); Zellner (1971); Bernardo (1979). For a continuous random variable $z$ with density $p(z)$, and is defined as:

$$\mathcal{H}(x) = -\int_{-\infty}^{\infty} p(x) \log \big(p(x)\big)\, dx.$$

**Applications of Entropy:** Entropy plays a crucial role in machine learning, Bayesian inference (BI), reinforcement learning (RL), and variational inference (VI): $(i)$ In classification & calibration, the entropy can be used to measure the uncertainty of the model's predicted class probabilities (Shyam, 2019), and in active learning (Wu et al., 2022) it is used to select the most uncertain examples for labeling. $(ii)$ In Bayesian Inference, the Maximum Entropy principle ensures the least informative prior Bernardo (1979). $(iii)$ In Reinforcement learning, it is used to prevent overly deterministic policies by being incorporated into the reward function Hazan & Van Soest (2019); Ahmed et al. (2019). $(iv)$ In variational inference & generative Models, the entropy appears in ELBO Kingma & Welling (2013) for posterior approximation and has been used to mitigate mode collapse in GANs and VAEs Alemi et al. (2017); Belghazi et al. (2018).

**Challenges in Entropy Estimation:** Despite its simple definition, entropy is analytically tractable for limited familieis of distributions. For instance, for a uniform $p(x) = \frac{1}{b-a}$ with $x \in [a, b]$ and $p(x) = 0$ for $x \notin [a, b]$ the entropy is $\mathcal{H}(p) = \frac{1}{2}[1 + \log(2\pi\sigma^2)]$. For a Gaussian $p(x) = \mathcal{N}(\mu, \sigma^2)$, the entropy is $\mathcal{H}(p(y|\mu, \sigma^2) = \frac{1}{2}\big(1 + \log(2\pi\sigma^2)\big)$. For general distributions, numerical integration (*e.g.*, Monte Carlo when samples are available or sampling is possible) is required as direct computation is often infeasible.

Different methods have been developed. These methods can be classified into:

- *Plug-in Estimators:* Estimate density from data, then apply entropy formula. Given a sample $x = \{x_i\}_{i=0}^{M-1}$, the plug-in method estimates the pdf $\hat{p}(x)$ from the data and then substitutes this estimate into the entropy formula: $\mathcal{H}^{\text{PLUGIN}}(p) \approx -\frac{1}{M}\sum_{i=0}^{M-1} \log \hat{p}(x_i)$. This approach was first proposed by Dmitriev et al. Dmitriev & Tarasenko (1973) and later investigated by others using kernel density estimator Joe (1989); Hall & Morton (1993); Moon et al. (2018); Pichler (2022), histogram estimator Györfi & Van der Meulen (1987); Hall & Morton (1993) and field-theoretic approaches Chen et al. (2018). Early approaches leverage kernels that capture pairwise distances between the particles. For instance, Parzen-Rosenblatt estimator Rosenblatt (1956); Parzen (1962): $\hat{p}(x) = \frac{1}{w^p n}\sum_{i=0}^{M-1} \kappa\left(\frac{x-x_i}{w}\right)$, where $w$ denotes the bandwidth and $\kappa$ is a kernel density. To improve density estimation for non-negative random variables, recent studies have suggested replacing Gaussian kernels with Poisson weight-based estimators to fit counts or rate-based data Chaubey & Sen (2013) defined as: $\hat{p}^{\text{POIS}}(x) = k\sum_{i=0}^{\infty}\left(F_n(\frac{i+1}{k}) - F_n(\frac{i}{k})\right)e^{-kx}\frac{(kx)^i}{i!}$, where $F_n(.)$ is the empirical distribution function, and $k$ is a smoothing parameter. Traditional off-the-shelf density estimators, however, often suffer from key drawbacks, such as non-differentiability, computational intractability, or an inability to adapt to changes in the underlying data distribution. These limitations make them unsuitable for applications requiring integration into neural network training pipelines as regularizers. The idea of learning kernel parameters end-to-end has also been explored in Viola & Sejnowski (1995); Schraudolph (2004); Pichler (2022), providing a foundation for modern differentiable approaches. Schraudolph (2004) extended the approach from Rosenblatt (1956); Parzen (1962) using a learnable kernel estimator: $\hat{p}(x) = \frac{1}{M}\sum_{i=0}^{M-1} \kappa_{\Sigma_i}(x - x_i)$, where $\Sigma = (\Sigma_1, \cdots, \Sigma_n)$ are distinct

diagonal covariance matrices learned from the data either via maximum likelihood estimation or the expectation maximization algorithm, and $\kappa_\Sigma(x) \sim \mathcal{N}(0, \Sigma)$ is a centered Gaussian density with covariance matrix $\Sigma$. Pichler (2022) introduced KNIFE with a density estimator defined as: $\hat{p}^{\text{KNIFE}}(x; \theta) = \sum_{i=0}^{M-1} \mu_i \kappa_{\Sigma_i}(x - b_i)$, where $\theta = (\Sigma, b, \mu)$ and $\sum_{i=0}^{M-1} \mu_i = 1$. The covariance matrices $\Sigma_i$ are symmetric and positive definite but not necessarily diagonal. Despite its advantages, the method has a significant limitation in its simple structure, being restricted to either individual Gaussian kernels or Gaussian Mixture Models (GMMs) with a fixed number of components $n$. This can limit its flexibility in modeling complex data distributions.

- *Sample-spacing Estimates* use distances between ordered samples (*e.g.*, Vasicek estimator Vasicek (1976)). Sample spacing methods rely on the spacing of sorted samples and were first introduced by Vasicek Vasicek (1976): $H^{Vasicek}(p) \approx -\frac{1}{M} \sum_{i=0}^{M-1} \log\left( \frac{n}{2m}\left(x_{i+1} - x_i\right)\right)$, where $x_i$ are the order statistics and $m$ is a positive integer smaller than $\frac{n}{2}$. One of the greatest weakness of sample-spacing-based estimator is the choice of spacing parameter $m$, which does not have the optimal form.

- *Nearest-Neighbor Methods*: leverage distances to the $k$-th nearest neighbor in the sample space Kraskov (2004).

- *Variational Inference*: Optimizes a surrogate distribution $q(x)$ to approximate $p(x)$ Kingma & Welling (2013). The entropy is computed as Kingma & Welling (2013): $\mathcal{H}(p) \approx -\mathbb{E}_{q(x)}[\log q(x)]$, where $q(y)$ is optimized to approximate $p(x)$. $q$ is chosen to be easy to sample from, *e.g.*, Gaussians, GMMs and Normalizing Flows Rezende & Mohamed (2015).

- *Mutual Information (MI) Estimators*: Approximate entropy indirectly via MI, *i.e.*, $I(x, y) = \mathcal{H}(p_x) + \mathcal{H}(p_{x|y})$, where $p_{x|y}$ is the conditional distribution $p(x|y)$. Neural networks were used to approximate the mutual information between two variables using the Donsker-Varadhan representation of the KL-Divergence Donsker & Varadhan (1975): $D_{\text{KL}}(p\|q) = \sup_{T \in \mathcal{T}} \left( \mathbb{E}_p[T(x)] - \log \mathbb{E}_q[e^{T(x)}] \right)$, where $\mathcal{T}$ is a class of real-valued functions defined on the support of $p$ and $q$ such that the two expectations are finite. Belghazi et al. (2018) express a lower bound on the MI as: $I_\theta(x; y) = \sup_\theta \left( \mathbb{E}_{p_{x,y}}[T_\theta(x, y)] - \log \mathbb{E}_{p_x \cdot p_y}[e^{T_\theta(x, y)}] \right)$, where $T_\theta(x, y)$ is the output of a neural network parameterized by $\theta$, and $p_x \cdot p_y$ is the product of the marginals. $T_\theta$ is trained to maximize this lower bound, providing an approximation of $I(x; y)$. As shown in Kumar et al. (2019), MI can be used as a surrogate for the entropy in specific scenarios where it reduces to it, *i.e.*, when $\mathcal{H}(p_{x|y}) = 0$.

- *Ensemble Methods:* Weight different entropy estimators adaptively Sricharan et al. (2013). The estimators in the ensemble are assigned different weights, and the overall entropy estimate is calculated as a weighted combination of the individual estimators where optimal weights are determined by solving a convex optimization problem. Ariel & Louzoun (2020) proposed an innovative approach to estimating the entropy of high-dimensional data by decomposing the target entropy into two components: $\mathcal{H} = \sum_{d=1}^{D} \mathcal{H}(p(x_d)) + \mathcal{H}_{\text{copula}}$, where $\mathcal{H}(p(x_d))$ is the marginal entropy of each dimension, and $\mathcal{H}_{\text{copula}}$ represents the entropy of the copula. The copula is defined through the decomposition $p(x) = p_1(x_1) \ldots p_D(x_D) c\Big(F_1(x_1), \ldots, F_D(x_D)\Big)$, where $p_d(x_d)$ is the marginal of dimension $d$, $F_d(x_d)$ is its corresponding cumulative distribution function, and $c$ is the density of the copula function, which is defined as a probability density of over the hypersquare $[0, 1]^D$ with marginals that are uniform on $[0, 1]$. The marginal entropies are estimated using sample-spacing or binning entropy estimation techniques. The copula entropy is estimated *recursively* by sampling from the copula density, splitting the data into two subgroups, then computing the entropy for each using the aforementioned decomposition. Kandasamy et al. (2015) proposed a leave-one-out technique to improve the robustness of entropy estimation using the von Mises expansion-based estimator. The key idea is to iteratively remove one data point from the sample and compute the entropy estimate using the remaining data points using sample based entropy estimation methods. This procedure helps reduce bias and ensures that the estimator is not overly influenced by any single data point. The leave-one-out entropy is given by: $\mathcal{H}^{\text{LOO}}(p) = \frac{1}{M} \sum_{i=0}^{M-1} \mathcal{H}_{-i}$ where $\mathcal{H}_{-i}$, is

| Method | Formula | Key Idea |
|---|---|---|
| Analytical | $\mathcal{H}(p)$ | Closed-form expression (when available). |
| Plug-in | $-\dfrac{1}{M}\displaystyle\sum_{i=0}^{M-1}\log \hat{p}(x_i)$ | Estimate a density then plug into the definition. |
| KDE | $-\dfrac{1}{M}\displaystyle\sum_{i=0}^{M-1}\log\left(\dfrac{1}{M\,h^d}\sum_{j=0}^{M-1}\kappa\left(\dfrac{x_i-x_j}{h}\right)\right)$ | Smooth the empirical density with a kernel. |
| KNIFE | $-\dfrac{1}{M}\displaystyle\sum_{i=0}^{M-1}\log\left(\sum_{j=0}^{M-1}\mu_j\,\kappa_{\Sigma_j}(x_i-b_j)\right)$ | Kernel mixture with learned weights/bandwidths. |
| Nearest-Neighbor (KL) | $\psi(M)-\psi(k)+\log c_d+\dfrac{d}{M}\displaystyle\sum_{i=0}^{M-1}\log \epsilon_i$ | Use $k$NN radii $\epsilon_i$ to estimate local volumes. |
| Vasícek (1D) | $-\dfrac{1}{M}\displaystyle\sum_{i=0}^{M-m-1}\log\left(\dfrac{M}{2m}\big(x_{(i+m)}-x_{(i-m)}\big)\right)$ | Entropy from spacings of order statistics. |
| Variational (VI) | $-\mathbb{E}_{q(x)}[\log q(x)]$ | Fit a tractable surrogate $q$. |
| MINE | $\displaystyle\sup_{\theta}\left(\mathbb{E}_{p_{x,y}}[T_\theta]-\log\mathbb{E}_{p_x p_y}[e^{T_\theta}]\right)$ | Train a critic via the DV bound (for MI/entropy-related objectives). |
| CADEE | $\displaystyle\sum_{i=1}^{d}\mathcal{H}(y_i)+\mathcal{H}_{\text{copula}}$ | Decompose multivariate entropy via a copula. |
| LOO | $\dfrac{1}{M}\displaystyle\sum_{i=0}^{M-1}\mathcal{H}_{-i}$ | Leave-one-out aggregation. |

Table 3: Differential-entropy estimators and related objectives. $M$: samples; $d$: dimension; $\kappa$: kernel; $h$: bandwidth; $c_d$: volume of the $d$-dimensional unit ball; $\psi$: digamma; $\epsilon_i$: $k$NN radius; $x_{(i)}$: $i$th order statistic.

computed using $\{x_1,...,x_{i-1},x_{i+1},...,x_n\}$. This approach provides a more robust estimate of the entropy by mitigating the influence of outliers or anomalous data points.

A summary of these methods is provided in Tab. 3.

## B.2 SAMPLING-BASED VARIATIONAL INFERENCE.

Bridging the gap between parametric variational inference (VI) and Markov Chain Monte Carlo (MCMC) has been a key research focus to achieve both expressivity and scalability in inference. A central challenge is deriving an analytical expression for the marginal distribution of the last sample in an MCMC chain, which is often intractable. To address this, prior work (Salimans et al., 2015; Geffner & Domke, 2023) introduced auxiliary variables to construct augmented variational distributions that include all samples from the chain. However, this approach requires optimizing a looser ELBO and estimating the reverse Markov kernel, which introduces additional parameters and complex design choices. Several extensions have been proposed to avoid estimating the reverse kernel: $(i)$ Hoffman (2017) optimize ELBO with respect to the initial distribution and only uses the MCMC steps to produce "better" samples to the target distribution. However, this method lacks direct feedback between the final marginal distribution and variational parameters, limiting full unification of VI and MCMC, $(ii)$ Caterini et al. (2018) propose a deterministic Hamiltonian MCMC by removing resampling and the accept-reject step. However, this sacrifices MCMC guarantees, $(iii)$ Thin et al. (2020) introduce *MetFlow*, a Metropolis-Hastings method that models the proposal distribution as a normalizing flow, removing the need for inverse kernel estimation. *MET-SVGD* has several advantages compared with the aforementioned approaches: It computes the exact loglikelihood, *i.e.*, via using the change of variable formula (Sec. A.2). Hence, there is no need in the variational approximation on the joint distribution of the samples of the Markov chain, to estimate the reverse dynamics. Besides, it leverages knowledge of the unnormalized density unlike classical flow models. This makes our approach very easy to integrate in modern day deep learning pipelines. The idea of approximating log-likelihoods for distributions known up to a normalization constant using

MCMC and the change-of-variable formula was first explored by (Dai et al., 2019), applying it to Hamiltonian Monte Carlo (*HMC*) and Langevin Dynamics (*LD*). Since, they augment the input with noise or velocity variable for *LD* and *HMC*, respectively, the derived log-likelihood of the sampling distribution turns out to be –counter-intuitively– independent of the sampler's dynamics and equal to the initial distribution, which is then parameterized using a normalizing flow model (Kobyzev et al., 2020). Our derived log-likelihood is more intuitive as it depends on the *SVGD* dynamics.

### B.3 NORMALIZING FLOWS AND RESIDUAL FLOWS

We review Normalizing Flows in general and focus on residual flows as *MET-SVGD* is one.

**Normalizing Flows** are generative models that produce tractable distributions where both sampling and density evaluation can be efficient and exact. This is achieved by transforming a simple probability distribution (*e.g.*, a standard normal) into a more complex distribution by a sequence of invertible and differentiable mappings. The density of a sample can be evaluated by transforming it back to the original simple distribution and then computing the product of the density of the inverse-transformed sample under this distribution and the associated change in volume induced by the sequence of inverse transformations. The change in volume is the product of the absolute values of the determinants of the Jacobians for each transformation, as required by the change of variables formula (App.A.2). Formally, Let $x = (x_1, x_2, \cdots, x_d) \in \mathbb{R}^d$ be a random variable with a known and tractable probability density function $p_x : \mathbb{R}^d \to \mathbb{R}$. Let $f$ be a bijective, differentiable function and $x = f(z)$. Then, using the change of variables formula, one can compute the probability density function of the random variable $y$:

$$p_x(x) = p_z(f^{-1}(x)) \left| \det \nabla_x f^{-1}(x) \right| \tag{11}$$

Intuitively, if the transformation $F$ is arbitrarily complex, one can generate any distribution $p_x$ from any base distribution $p_z$ under reasonable assumptions on the two distributions. This has been formally proven in Bogachev et al. (2005). However, constructing arbitrarily complex non-linear invertible functions can be difficult. Additionally, $f$ should be sufficiently expressive to model the distribution of interest and computationally efficient, both in terms of computing $f$, its inverse, and the determinant of the Jacobian $\nabla_x f^{-1}(x)$.

This motivated the design of different types of flows: (1) Planar Flows (Rezende & Mohamed, 2015), (2) Radial Flows (Rezende & Mohamed, 2015), (3) Coupling Flows (Dinh et al., 2014), (4) Autoregressive Flows (Papamakarios et al., 2017), and (5) Residual Flows (Chen et al., 2019), which we focus on due to relevance to *MET-SVGD*.

**Residual Flows** are compositions of the functions of the form $f(x) = x + \phi(x)$. The first attempts to build a reversible network architecture based on residual connections was motivated by saving memory (each layer activation can be reconstructed from the previous layer) (Gomez et al., 2017; Jacobsen et al., 2018) and was achieved via partitioning units in each layer into two groups and defining coupling functions

$$y^A = x^A + \phi_1(x^B), \;\; y^B = x^B + \phi_2(y^A),$$

where $x = (x^A, x^B)$ and $y = (y^A, y^B)$ are respectively the input and output activations, and $\phi_1 : \mathbb{R}^{D-d} \to \mathbb{R}^d$ and $\phi_2 : \mathbb{R}^d \to \mathbb{R}^{D-d}$ are residual blocks. The Jacobian of such transformations is, however, inefficient to compute, and constraints on the architecture. Behrmann et al. (2019) proved that such functions are invertible if we constrain $\phi_1$ and $\phi_2$ to be Lipschitz continuous with Lipschitz constant satisfying $\text{Lip}(\phi) < 1$. We adapt this condition to *SVGD* (App. D.2). Controlling the Lipschitz constant of a neural network is not trivial. One approach is to regularize the spectral norm of the Jacobian of $\phi$ Sokolić et al. (2017). However, this only reduces it locally and does not guarantee the aforementioned condition. Instead, Jacobsen et al. (2018) proposes constraining the spectral radius of each convolutional layer in this network to be less than one, which incurs an additional overhead but yields compelling results. In the context of residual flows, the density was also derived using the change of variable formula (App. A.2). A different approach than ours was proposed to approximate the log-det term:

$$\log |\det(I + \nabla_x \phi(x)| \overset{(i)}{=} \text{Tr}(\log(I + \nabla_x \phi(x))) \overset{(ii)}{=} \sum_{k=1}^{\inf} (-1)^{k+1} \frac{\text{Tr}(\nabla_x \phi(x))^k}{k}$$

Where $(i)$ is obtained using $\log \det(A) = \mathrm{Tr}(\log(A))$ for non-singular $A \in \mathbb{R}^{d \times d}$ Withers & Nadarajah (2010), and $(ii)$ follows from replacing the trace of the matrix by its power series. By truncating this series, one can calculate an approximation to the log Jacobian determinant. To efficiently compute each member of the truncated series, the Hutchinson trick is used. However, this resulted in a biased estimate of the log Jacobian determinant. An unbiased stochastic estimator was proposed by (Chen et al., 2019). In a model they called a Residual flow, the authors used a Russian roulette estimator instead of truncation. Informally, while calculating the series, one flips a coin to decide if the calculation should be continued or stopped. Differently, we adopt a first-order approximation and derive a bound on the step-size for it to hold. This is both more efficient and accurate. Also, the specific form of $\phi$ enables deriving a bound on the step-size for invertibility to hold.

### B.4 STEIN VARIATIONAL GRADIENT DESCENT

In Sec. B.4.1, we provide a formal derivation of *SVGD*. Then, in Sec. B.4.2, we discuss how the choice of the RBF bandwidth shapes the interaction dynamics between particles. After that, in Sec. B.4.3, we summarize known convergence guarantees for both the infinite-particle and finite-particle regimes. Finally, in Sec. B.4.4, we review notable recent *SVGD* variants.

#### B.4.1 SVGD DERIVATION

(Liu & Wang, 2016) Let $p(x)$ be a smooth, strictly positive density supported on $\mathcal{X} \subseteq \mathbb{R}^d$. The goal is to approximate $p$ via a variational distribution $q \in \mathcal{Q}$ *i.e.*,

$$q^* = \arg\min_{q \in \mathcal{Q}} D_{KL}(q\|p).$$

$\mathcal{Q}$ is defined by the family of distributions obtained by transforming a reference density $q^0$ via an invertible map $f : \mathcal{X} \to \mathcal{X}$, where for any particle $x \sim q^0$, we define $y = f(x)$. The distributions of $y$ and $x$ are related via the CVF (App. A.2):

$$q(y) = q_0(f^{-1}(y)) \cdot |\det(\nabla_y f^{-1}(y))|$$

In this setup, $f(x)$ is chosen to have a specific form: $f(x) = x + \epsilon\phi(x)$, where $\epsilon$ is a step-size and $\phi$ is the infinitesimal perturbation direction that maximally decreases the KL divergence between $q$ and $p$:

$$\phi^* = \arg\max_{\phi \in \mathcal{F}} -\nabla_\epsilon D_{KL}(q\|p)\big|_{\epsilon=0}$$

This maximization has a closed form expression if we constrain the space of perturbations $\mathcal{F}$ to be the unit ball of the reproducing kernel Hilbert space (RKHS) $\mathcal{H}^D$ associated with a positive definite kernel $\kappa(\cdot, \cdot)$, *i.e.*, $\mathcal{F} = \{\phi \in \mathcal{H}^D : \|\phi\|_{\mathcal{H}^D} \leq 1\}$. In this case $\arg\max_{\phi \in \mathcal{F}} -\nabla_\epsilon D_{KL}(q\|p)\big|_{\epsilon=0} = \arg\max_{\phi \in \mathcal{F}} \mathbb{E}_q[\mathrm{Tr}(\mathcal{A}_p\phi)]$, where $\mathcal{A}_p$ is the stein operator:

$$\mathcal{A}_p\phi(x) = \phi(x)\nabla_x \log p(x)^\top + \nabla_x \phi(x)$$

The optimal perturbation direction $\phi^*$ is, hence, the one that maximizes the Stein Discrepancy (Liu et al., 2016):

$$\mathbb{S}(q, p) = \max_{\phi \in \mathcal{F}} \{(\mathbb{E}_q[\mathrm{Tr}(\mathcal{A}_p\phi)])^2 \quad s.t \quad \|\phi\|_{\mathcal{F}} \leq 1\}.$$

This optimization has a closed form solution given by $\phi_{p,q}^*/\|\phi_{p,q}^*\|_{\mathcal{H}^D}$, with

$$\phi_{p,q}^*(.) = \mathbb{E}_q\Big[\kappa(x, .)\nabla_x \log p + \nabla_x \kappa(x, .)\Big].$$

*Proof.*

$$D_{KL}(q\|p) \overset{(i)}{=} \int_\mathcal{X} q(y) \log \frac{q(y)}{p(y)} dy$$

$$\overset{(ii)}{=} \int_\mathcal{X} q(f(x)) \log \frac{q(f(x))}{p(f(x))} |\det \nabla_x f(x)| dx$$

$$\overset{(iii)}{=} \int_\mathcal{X} q_0(x)|\det \nabla_x f(x)|^{-1} \log \frac{q_0(x)}{p(f(x))|\det \nabla_x f(x)|} |\det \nabla_x f(x)| dx$$

$$\overset{(iv)}{=} \int_\mathcal{X} q_0(x) \log \frac{q_0(x)}{p(f(x))|\det \nabla_x f(x)|} dx$$

$(i)$ by definition of the KL divergence

$(ii)$ by applying the change of variable with $y = f(x)$

$(iii)$ by applying the change of variable formula to $q(F(x))$

$(iv)$ by simplifying inverse terms

Because $q_0$ does not depend on $\epsilon$, we have

$$\nabla_\epsilon D_{KL}\big|_{\epsilon=0} = -\mathbb{E}_{x\sim q_0}\left[\nabla_\epsilon \log p(f(x))\big|_{\epsilon=0} + \nabla_\epsilon \log|\det \nabla_x f(x)|\big|_{\epsilon=0}\right] \tag{12}$$

*First term.* By the chain rule and with $s_p = \nabla_x \log p$,

$$\nabla_\epsilon \log p(f(x))\big|_{\epsilon=0} = s_p(f(x))^\top \nabla_\epsilon f(x)\big|_{\epsilon=0}.$$

*Second term.* The identity $\nabla_\epsilon \log|\det A| = \mathrm{Tr}(A^{-1}\nabla_\epsilon A)$ gives

$$\nabla_\epsilon \log|\det(\nabla_x f(x))|\big|_{\epsilon=0} = \mathrm{Tr}\left((\nabla_x f(x))^{-1}\nabla_\epsilon \nabla_x f(x)\big|_{\epsilon=0}\right).$$

We evaluate at $\epsilon = 0$ and obtain:

$$f(x)\big|_{\epsilon=0} = x, \qquad \nabla_\epsilon f(x)\big|_{\epsilon=0} = \phi(x),$$

$$\nabla_x f(x)\big|_{\epsilon=0} = I, \qquad \nabla_\epsilon \nabla_x f(x)\big|_{\epsilon=0} = \nabla_x \phi(x).$$

Substituting these expressions into Eq. 12 gives:

$$-\nabla_\epsilon D_{\mathrm{KL}}(q\,\|\,p)\Big|_{\epsilon=0} = \mathbb{E}_{x\sim q}\left[s_p(x)^\top \phi(x) + \mathrm{Tr}(\nabla_x \phi(x))\right] = \mathbb{E}_{x\sim q}\left[\mathrm{Tr}\left(\mathcal{A}_p \phi(x)\right)\right].$$

$\square$

### B.4.2   RBF KERNEL VARIANCE INTERPRETATION.

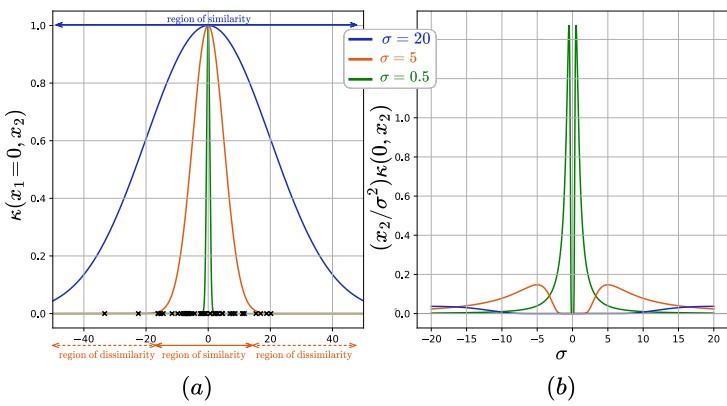

$(a)$ $\qquad\qquad\qquad\qquad$ $(b)$

Figure 12: (a) Visualization of the neighborhood of particle $x_1 = 0$ as measured by the RBF kernel $\kappa(0, x_2)$ for different kernel bandwidth values $\sigma$. (b) Repulsion force at $x_1 = 0$.

For $\mathcal{X} \subseteq \mathbb{R}$, the RBF kernel is a function $\kappa : \mathcal{X} \times \mathcal{X} \to \mathbb{R}$ defined as $\kappa(x_1, x_2) = \exp(-\frac{\|x_1-x_2\|^2}{2\sigma^2})$, where $\sigma$ is referred to as kernel bandwidth. In the context of the *SVGD* update rule,

$$x^{l+1} = x^l + \epsilon\mathbb{E}_{x_j^l}\Big[\underbrace{\kappa(x^l, x_j^l)\nabla_{x_j^l}\log p(x_j^l)}_{\text{drift term}} + \underbrace{\frac{(x^l - x_j^l)}{\sigma^2}\kappa(x^l, x_j^l)}_{\text{repulsion term}}\Big],$$

the RBF kernel in the drift term specifies how much $x^l$ should move along $\nabla_{x_j^l}\log p(x_j^l)$, and in the repulsion term modulates how far $x^l$ should be pushed away from $x_j^l$ along the direction $x^l - x_j^l$.

Intuitively, $\kappa(x^l, x_j^l)$ measures how similar $x^l$ and $x_j^l$ are based on the euclidean distance. The maximum value that the RBF kernel can take is 1 for $x^l = x_j^l$. When a large distance separates $x^l$ and $x_j^l$, the kernel value is close to 0, and the particles do not affect each other. The width of the region of similarity is controlled by $\sigma$, which we illustrate in Fig. 12.a.

Setting $\sigma$ is not obvious. For the repulsion term, both a very small or a very large $\sigma$ value can result in setting the repulsion term to 0 as shown in Fig. 12.b. Classically, the median trick is used to set $\sigma$, *i.e.*, $\sigma_{\text{med}} = \text{median}\{\|x_i^l - x_j^l\|\}_{i,j=0}^{M-1}/\log M$ with $M$ being the number of particles. In our experiments, we show that this is suboptimal and that that a more optimal $\sigma$ can be learnt end-to-end via minimizing the reverse KL-divergence (Sec. 3.2).

### B.4.3  SVGD CONVERGENCE RATE

*SVGD* is difficult to analyze theoretically because it involves a system of particles that interact with each other in a complex way. In the infinite particles case, Liu (2017) proves that *SVGD* converges (weakly) to $p$ in kernelized Stein discrepancy (App. A.8). Korba et al. (2020); Salim et al. (2022); Sun et al. (2023) refined these results with path-independent constants, weaker smoothness conditions, and explicit rates of convergence. Duncan et al. (2023) provides conditions for exponential convergence. For the finite particles case, Liu (2017) shows that finite particles *SVGD* converges to infinite particles *SVGD* in bounded-Lipschitz distance but only under boundedness assumptions violated by most applications of *SVGD*. Korba et al. (2020) explicitly bounded the expected squared Wasserstein distance between $M$-particle and continuous *SVGD* but only under the assumption of bounded $\log p$. Also they do not provide convergence rates. Liu et al. (2024) show that *SVGD* with finite particles achieves linear convergence in KL divergence under a very limited setting where the target distribution is Gaussian. Shi & Mackey (2024) shows that *SVGD* convergence rate is $\mathcal{O}(1/\sqrt{\log \log M})$ under the assumption that the target is sub-Gaussian with a Lipschitz score, with $M$ being the number of particles.

### B.4.4  SVGD VARIANTS

There have been many *SVGD* variants that were proposed to tackle different limitations. Stein Variational Newton Method (Detommaso et al., 2018) accelerates *SVGD* by incorporating second-order information to improve convergence and kernel effectiveness. Matrix-valued-kernel SVGD (Wang et al., 2019) incorporates geometric preconditioning (via Hessians, Fisher information, or other matrices) into the kernel, accelerating exploration in the probability landscape. RBM-SVGD (Li et al., 2019) is a stochastic variant of *SVGD* that applies the Random Batch Method to reduce computational cost, particularly in the case of long-range kernels, while retaining *SVGD*'s efficiency for sampling from probability distributions. Stochastic SVGD (Gorham et al., 2020) introduces stochastic mini-batch estimators of the Stein operator so that *SVGD* can operate efficiently when the score function is expensive to compute, while still converging almost surely to standard *SVGD*. Mirrored SVGD (Shi et al., 2022) extends *SVGD* to constrained domains and non-Euclidean geometries by performing updates in a dual space using mirrored Stein operators and adaptive kernels. Sliced SVGD (Gong et al., 2021) addresses *SVGD*'s high-dimensional variance underestimation problem (Ba et al., 2022) by projecting updates onto a sequence of fixed one dimensional slices. Grassmann SVGD (Liu et al., 2022b) tackles the same variance-underestimation problem by learning adaptive multi-dimensional subspace projections for both the data and the score. $\beta$-SVGD (Sun & Richtárik, 2023) proposes a weighted version of *SVGD* that uses importance weights to accelerate convergence, particularly when the initial distribution is far from the target.

Our motivation is fundamentally different. We derived a closed-form expression of the *SVGD* density to leverage it for entropy estimation of targets known up to a normalization constant.

### B.5  MARKOV CHAINS

Suppose we want to sample from a complex probability distribution $p$. The Markov chain is completely defined by the initial state $\Pi^0$ and the transition matrix $Q = [q(x_i|x_j)]$, where $q(x_i|x_j)$ is the probability of transitioning from state $i$ to $j$.

Two key assumptions are make for Markov Chains convergence to a unique stationary distribution regardless of the initial state are:

- **Irreducibility**: This means that it's possible to go from any state to any other state for a large enough steps $l$

- **Aperiodicity**: States can be returned to at irregular time intervals. Formally, there exists $l$ such that for all $l' \geq l$, $\mathbb{P}(x_{l'} = i \mid x_0 = i) > 0$

A distribution $\Pi^*$ is a **stationary probability distribution** if $\Pi^* = \Pi^* Q$. In general, stationary probability distributions are not unique. The **irreducibility** condition guarantees that the stationary distribution is unique. $\Pi^*$ is the limiting distribution if for every initial probability distribution $\Pi^0$, $\lim_{n \to \infty} \Pi^n = \Pi^*$. The **aperiodicity** condition is necessary for the limit to exist.

To make the **stationary distribution** equal to the target distribution $p$ that we want to sample form, we additionally assume that the Markov chain is **reversible**.

A Markov chain is **reversible** if there exists a distribution $\Pi^*$ which satisfies the detailed balance conditions:

$$\forall i, j, \quad \Pi_i^* Q_{ij} = \Pi_j^* Q_{ji}.$$

Since we want the stationary distribution of the Markov chain to be $p(x)$, it suffices to design the transition matrix $P$ so the Markov chain satisfies detailed balance with respect to $p(x)$.

### B.6    METROPOLIS–HASTINGS

The Metropolis–Hastings algorithm's goal is to generate a Markov Chain $\{x^{(l)}\}_{l=0}^{\infty}$ that simulates samples from a given probability distribution $p$. The chain starts with samples from an initial distribution $q^{(0)}$ and updates its state by leveraging a proposal distribution $q(\tilde{x}|x^{(l)})$ as

$$x^{(l+1)}|x^{(l)} = \begin{cases} \tilde{x}, & \text{if } u^l \leq \alpha^l = \min\left(1, \frac{p(\tilde{x})q(x^{(l)}|\tilde{x})}{p(x^{(l)})q(\tilde{x}|x^{(l)})}\right) \\ x^{(l)}, & \text{otherwise} \end{cases}$$

where $u^{(l)} \sim \mathcal{U}(0,1)$ and $\alpha^l$ is called the acceptance probability. This guarantees that $p$ is the stationary distribution of the chain (Tierney, 1994). As an example, the Metropolis-Adjusted Langevin Algorithm employs the following proposal distribution

$$q(\tilde{x}^{(l+1)}|x^{(l)}) = \mathcal{N}_d\left(x^{(l)} + \epsilon \nabla \log p(x^{(l)}), 2\epsilon I_d\right).$$

If the proposal kernel makes the chain irreducible and aperiodic, then the chain is ergodic and converges to $p$ in total variation (strong convergence).

$$\lim_{l \to \infty} \left\| q^l(x, \cdot) - \pi(\cdot) \right\|_{\text{TV}} = 0,$$

—-

# C    SVGD DENSITY DERIVATION

**Notation:** In this section, for conciseness, we introduce three new quantities:

$$\gamma = \frac{1}{2\sigma^2}, \quad \delta_{i,j} = x_i^l - x_j^l, \quad \text{and} \quad s_p(x_j^l) = \nabla_{x_j^l} \log p(x_j^l).$$

We express $\kappa(x_i^l, x_j^l)$, $\nabla_{x_i^l} \kappa(x_i^l, x_j^l)$, $\nabla_{x_j^l} \kappa(x_i^l, x_j^l)$, and $\nabla_{x_i^l} \nabla_{x_j^l} \kappa(x_i^l, x_j^l)$ as:

- $\kappa(x_i^l, x_j^l) = \exp(-\gamma \|x_i^l - x_j^l\|^2)$
- $\nabla_{x_i^l} \kappa(x_i^l, x_j^l) = 2\gamma \delta_{i,j} \kappa(x_i^l, x_j^l)$
- $\nabla_{x_j^l} \kappa(x_i^l, x_j^l) = -2\gamma \delta_{i,j} \kappa(x_i^l, x_j^l) = -\nabla_{x_j^l} \kappa(x_i^l, x_j^l)$
- $\nabla_{x_i^l} \nabla_{x_j^l} \kappa(x_i^l, x_j^l) = \nabla_{x_i^l} \left( 2\gamma \delta_{i,j} \kappa(x_i^l, x_j^l) \right) = 2\gamma \left( I - 2\gamma \delta_{i,j} \delta_{i,j}^T \right) \kappa(x_i^l, x_j^l)$

**Theorem.** *Let $f : \mathbb{R}^d \rightarrow \mathbb{R}^d$ be a transformation of the form $f(x) = x + \epsilon\phi(x)$. We denote by $q^L(x^L)$ the distribution obtained from repeatedly ($L$ times) applying $f$ to a set of particles $\{x^0\}_{i=0}^{M-1}$ from an initial distribution $q^0$, i.e., $x^L = \underbrace{f \circ \cdots \circ f}_{L \text{ times}}(x^0)$. Under the condition $\epsilon < \epsilon_{UB}^l = 1/\sup_x \sqrt{\mathrm{Tr}(\nabla\phi(x^l)\nabla\phi^T(x^l))}, \forall l \in [0, L-1]$, the closed-form expression of $\log q^L(x^L)$ is:*

$$\log q^L(x^L) = \log q^0(x^0) - \epsilon \sum_{l=0}^{L-1} \mathrm{Tr}(\nabla_{x^l}\phi(x^l)) + \mathcal{O}\left( \left(\epsilon/\epsilon_{UB}^l\right)^2 \right) \tag{13}$$

*Proof.* In Sec. D.4, we show that under the constraint $\epsilon < \epsilon^l$, $f$ is invertible. Thus, we use the change of variable formula (A.2) to derive the distribution of $x^{l+1} = x^l + \epsilon\phi(x^l)$:

$$q^{l+1}(x^{l+1}) = q^l(x^l) \left| \det \nabla_{x^l}\phi(x^l) \right|^{-1}, \forall l \in [0, L-1].$$

By induction, we derive the probability distribution of sample $x^L$:

$$q^L(x^L) = q^0(x^0) \prod_{l=0}^{L-1} \left| \det \left( I + \epsilon\nabla_{x^l}\phi(x^l) \right) \right|^{-1}$$

By taking the $\log$ for both sides, we obtain:

$$\log q^L(x^L) = \log q^0(x^0) - \sum_{l=0}^{L-1} \log \left| \det \left( I + \epsilon\nabla_{x^l}\phi(x^l) \right) \right|.$$

This, however, requires computing the Jacobian $\nabla_{x^l}\phi(x^l)$, which is quadratic in the dimensionality $d$. In Sec. C.1, we show that $\log \left| \det \left( I + \epsilon\nabla_{x^l}\phi(x^l) \right) \right| = \epsilon \mathrm{Tr}(\nabla_{x^l}\phi(x^l)) + \mathcal{O}\left( \left(\epsilon/\epsilon_{UB}^l\right)^2 \right)$ under the constraint on the step-size stated in the theorem. $\square$

We derive the expression of $\mathrm{Tr}\left( \nabla_{x^l}\phi(x^l) \right)$ for the RBF, Bilinear, and DKEF kernels in Sec. C.2, Sec. C.3, and Sec. C.4, respectively.

## C.1    SUFFICIENT CONDITION FOR
$$\log \left| \det \left( I + \epsilon\nabla_{x^l}\phi(x^l) \right) \right| = \epsilon \mathrm{Tr}(\nabla_{x^l}\phi(x^l)) + \mathcal{O}\left( \left(\epsilon/\epsilon_{UB}^l\right)^2 \right)$$

Let $\phi : \mathbb{R}^d \rightarrow \mathbb{R}^d$ be a differentiable vector-valued map. $\log |\det(I + \epsilon\nabla_{x^l}\phi(x^l))| = \epsilon \mathrm{Tr}(\nabla_{x^l}\phi(x^l)) + \mathcal{O}\left( \left(\epsilon/\epsilon_{UB}^l\right)^2 \right)$ if $\epsilon < \epsilon_{UB}^l = 1/\sup_x \sqrt{\mathrm{Tr}(\nabla_{x^l}\phi(x^l)\nabla_{x^l}\phi^T(x^l))}$, with $\nabla$ the gradient operator w.r.t the input.

*Proof.* In the following we denote by $\lambda_i(A)$ the eigenvalue of matrix $A$

$$\left| \det \left( I + \epsilon \nabla_{x^l} \phi(x^l) \right) \right| \stackrel{(i)}{=} \left| \prod_{i=1}^d \lambda_i (I + \epsilon \nabla_{x^l} \phi(x^l)) \right| = \prod_{i=1}^d \left| \lambda_i (I + \epsilon \nabla_{x^l} \phi(x^l)) \right|$$

$$\stackrel{(ii)}{=} \prod_{j=1}^d \left| 1 + \epsilon \lambda_j (\nabla_{x^l} \phi(x^l)) \right| = \exp \left( \sum_{j=1}^d \ln \left| 1 + \epsilon \lambda_j (\nabla_{x^l} \phi(x^l)) \right| \right)$$

$$= \exp \left( \sum_{j=1}^d \ln \left( 1 + \epsilon \lambda_j (\nabla_{x^l} \phi(x^l)) \right) \right) \quad \text{if } \lambda_j (\nabla_{x^l} \phi(x^l)) > \frac{-1}{\epsilon}$$

$$\stackrel{(iii)}{=} \exp \left( \sum_{j=1}^d \epsilon \lambda_j (\nabla_{x^l} \phi(x^l)) + \mathcal{O}((\epsilon \lambda_j^l)^2) \right) \quad \text{if } |\epsilon \lambda_j (\nabla_{x^l} \phi(x^l))| < 1$$

$(i)$ By definition of the determinant.

$(ii)$ Let $\lambda_i$ be the eigenvalue of $(I + \epsilon \nabla_{x^l} \phi(x^l))$ associated with the eigenvector $v_i$. We show that $\lambda_i - 1$ is the eigenvalue associated with $\epsilon \nabla_{x^l} \phi(x^l)$:

$$(I + \epsilon \nabla_{x^l} \phi(x^l)) v_i = \lambda_i v_i \implies \epsilon \nabla_{x^l} \phi(x^l) v_i = (\lambda_i - 1) v_i \implies \lambda_j = (\lambda_i - 1) \text{ is an eigenvalue of } \epsilon \nabla_{x^l} \phi(x^l)$$

$(iii)$ We use Taylor expansion of $\ln(1 + \epsilon a) = \sum_i \frac{(-1)^{i-1} (\epsilon a)^i}{i} = \epsilon a + \mathcal{O}((\epsilon a)^2)$ around $\epsilon a \to 0$.

Hence, if $\epsilon |\lambda_i (\nabla_{x^l} \phi(x^l))| \leq \epsilon |\lambda_{\max} (\nabla_{x^l} \phi(x^l))| \leq \epsilon \sup_x \sqrt{\text{Tr}(\nabla \phi(x^l) \nabla \phi^T(x^l)} < 1$, then $\log \left| \det \left( I + \epsilon \nabla_{x^l} \phi(x^l) \right) \right| = \epsilon \text{Tr}(\nabla_{x^l} \phi(x^l)) + \mathcal{O}\left( \epsilon^2 \lambda_{\max}^2 (\nabla_{x^l} \phi(x^l)) \right)$.

Moreover,

$$\mathcal{O}\left( \left( \epsilon |\lambda_{\max} (\nabla_{x^l} \phi(x^l))| \right)^2 \right) \stackrel{(i)}{=} \mathcal{O}\left( \left( \frac{\epsilon_{\text{UB}}^l}{\epsilon_{\text{UB}}^l} \epsilon |\lambda_{\max} (\nabla_{x^l} \phi(x^l))| \right)^2 \right)$$

$$\stackrel{(ii)}{=} \mathcal{O}\left( \left( \frac{\epsilon}{\epsilon_{\text{UB}}^l} \frac{|\lambda_{\max} (\nabla_{x^l} \phi(x^l))|}{\sup_x \sqrt{\text{Tr}\left( \nabla_{x^l} \phi(x^l)^\top \nabla_{x^l} \phi(x_i^l) \right)}} \right)^2 \right)$$

$$\stackrel{(iii)}{=} \mathcal{O}\left( \left( \frac{\epsilon}{\epsilon_{\text{UB}}^l} \right)^2 \right)$$

$(i)$ Multiply by $1 = \epsilon_{\text{UB}}^l / \epsilon_{\text{UB}}^l$

$(ii)$ Substitute in the expression of $\epsilon_{\text{UB}}^l$

$(iii)$ Using the fact that $|\lambda_{\max} \left( \nabla_{x^l} \phi(x^l) \right)| / \sup_{x^l} \sqrt{\text{Tr}\left( \nabla_{x^l} \phi(x^l)^\top \nabla_{x^l} \phi(x^l) \right)} \leq 1$

Thus, $\log |\det(I + \epsilon \nabla_{x^l} \phi(x^l))| = \epsilon \text{Tr}(\nabla_{x^l} \phi(x^l)) + \mathcal{O}\left( (\epsilon / \epsilon_{\text{UB}}^l)^2 \right)$. $\qquad \square$

## C.2 COMPUTING $\text{Tr}(\nabla_{x^l} \phi(x^l))$ WITH RBF KERNEL

We show that the closed-form estimate of the log-likelihood $\log q^L(x^L)$ for the *SVGD*-based sampler with an RBF kernel $\kappa(\cdot, \cdot)$ is

$$\log q^L(x^L) = \log q^0(x^0) - \frac{\epsilon}{M} \sum_{l=0}^{L-1} \left[ \sum_{\substack{j=0 \\ x^l \neq x_j^l}}^{M-1} \left( \frac{1}{\sigma^2} \kappa(x_j^l, x^l) \left( -(x^l - x_j^l)^\top \nabla_{x_j^l} s_p(x_j^l) - \frac{1}{\sigma^2} \|x^l - x_j^l\|^2 + d \right) \right) \right.$$

$$+ \mathrm{Tr}\left(\nabla^2_{x^l_i} \log p(x^l)\right) + \mathcal{O}\left(\left(\epsilon/\epsilon^l_{\mathrm{UB}}\right)^2\right)\Bigg],$$

where the error term $\mathcal{O}\left(\left(\epsilon/\epsilon^l_{\mathrm{UB}}\right)^2\right)$ is explained in Sec. C.1.

*Proof.* We evaluate all terms under the RBF kernel:

$$\log q^L(x^L) = \log q^0(x^0) - \frac{\epsilon}{M} \sum_{l=0}^{L-1} \Bigg[ \sum_{\substack{j=0 \\ x^l \neq x^l_j}}^{M-1} \Big[ \underbrace{\mathrm{Tr}\left(\nabla_{x^l}(\kappa(x^l, x^l_j)\nabla_{x^l_j} \log p(x^l_j))\right)}_{①} + \underbrace{\mathrm{Tr}\left(\nabla_{x^l}\nabla_{x^l_j} \kappa(x^l, x^l_j)\right)}_{②} \Big]$$

$$+ \underbrace{\mathrm{Tr}\left(\nabla^2_{x^l} \log p(x^l)\right)}_{③} + \mathcal{O}\left(\left(\epsilon/\epsilon^l_{\mathrm{UB}}\right)^2\right) \Bigg] \tag{14}$$

In the following, we denote by $()^{(k)}$ the $k$-th dimension of the vector.

**Term ①:**

$$
\begin{aligned}
\mathrm{Tr}\left(\nabla_{x^l}(\kappa(x^l, x^l_j)\nabla_{x^l_j} s_p(x^l_j)^T)\right) &= \mathrm{Tr}\left(\nabla_{x^l}\kappa(x^l, x^l_j)(\nabla_{x^l_j} s_p(x^l_j))^T + \kappa(x^l, x^l_j)\nabla_{x^l}\nabla_{x^l_j} s_p(x^l_j)\right) \\
&= \sum_{t=1}^d \frac{\partial \kappa(x^l, x^l_j)}{\partial (x^l)^{(t)}} \frac{\partial s_p(x^l_j)}{\partial (x^l_j)^{(t)}} + 0 \\
&= (\nabla_{x^l}\kappa(x^l, x^l_j))^T \nabla_{x^l_j} s_p(x^l_j) \\
&= -\frac{1}{\sigma^2}\kappa(x^l, x^l_j)(x^l - x^l_j)^\top \nabla_{x^l_j} s_p(x^l_j)
\end{aligned}
$$

**Term ②:**

$$
\begin{aligned}
\mathrm{Tr}\left(\nabla_{x^l}\nabla_{x^l_j}\kappa(x^l, x^l_j)\right) &= \mathrm{Tr}\left(\nabla_{x^l}\left(\frac{1}{\sigma^2}\kappa(x^l, x^l_j)(x^l - x^l_j)\right)\right) \\
&= \frac{1}{\sigma^2}\sum_{k=1}^d \left(\frac{\partial \kappa(x^l, x^l_j)}{\partial (x^l)^{(k)}}(x^l - x^l_j)^{(k)} + \kappa(x^l, x^l_j)\right) \\
&= \frac{1}{\sigma^2}\left(\nabla_{x^l}\kappa(x^l, x^l_j)^\top(x^l - x^l_j) + d \times \kappa(x^l, x^l_j)\right) \\
&= \frac{1}{\sigma^2}\left(\nabla_{x^l}\kappa(x^l, x^l_j)^\top(x^l - x^l_j) + d \times \kappa(x^l, x^l_j)\right) \\
&= -\frac{1}{\sigma^4} \times \kappa(x^l, x^l_j)\|x^l - x^l_j\|^2 + \frac{1}{\sigma^2} \times d \times \kappa(x^l, x^l_j) \\
&= \kappa(x^l, x^l_j)\left(-\frac{1}{\sigma^4}\|x^l - x^l_j\|^2 + \frac{d}{\sigma^2}\right)
\end{aligned}
$$

**Term ③: Using Hutchinson's Trace Estimator Hutchinson (1989)**

$$\mathrm{Tr}\left(\nabla^2_{x^l} \log p(x^l)\right) = \mathbb{E}_{v \sim p_v}\left[v^T \nabla^2_{x^l} \log p(x^l) v\right] \approx \frac{1}{V}\sum_k v_k^T \nabla^2_{x^l} \log p(x^l) v_k$$

where $V$ is the number of vectors used to estimate the expectation, and $p_v$ is a distribution with zero mean and identity covariance, *i.e.*, $\mathbb{E}_{v \sim p_v}[v] = 0$, $\mathbb{E}_{v \sim p_v}[vv^T] = I$. Common choices include Rademacher or standard Gaussian vectors, in which case Hutchinson's estimator is unbiased, and its variance is of the order of $\mathcal{O}(1/V)$, where $V$ is the number of Hutchinson probe vectors. In practice, prior work (e.g., Song et al., 2020) typically uses a single Hutchinson vector per sample $x$. This works well because each element of the minibatch receives an independent draw of $v$, so minibatch

averaging implicitly acts as multiple Hutchinson probes of similar samples and significantly reduces variance. This is a standard and stable setting in deep-learning–based trace estimation. We follow the same procedure.

Note that the Hutchinson estimator in the *MET-SVGD* density is scaled by the number of particles $M$. So, the step-wise Hutchinson estimator variance contribution to the *SVGD* induced distribution is further scaled by $M^2$, *i.e.*, it's of the order of $\mathcal{O}(1/(VM^2))$. This is unlike *LD* where the step-wise variance is $\mathcal{O}(1/V)$ as we show in App.C.5.

By combining **Terms** ①, ② and ③, we obtain:

$$\log q^L(x^L) = \log q^0(x^0) - \frac{\epsilon}{M} \sum_{l=0}^{L-1} \left[ \sum_{j=0}^{M-1} \frac{1}{\sigma^2} \kappa(x_j^l, x_j^l) \left( -(x^l - x_j^l)^\top \nabla_{x_j^l} s_p(x_j^l) - \frac{1}{\sigma^2} \|x^l - x_j^l\|^2 + d \right) \right.$$

$$\left. + \mathbb{E}_{v \sim p_v} \left[ v^T \nabla_{x^l}^2 \log p(x^l) v \right] + \mathcal{O}\left( \left( \epsilon / \epsilon_{\mathrm{UB}}^l \right)^2 \right) \right]$$

$\square$

## C.3 COMPUTING $\mathrm{Tr}(\nabla_{x^l} \phi(x^l))$ WITH BILINEAR KERNEL

Liu et al. (2024) show that, for a Gaussian initial distribution, $q^0(x) = \mathcal{N}(\mu_0, \Sigma_0)$, and a target distribution $p(x) = \mathcal{N}(b, Q)$ with $Q \in \mathbb{R}^{d \times d}$, applying *SVGD* with a Bilinear kernel $\kappa(x_i, x_j) = \frac{x_j^T x_i}{C} + 1$, where $C \in \mathbb{R}^+$, produces a Gaussian density $q^l$ at every step with mean $\mu^l$ and covariance matrix $\Sigma^l$ given by:

- $\mu^{l+1} = \mu^l + \epsilon^l \left[ \left( I - (\Sigma^l + \mu^l \mu^{l^T}) Q^{-1} + \mu^l b^T Q^{-1} \right) \frac{\mu^l}{C} + (b - \mu^l)^T Q^{-1} \right]$

- $\Sigma^{l+1} = \Sigma^l + \epsilon^l \left[ 2\frac{\Sigma^l}{C} - \frac{\Sigma^l}{C} Q^{-1} (\Sigma^l + (\mu^l - b)\mu^{l^T}) - (\Sigma^l + \mu^l(\mu^l - b)^T) Q^{-1} \frac{\Sigma^l}{C} \right]$

*Proof.* For $\kappa(x_i^l, x_j^l) = \frac{x_j^{l^T} x_i^l}{C} + 1$, we compute $\nabla_{x_j^l} \kappa(x_i^l, x_j^l) = \frac{x_i^l}{C}$ and $\nabla_{x_i^l} \nabla_{x_j^l} \kappa(x_i^l, x_j^l) = \frac{I}{C}$. There resulting *SVGD* update rule is:

$$(x_i^{l+1} - x_i^l)/\epsilon^l = \frac{1}{M} \sum_{j=0}^{M-1} \nabla_{x_j} \kappa(x_i^l, x_j^l) + \frac{1}{M} \sum_{j=0}^{M-1} \kappa(x_i^l, x_j^l) \nabla_{x^l} \log p(x_j^l)$$

$$= \frac{1}{M} \sum_{j=0}^{M-1} \frac{x_i^l}{C} + \frac{1}{M} \sum_{j=0}^{M-1} \left( \frac{x_j^{l^T} x_i^l}{C} + 1 \right) \nabla_{x_j^l} \log p(x_j^l)$$

$$= \frac{x_i^l}{C} - \frac{1}{M} \sum_{j=0}^{M-1} Q^{-1}(x_j^l - b) \left( \frac{x_j^{l^T} x_i^l}{C} + 1 \right)$$

$$= \frac{x_i^l}{C} - \frac{1}{M} \sum_{j=0}^{M-1} Q^{-1}(x_j^l - b) - \frac{1}{M} \sum_{j=0}^{M-1} Q^{-1}(x_j^l - b) \frac{x_j^{l^T}}{C} x_i^l$$

$$= \frac{x_i^l}{C} - \frac{Q^{-1}}{M} \sum_{j=0}^{M-1}(x_j^l - b) - \frac{Q^{-1}}{M} \sum_{j=0}^{M-1} x_j^l \frac{x_j^{l^T}}{C} x_i^l + \frac{Q^{-1}b}{M} \sum_{j=0}^{M-1} \frac{x_j^{l^T}}{C} x_i^l$$

$$= \frac{x_i^l}{C} - Q^{-1}(\mu^l - b) - \frac{Q^{-1}}{C} \left( \Sigma^l + \mu^l \mu^{l^T} \right) x_i^l + \frac{Q^{-1}}{C} b \mu^{l^T} x_i^l$$

$$= \left( \frac{I}{C} - \frac{Q^{-1}}{C} \left( \Sigma^l + \mu^l \mu^{l^T} \right) + \frac{Q^{-1}}{C} b \mu^{l^T} \right) x_i^l - Q^{-1}(\mu^l - b)$$

thus, when $\epsilon \to 0$, we obtain:

$$\frac{\partial x_i^l}{\partial l} = \left( \frac{I}{C} - \frac{Q^{-1}}{C} \left( \Sigma^l + \mu^l \mu^{l^T} \right) + \frac{Q^{-1}}{C} b \mu^{l^T} \right) x_i^l - Q^{-1}(\mu^l - b). \tag{15}$$

Taking into account $\mu^l = \frac{1}{M} \sum_{j=0}^{M-1} x_j^l$ and $\Sigma^l = \frac{1}{M} \sum_{j=0}^{M-1} x_j^l x_j^{lT} - \mu^l \mu^{lT}$ and summing up the above expression with respect to the number of particles, we get

$$\frac{1}{M} \sum_{i=0}^{M-1} (x_i^{l+1} - x_i^l)/\epsilon^l = \left( \frac{I}{C} - \frac{Q^{-1}}{C} \left( \Sigma^l + \mu^l \mu^{lT} \right) + \frac{Q^{-1}}{C} b \mu^{lT} \right) \mu^l - Q^{-1}(\mu^l - b)$$

$$= \left( \frac{I}{C} - \frac{Q^{-1}}{C} \Sigma^l - \frac{Q^{-1}}{C} \left( \mu^l - b \right) \mu^{lT} \right) \mu^l - Q^{-1}(\mu^l - b)$$

$$= \left( I - Q^{-1} \Sigma^l \right) \frac{\mu^l}{C} - \frac{Q^{-1}}{C} \left( \mu^l - b \right) \mu^{lT} \mu^l - Q^{-1}(\mu^l - b)$$

$$= \left( I - Q^{-1} \Sigma^l \right) \frac{\mu^l}{C} - Q^{-1} \left( \mu^l - b \right) \left( \frac{\mu^{lT} \mu^l}{C} + 1 \right).$$

Hence,

$$(\mu^{l+1} - \mu^l)/\epsilon^l = \left( I - Q^{-1} \Sigma^l \right) \frac{\mu^l}{C} - Q^{-1} \left( \mu^l - b \right) \left( \frac{\mu^{lT} \mu^l}{C} + 1 \right) = \frac{\partial \mu^l}{\partial l}. \tag{16}$$

Finally, given that

$$\frac{\partial \Sigma^l}{\partial l} = \frac{\partial}{\partial l} \left( \frac{1}{M} \sum_{j=0}^{M-1} x_j^l x_j^{lT} - \mu^l \mu^{lT} \right)$$

and leveraging Eq. 15 and Eq. 16, we obtain

$$\left( \Sigma^{l+1} - \Sigma^l \right)/\epsilon^l = 2\frac{\Sigma^l}{C} - \frac{\Sigma^l}{C} Q^{-1} \left( \Sigma^l + (\mu^l - b)\mu^l \right) - \left( \Sigma^l + \mu^l (\mu^l - b)^T \right) Q^{-1} \frac{\Sigma^l}{C}$$

$\square$

We use this property to verify that the intermediate distributions derived by our formula with the bilinear kernel are accurate. We experiment with models for $C$:

- $C_{ij}^l = ||x_i^l|| \, ||x_j^l||$
- $C_{ij}^l = \max_{i,j} ||x_i^l|| \, ||x_j^l||$
- $C_{\theta_2}(x_i^l, x_j^l) = ||x_i^l|| \, ||x_j^l|| + |\theta_2|$, where $\theta_2 \in \mathbb{R}$ is a learnable parameter

Fig. 13 shows that the configurations in which the kernel bandwidth $C$ is not learnable (blue and green) fail to recover the ground-truth entropy, whereas learning $C$ enables the recovery of the entropy of the derived Gaussians at every step $l$.

### C.4 COMPUTING $\mathrm{Tr}(\nabla_{x^l} \phi(x^l))$ WITH DKEF KERNEL

The DKEF kernel (Wenliang et al., 2018) is a flexible kernel constructed by augmenting the RBF kernel with an embedding function $\psi : \mathbb{R}^d \to \mathbb{R}^m$, i.e., $\kappa(x_i^l, x_j^l) = \exp\left(-\|\psi(x_i^l) - \psi(x_j^l)\|^2\right)$. However, despite the flexibility that $\psi$ offers, the kernel is inefficient because it requires computing $\nabla_x \psi(x)$ in the entropy derivation.

*Proof.* In the following, we compute the terms ①, ②, and from Eq. 14 using $\kappa(x_i^l, x_j^l) = \exp\left(-\|\psi(x_i^l) - \psi(x_j^l)\|^2\right)$. Term ③ is computed

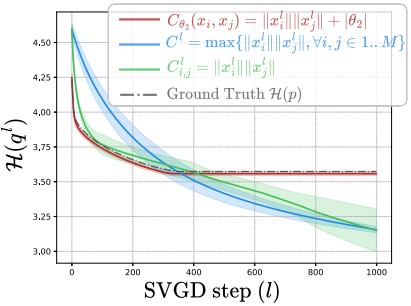

Figure 13: Bilinear kernel. $q_\theta^l$ coincides with theoretically derived intermediate distributions by Liu et al. (2024).

exactly as in Sec. C.2, *i.e.*, using Hutchinson's estimator (Hutchinson, 1989).

**Term ①:**

$$
\begin{aligned}
\mathrm{Tr}\left(\nabla_{x_i^l}(\kappa(x_i^l, x_j^l)\nabla_{x_j^l}\log p(x_j^l))\right) &= \mathrm{Tr}\left(\nabla_{x_i^l}\kappa(x_i^l, x_j^l)(\nabla_{x_j^l}\log p(x_j^l))^\top + \kappa(x_i^l, x_j^l)\nabla_{x_i^l}\nabla_{x_j^l}\log p(x_j^l))\right) \\
&= \sum_{t=1}^{d} \frac{\partial \kappa(x_i^l, x_j^l)}{\partial (x_i^l)^{(t)}}\frac{\partial \log p(x_j^l)}{\partial (x_j^l)^{(t)}} + 0 \\
&= (\nabla_{x_i^l}\kappa(x_i^l, x_j^l))^\top \nabla_{x_j^l}\log p(x_j^l) \\
&= -2\kappa(x_i^l, x_j^l)\left[\boxed{\nabla_{x_i^l}\psi(x_i^l)}^\top (\psi(x_i^l) - \psi(x_j^l))\right]^\top \nabla_{x_j^l}\log p(x_j^l)
\end{aligned}
$$

**Term ②**

$$
\begin{aligned}
\mathrm{Tr}\left(\nabla_{x_i^l}\nabla_{x_i^l}\kappa(x_i^l, x_j^l)\right) &= \mathrm{Tr}\left[\nabla_{x_i^l}\left(2\kappa(x_i^l, x_j^l)\nabla_{x_i^l}\psi(x_i^l)^\top (\psi(x_i^l) - \psi(x_j^l))\right)\right] \\
&= 2\left[\nabla_{x_i^l}\psi(x_i^l)^\top (\psi(x_i^l) - \psi(x_j^l))\right]^\top \nabla_{x_i^l}\kappa(x_i^l, x_j^l) \\
&\quad + 2\kappa(x_i^l, x_j^l)\,\mathrm{Tr}\left(\nabla_{x_i^l}\left[\nabla_{x_i^l}\psi(x_i^l)^\top (\psi(x_i^l) - \psi(x_j^l))\right]\right) \\
&= 2\left[\boxed{\nabla_{x_i^l}\psi(x_i^l)}^\top (\psi(x_i^l) - \psi(x_j^l))\right]^\top \nabla_{x_i^l}\kappa(x_i^l, x_j^l) \\
&\quad + 2\kappa(x_i^l, x_j^l)\,\mathrm{Tr}\left(\boxed{\nabla_{x_j^l}\psi(x_j^l)}^\top \boxed{\nabla_{x_i^l}\psi(x_i^l)}\right) \\
&\quad + 2\kappa(x_i^l, x_j^l)\,\mathrm{Tr}\left(\nabla_{x_i^l}\left[\boxed{\nabla_{x_j^l}\psi(x_j^l)}^\top (\psi(x_i^l) - \psi(x_j^l))\right]\right)
\end{aligned}
$$

$\nabla_{x_i^l}\psi(x_i^l)$ in **Term ①** and **Term ②**, introduces significant computational overhead and renders the density intractable. $\qquad\square$

### C.5 Derivation of the LD density

**Theorem.** *Let $p(x) = \bar{p}(x)/Z$ be a smooth, strictly positive target density on $\mathbb{R}^d$, and $q_0$ a strictly positive reference distribution. The closed-form estimate of the log-likelihood $\log q^L(x^L)$ for the LD-based sampler (Sec. A.1) after $L$ steps with a pre-specified noise schedule $\{\xi_l\}_{l=0}^{L-1}$ is*

$$
\log q^L(x^L) = \log q^0(x^0) - \epsilon \sum_{l=0}^{L-1} \mathbb{E}_{v\sim p_v}\left[v^T \nabla_{x^l}^2 \log p(x^l)v\right] + \mathcal{O}\left(\epsilon^2\right)
$$

*where $p_v$ is a distribution with zero mean and identity covariance, i.e., $\mathbb{E}_{v\sim p_v}[v] = 0$, $\mathbb{E}_{v\sim p_v}[vv^T] = I$, if $\epsilon\|\nabla_{x^l}^2 \log p(x^l)\|_\infty \ll 1$ for all $l \in [0, L-1]$.*

*Proof.*

$$
\begin{aligned}
\log q^{l+1}(x^{l+1}) &\overset{(i)}{=} \log q^l(x^l) - \log\left|\det\left(I + \epsilon \nabla_{x^l}^2 \log p(x^l)\right)\right| \\
&\overset{(ii)}{=} \log q^l(x^l) - \epsilon\,\mathrm{Tr}\left(\nabla_{x^l}^2 \log p(x^l)\right) + \mathcal{O}\left(\epsilon^2\right) \\
&\overset{(iii)}{=} \log q^l(x^l) - \epsilon \mathbb{E}_{v\sim p_v}\left[v^T \nabla_{x^l}^2 \log p(x^l)v\right] + \mathcal{O}\left(\epsilon^2\right)
\end{aligned}
$$

$(i)$ by the CVF (Sec. A.2)

$(ii)$ by the corollary of Jacobi's formula (Sec. A.4)

$(iii)$ using Hutchinson's Estimator, where $p_v$ is chosen such that $\mathbb{E}_{v\sim p_v}[v] = 0$ and $\mathbb{E}_{v\sim p_v}[vv^T] = I$ (eg. $p_v$ is Radamacher distribution)

The statement of the theorem follows by induction. $\qquad\square$

# D   DERIVATION OF STEP-SIZE BOUNDS

This appendix details the derivation of the step-size conditions in Proposition 3.1, Proposition 3.2, and Corollary 3.3 that ensure that *SVGD* updates are invertible and that $\log|\det \nabla_{x^l}\phi(x^l)|$ admits an accurate first-order approximation, where $\phi$ is the *SVGD* velocity field.

The structure of this appendix is as follows:

- In Sec. D.1, we review the Banach theorem
- In Sec. D.2, we leverage the Banach theorem to derive a step-size condition for the global invertibility of *SVGD* updates (Proposition 3.1): $\epsilon \max_{x^l} \|\nabla\phi(x^l)\|_2 < 1$
- In Sec. D.3, we derive another step-size condition for accurately estimating $\log|\det \nabla_{x^l}\phi^l(x)|$ (Proposition 3.2): $\epsilon |\lambda_{\max}(\nabla_{x^l}\phi(x^l))| < 1$
- In Sec. D.4, we unify the two derived step-size conditions into one upper bound that can be computed efficiently (Corollary 3.3): $\epsilon < \min_l \left(1/\sup_{x^l} \sqrt{\mathrm{Tr}(\nabla_{x^l}\phi(x^l)\nabla_{x^l}\phi^T(x^l))}\right)$
- In Sec. D.5, we show how to efficiently compute this upper bound

## D.1   BANACH THEOREM

We begin by introducing the concepts of a cauchy sequence and a contractive mapping. Next, we discuss the Banach Fixed Point theorem.

**Cauchy Sequence.** If a sequence $\{x_l\}_{l\in\mathbb{N}}$ satisfies **either** of the following conditions:

1. $|x_{l+1} - x_l| \le \alpha^l, \quad \forall l \in \mathbb{N}$
2. $|x_{l+2} - x_{l+1}| \le \alpha|x_{l+1} - x_l|, \quad \forall l \in \mathbb{N}$,

where $0 < \alpha < 1$, then $\{x_l\}$ is a Cauchy sequence.

**Contractive Mapping.** Let $(\mathcal{X}, d)$ be a metric space with $d$ a distance function and let $\phi : \mathcal{X} \to \mathcal{X}$ be a mapping on $\mathcal{X}$. $\phi$ is called a contraction if and only if:

$$\exists K \in [0, 1[ \quad \text{s.t.} \quad d(\phi(x), \phi(\tilde{x})) \le K d(x, \tilde{x}), \quad \forall x, \tilde{x} \in \mathcal{X} \tag{17}$$

**Banach Fixed Point Theorem.** Let $(X, d)$ be a complete metric space (*i.e.*, all **Cauchy Sequences** are convergent) with $d$ a distance function. If $\phi$ is a contraction, then it has a **unique fixed point** $x^* \in X$, *i.e.*, $\phi(x^*) = x^*$ and

$$\forall x_0 \in \mathcal{X}, \quad \lim_{l\to\infty} \phi^l(x_0) = x^*, \quad \text{with} \quad \phi^l(x_0) = \underbrace{\phi \circ \phi \circ \cdots \circ \phi}_{l \text{ times}}(x_0) = x_l.$$

*Proof.* The proof is structured in two main parts: we first establish the *existence* of a fixed point by showing that $(x_l)_{l\in\mathbb{N}}$ is a Cauchy sequence. Then prove *uniqueness* of the fixed point using a proof by contradiction.

**Step 1: Existence of a fixed point.** $(x_l)_{l\in\mathbb{N}}$ is a Cauchy sequence, we distinguish two cases: consecutive samples and non-consecutive samples.

- *consecutive samples:*

$$d(x_{l+1}, x_l) = d(\phi(x_l), \phi(x_{l-1})) \le K d(x_l, x_{l-1}) \le K^2 d(x_{l-1}, x_{l-2}) \le \cdots \le K^l d(x_1, x_0)$$

- *non-consecutive samples $x_l$ and $x_{l+k}$ with $k \in \mathbb{N}^*$*

$$d(x_{l+k}, x_l) \le d(x_{l+k}, x_{l+k-1}) + d(x_{l+k-1}, x_{l+k-2}) + \cdots + d(x_{l+1}, x_l)$$

$$\le (K^{l+k-1} + K^{l+k-2} + \cdots + K^l) d(x_1, x_0)$$

$$\leq K^l \underbrace{\sum_{t=0}^{k-1} K^t\, d(x_1, x_0)}_{\leq \sum_{t=0}^{\infty} K^t}$$

$$\leq K^l \left( \sum_{t=0}^{\infty} K^t \right) d(x_1, x_0) = \frac{K^l}{1 - K}\, d(x_1, x_0)$$

It follows that $\{x_l\}_{l \in \mathbb{N}}$ is a Cauchy sequence since $d(x_{l+k}, x_l) \to 0$ as $l \to \infty$. Because the metric space is complete, this implies convergence to a limit $x^* \in \mathcal{X}$: *i.e.*, , $x^* = \lim_{l \to \infty} x_l$. Additionally, since $\phi$ is continuous,

$$\phi(x^*) = \phi\left( \lim_{l \to \infty} x_l \right) = \lim_{l \to \infty} \phi(x_l) = \lim_{l \to \infty} x_{l+1} = x^*.$$

Hence, $x^*$ is a fixed point of $\phi$.

**Step 2: Uniqueness of the fixed point.** Assume that there exist two distinct fixed points $x^*$ and $\hat{x}$ such that $\phi(x^*) = x^*$ and $\phi(\hat{x}) = \hat{x}$. Then, If $x^* \neq \hat{x} \quad \Rightarrow \quad d(x^*, \hat{x}) = d(\phi(x^*), \phi(\hat{x})) \leq K\, d(x^*, \hat{x})$ Which implies $\Rightarrow \frac{d(x^*, \hat{x})}{d(x^*, \hat{x})} \leq K \Rightarrow 1 \leq K$. which contradicts the assumption that $K < 1$. Hence, the fixed point exists and is unique. We can compute it using the following algorithm:

$\square$

### D.2 SUFFICIENT CONDITION FOR $f(x) = x + \epsilon \phi(x)$ INVERTIBILITY (PROP. 3.1)

**Proposition 3.1** (Sufficient condition for *global SVGD* invertibility). *Let $p(x) = \bar{p}(x)/Z$ be a Lipschitz continuous target density on $\mathcal{X} \subset \mathbb{R}^d$, and let $f(x) = x + \epsilon \phi(x)$ be the SVGD update map with step size $\epsilon > 0$, where $\phi$ is the velocity field defined in Eq. 1 computed with an RBF kernel. If $\epsilon \max_x \|\nabla \phi(x)\|_2 < 1$, then $f$ is globally invertible.*

*Proof.* The proof follows in three steps:

- We first show that the *SVGD* velocity field $\phi$ is Lipschitz continuous under the proposition's assumption that the kernel used is the RBF kernel and that $p(x) = \bar{p}(x)/Z$ is Lipschitz continuous with Lipschitz constant $L_p$

- Then, we prove the proposition using the Banach theorem (Theorem D.1)

**Proof of the Lipschitz continuity of $\phi$:**

For the *SVGD* update rule:

$$\phi(x) = \mathbb{E}_{x_j \sim q} \left[ \kappa(x, x_j) \nabla_{x_j} \log \bar{p}(x_j) + \nabla_{x_j} \kappa(x, x_j) \right],$$

for any $x, y \in \mathbb{R}^d$, we compute:

$$\phi(x) - \phi(y) = \mathbb{E}_{x_j \sim q} \left[ \left( k(x, x_j) - k(y, x_j) \right) \nabla_{x_j} \log \bar{p}(x_j) + \left( \nabla_{x_i} k(x, x_j) - \nabla_{x_j} k(y, x_j) \right) \right].$$

Using the triangle inequality, we obtain:

$$\|\phi(x) - \phi(y)\| \leq \mathbb{E}_{x_j \sim q} \left[ |k(x, x_j) - k(y, x_j)|\, \|\nabla_{x_i} \log \bar{p}(x_j)\| + \|\nabla_{x_j} k(x, x_j) - \nabla_{x_i} k(y, x_j)\| \right].$$

The RBF kernel and its derivative are Lipschitz continuous. To show this, we use the mean value theorem:

$$\left| k(x, x_j) - k(y, x_j) \right| \leq \sup_z \|\nabla_z k(z, y)\|\, \|x - y\| = \frac{e^{-1/2}}{\sigma}\, \|x - y\| = L_k \|x - y\|.$$

It's easy to show that $\sup_z \|\nabla_z k(z, y)\| = \frac{e^{-1/2}}{\sigma}$ by solving $\nabla_z \|\nabla_z k(z, y)\| = 0$.

Similarly, $\nabla_{x_j} k(x, x_j)$ is Lipschitz continuous. By leveraging the multi-variate mean value theorem, we obtain:

$$\|\nabla_{x_j} k(x, x_j) - \nabla_{x_j} k(y, x_j)\| \leq \sup_z \|\nabla_z \nabla_{x_j} k(z, x_j))\|_2 \, \|x - y\| = \frac{1}{\sigma^2} \|x - y\| = L_{\nabla k} \|x - y\|$$

where

$$\|\nabla_z \nabla_{x_i} k(z, x_j)\|_2 = \left\| k(z, x_j) \left( \frac{1}{\sigma^2} I - \frac{1}{\sigma^4} (z - x_j)(z - x_j)^T \right) \right\|_2 = \frac{1}{\sigma^2} k(z, x_j) \max \left( 1, \left| 1 - \frac{1}{\sigma^2} \|z - x_j\|^2 \right| \right)$$

We conclude that:

$$\|\phi(x) - \phi(y)\| \leq \mathbb{E}_{x_j \sim q} \left[ (L_k \, L_p + L_{\nabla k}) \, \|x - y\| \right] = (L_k \, L_p + L_{\nabla k}) \|x - y\|.$$

Hence, $\phi$ is Lipschitz continuous.

**Proof of the Proposition:**

Assume that $\mathrm{Lip}(\epsilon \phi) < 1$, *i.e.*, $\epsilon \phi$ is contractive. We want to show that $f(x) = x + \epsilon \phi(x)$ is invertible. More concretely, given $y = x + \phi(x)$, the goal is to find $x$. We denote $y$ by $c$ resulting in $x = c - \epsilon \phi(x)$. Hence, we are interested in the invertibility of the function $g(x) = c - \epsilon \phi(x)$. For this, we show that $g$ is a contractive mapping:

$$d(g(x), g(\tilde{x})) = d(c - \epsilon \phi(x), c - \epsilon \phi(\tilde{x}))$$
$$\stackrel{(i)}{=} d(-\epsilon \phi(x), -\epsilon \phi(\tilde{x}))$$
$$\stackrel{(ii)}{=} d(\epsilon \phi(x), \epsilon \phi(\tilde{x}))$$
$$\stackrel{(iii)}{<} d(x, \tilde{x})$$

$(i)$ The distance is translation invariant.

$(ii)$ The distance is absolutely homogeneous.

$(iii)$ $\epsilon \phi$ is a contractive mapping

Therefore, $g(x)$ is a contractive mapping, and by the **Banach fixed point theorem** (Theorem D.1), it has a unique fixed point. This implies that the inverse of the mapping $y = x + \epsilon \phi(x)$ exists and is unique. Hence, the *SVGD* update is invertible if $\mathrm{Lip}(\epsilon \phi) < 1$.

---

**Algorithm 3** BANACHFIXEDPOINT

---

**input** Particles $\{x_j^l\}_{j=0}^{M-1}$; maximum number of iterations $B$; unnormalized density $\bar{p}$; kernel bandwidth $\sigma_{\theta_2}^l$; step-size $\epsilon_{\theta_3}^l$.
**output** Fixed point $x_i^{l,*}$ satisfying $x_i^l = x_i^{l,*} + \epsilon_{\theta_3}^l \phi_\theta(x_i^{l,*})$
1: $x_i^{l,*} \leftarrow x_i^l$
2: **for** $b = 0 \cdots B - 1$ **do**
3:    $x_i^{l,*} \leftarrow x_i^l - \epsilon \phi(x_i^{l,*})$
4: **end for**
5: **return** $x_i^{l,*}$

---

We compute $\mathrm{Lip}(\phi)$ as:

$$\mathrm{Lip}(\phi) = \sup_{x \neq y} \frac{\|\phi(x) - \phi(y)\|}{\|x - y\|} \stackrel{(i)}{=} \sup_x \|\nabla_x \phi(x)\|_2 \stackrel{(ii)}{=} \sup_x \sigma_{\max}(\nabla_x \phi(x))$$

$(i)$ We consider the $\ell_2$ norm in computing the operator norm.

(*ii*) Using the definition of the Lipschitz constant via Jacobian norm: $\|\phi(x) - \phi(y)\| \leq \sup_x \|\nabla_x \phi(x)\|_2 \cdot \|x - y\|$.

$\square$

### D.3 A SUFFICIENT CONDITION FOR $\log |\det(I + \epsilon \nabla_x \phi(x))| = \epsilon \operatorname{Tr}(\nabla_x \phi(x)) + \mathcal{O}\left(\epsilon^2 \lambda_{\max}^2(\nabla_x \phi(x))\right)$ APPROXIMATION (PROP. 3.2)

**Proposition 3.2** (Condition for log-det Approximation) *Let $\phi : \mathbb{R}^d \to \mathbb{R}^d$ be a differentiable vector-valued map.* $\log |\det(I + \epsilon \nabla_x \phi(x))| = \epsilon \operatorname{Tr}(\nabla_x \phi(x)) + \mathcal{O}\left(\epsilon^2 \lambda_{max}^2(\nabla_x \phi(x))\right)$, *if* $\epsilon |\lambda_{max}(\nabla_x \phi(x))| < 1$, *with $\lambda_{max}$ being the largest eigenvalue value in magnitude.*

*Proof.* We discuss two approaches leveraging the corollary of Jacobi's formula and the bounds on the eigenvalues of $\nabla_{x^l} \phi(x^l)$:

**Method 1 (*P-SVGD*):** Let $A = I + \epsilon \nabla_x \phi(x)$. Under the assumption $\epsilon \|\nabla_x \phi(x)\|_\infty \ll 1$, *i.e.*, $\|A - I\|_\infty \ll 1$, we apply the collorary of Jacobi's formula (App. A.4) and get

$$\log |\det(I + \epsilon \nabla_x \phi(x))| = \epsilon \operatorname{Tr}\left((I + \epsilon \nabla_x \phi(x) - I)\right) + \mathcal{O}(\epsilon^2 \|\nabla_x \phi(x)\|_\infty^2)$$
$$\overset{(i)}{=} \epsilon \operatorname{Tr}\left(\nabla_x \phi(x)\right) + \mathcal{O}(\epsilon^2)$$

(*i*) In practice, since this bound is informal, (Messaoud et al., 2024) recommend choosing a very small learning rate, hence $\mathcal{O}(\epsilon^2)$.

**Method 2 (*MET-SVGD*):** In the following we denote by $\lambda_i(A)$ the eigenvalue of matrix $A$

$$\left|\det\left(I + \epsilon \nabla_x \phi(x^l)\right)\right| \overset{(i)}{=} \left|\prod_{i=1}^d \lambda_i(I + \epsilon \nabla_x \phi(x))\right| = \prod_{i=1}^d |\lambda_i(I + \epsilon \nabla_x \phi(x))|$$

$$\overset{(ii)}{=} \prod_{j=1}^d |1 + \epsilon \lambda_j(\nabla_x \phi(x))| = \exp\left(\sum_{j=1}^d \ln |1 + \epsilon \lambda_j(\nabla_x \phi(x))|\right)$$

$$= \exp\left(\sum_{j=1}^d \ln\left(1 + \epsilon \lambda_j(\nabla_x \phi(x))\right)\right) \quad \text{if } \lambda_j(\nabla_x \phi(x)) > \frac{-1}{\epsilon}$$

$$\overset{(iii)}{=} \exp\left(\sum_{j=1}^d \epsilon \lambda_j(\nabla_x \phi(x)) + \mathcal{O}((\epsilon \lambda_j^l)^2)\right) \quad \text{if } |\epsilon \lambda_j(\nabla_x \phi(x))| < 1$$

(*i*) By definition of the determinant.

(*ii*) Let $\lambda_i$ be the eigenvalue of $(I + \epsilon \nabla_x \phi(x))$ associated with the eigenvector $v_i$. We show that $\lambda_i - 1$ is the eigenvalue associated with $\epsilon \nabla_x \phi(x)$:

$$(I + \epsilon \nabla_x \phi(x))v_i = \lambda_i v_i \implies \epsilon \nabla_x \phi(x) v_i = (\lambda_i - 1) v_i \implies \lambda_j = (\lambda_i - 1) \text{ is an eigenvalue of } \epsilon \nabla_x \phi(x)$$

(*iii*) We use Taylor expansion of $\ln(1 + \epsilon a) = \sum_i \frac{(-1)^{i-1}(\epsilon a)^i}{i} = \epsilon a + \mathcal{O}((\epsilon a)^2)$ around $\epsilon a \to 0$.

Hence, if $\epsilon |\lambda_i(\nabla_x \phi(x))| \leq \epsilon |\lambda_{\max}(\nabla_x \phi(x))| < 1$, then $\log \left|\det\left(I + \epsilon \nabla_x \phi(x)\right)\right| = \epsilon \operatorname{Tr}(\nabla_x \phi(x)) + \mathcal{O}\left(\epsilon^2 \lambda_{\max}^2(\nabla_{x^l} \phi(x))\right)$. $\square$

## D.4 Unifying the sufficient conditions for invertibility and
$$\log|\det(I + \epsilon\nabla_{x^l}\phi(x^l))| = \epsilon\operatorname{Tr}(\nabla_{x^l}\phi(x^l)) + \mathcal{O}\left(\epsilon^2\lambda_{\max}^2(\nabla_{x^l}\phi(x^l))\right)$$

**Corollary 3.3** *Given a Lipschitz continuous target $p = \frac{\bar{p}}{Z}$, the distribution induced by the SVGD update (Eq. 1) using an RBF kernel is given by Eq. 2 if $\epsilon < \min_l \epsilon_{UB}^l$ with $\epsilon_{UB}^l = 1/\sup_{x^l}\sqrt{\operatorname{Tr}(\nabla_{x^l}\phi(x^l)\nabla_{x^l}\phi^T(x^l))}$ for all $l \in [0, L-1]$.*

*Proof.* Following (Wolkowicz & Styan, 1980), let $A$ be an $d \times d$ complex matrix, and let $A^*$ be the Hermitian of $A$. Then:
$$|\lambda_i| \le \sigma_i \le (\operatorname{Tr}(A^*A))^{1/2} \quad \forall i \in [1..d],$$
where $\sigma_i$ is the i-th singular value of $A$.

It follows that:
$$|\lambda_{\max}\{\nabla_{x^l}\phi(x^l)\}| \le \sup_{x^l}||\nabla_{x^l}\phi(x^l)||_2 \le \sup_{x^l}\sqrt{\operatorname{Tr}\left(\nabla_{x^l}\phi(x^l)^\top\nabla_{x^l}\phi(x^l)\right)}$$

Therefore, choosing $\epsilon$ such that $\epsilon < \min_l\left(1/\sup_{x^l}\sqrt{\operatorname{Tr}\left(\nabla_{x^l}\phi(x^l)^\top\nabla_{x^l}\phi(x^l)\right)}\right)$ satisfies both of:

- $\epsilon\max_{x^l}||\nabla\phi(x^l)||_2 < 1$ (from Proposition 3.1)

- $\epsilon|\lambda_{\max}\left(\nabla_{x^l}\phi(x^l)\right)| < 1$ (from Proposition 3.2)

In Proposition 3.2, we use $\mathcal{O}\left(\epsilon^2\lambda_{\max}^2(\nabla_{x^l}\phi(x^l))\right)$. We re-express this as a function of $\epsilon_{UB}^l$:

$$\mathcal{O}\left(\left(\epsilon|\lambda_{\max}(\nabla_{x^l}\phi(x^l))|\right)^2\right) \overset{(i)}{=} \mathcal{O}\left(\left(\frac{\epsilon_{UB}^l}{\epsilon_{UB}^l}\epsilon|\lambda_{\max}(\nabla_{x^l}\phi(x^l))|\right)^2\right)$$

$$\overset{(ii)}{=} \mathcal{O}\left(\left(\frac{\epsilon}{\epsilon_{UB}^l}\frac{|\lambda_{\max}(\nabla_{x^l}\phi(x^l))|}{\sup_x\sqrt{\operatorname{Tr}\left(\nabla_{x^l}\phi(x^l)^\top\nabla_{x^l}\phi(x_i^l)\right)}}\right)^2\right)$$

$$\overset{(iii)}{=} \mathcal{O}\left(\left(\frac{\epsilon}{\epsilon_{UB}^l}\right)^2\right)$$

(*i*) Multiply by $1 = \epsilon_{UB}^l/\epsilon_{UB}^l$

(*ii*) Substitute in the expression of $\epsilon_{UB}^l$

(*iii*) Using the fact that $|\lambda_{\max}\left(\nabla_{x^l}\phi(x^l)\right)|/\sup_{x^l}\sqrt{\operatorname{Tr}\left(\nabla_{x^l}\phi(x^l)^\top\nabla_{x^l}\phi(x^l)\right)} \le 1$

It follows that
$$\log|\det(I + \epsilon\nabla_{x^l}\phi(x^l))| = \epsilon\operatorname{Tr}(\nabla_{x^l}\phi(x^l)) + \mathcal{O}\left((\epsilon/\epsilon_{UB}^l)^2\right) \tag{18}$$
$\square$

## D.5 Computing $\operatorname{Tr}\left(\nabla_{x_i^l}\phi(x_i^l)^\top\nabla_{x_i^l}\phi(x_i^l)\right)$

In this section, we show that $\operatorname{Tr}\left(\nabla_{x_i^l}\phi(x_i^l)^\top\nabla_{x_i^l}\phi(x_i^l)\right)$ can be computed efficiently, *i.e.*, using only first-order derivatives and vector dot products.

For conciseness, we introduce the quantity $s_p(x_j^l) = \nabla_{x_j}\log p(x_j^l)$.

$$\nabla_{x_i^l}\phi(x_i^l) = \underbrace{\frac{1}{M}\sum_{j=0,j\neq i}^{M-1}\nabla_{x_i^l}\kappa(x_i^l, x_j^l)s_p(x_j^l)^T + \nabla_{x_i^l}\nabla_{x_j^l}\kappa(x_i^l, x_j^l)}_{A_i} + \underbrace{\frac{1}{M}\nabla_{x_i^l}s_p(x_i^l)^T}_{B_i}$$

Next we compute $\text{Tr}(\nabla_{x_i^l}\phi(x_i^l))$. We denote by $A_i$ and $B_i$ the two terms of $\nabla_{x_i^l}\phi(x_i^l)$:

$$
\begin{aligned}
\text{Tr}\left((\nabla_{x_i^l}\phi(x_i^l))^T\nabla_{x_i^l}\phi(x_i^l)\right) &= \text{Tr}\left((A_i+B_i)^T(A_i+B_i)\right)\\
&= \text{Tr}\left(A_i^T A_i + B_i^T B_i + A_i^T B_i + A_i B_i^T\right)\\
&= \text{Tr}(A_i^T A_i) + \text{Tr}(B_i^T B_i) + 2\,\text{Tr}(A_i B_i^T)\\
&= \underbrace{\text{Tr}(A_i^T A_i)}_{(1)} + \underbrace{\text{Tr}(B_i^T B_i)}_{(2)} + \underbrace{2\,\text{Tr}(B_i^T A_i)}_{(3)}
\end{aligned}
$$

For a term by term breakdown:

Term (1) $= \text{Tr}(A_i^T A_i)$

$$
= \text{Tr}\left(\Big(\frac{1}{M}\sum_{\substack{j=0\\j\neq i}}^{M-1}\overbrace{\nabla_{x_i^l}\kappa(x_i^l,x_j^l)s_p(x_j^l)^T}^{C_{i,j}}+\overbrace{\nabla_{x_i^l}\nabla_{x_j^l}\kappa(x_i^l,x_j^l)}^{D_{i,j}}\Big)^T\Big(\frac{1}{M}\sum_{\substack{r=0\\r\neq i}}^{M-1}\underbrace{\nabla_{x_i^l}\kappa(x_i^l,x_r^l)s_p(x_r^l)^T}_{C_{i,r}}+\underbrace{\nabla_{x_i^l}\nabla_{x_r^l}\kappa(x_i^l,x_r^l)}_{D_{i,r}}\Big)\right)
$$

$$
= \text{Tr}\left(\frac{1}{M^2}\sum_{\substack{j=0\\j\neq i}}^{M-1}\sum_{\substack{r=0\\r\neq i}}^{M-1}\Big(C_{i,j}{}^T + D_{i,j}{}^T\Big)\Big(C_{i,r}+D_{i,r}\Big)\right)
$$

$$
= \frac{1}{M^2}\sum_{\substack{j=0\\j\neq i}}^{M-1}\sum_{\substack{r=0\\r\neq i}}^{M-1}\underbrace{\text{Tr}(C_{i,j}^T C_{i,r}+D_{i,j}^T C_{i,r})}_{(1a)}+\underbrace{\text{Tr}(D_{i,j}^T D_{i,r}+C_{i,j}^T D_{i,r})}_{(1b)}
$$

Term (1a) $= \text{Tr}(C_{i,r}^T C_{i,j}+D_{i,r}^T C_{i,j})$

$$
\begin{aligned}
&= \text{Tr}\left((\nabla_{x_i^l}\kappa(x_i^l,x_r^l)s_p(x_r^l)^T)^T(\nabla_{x_i^l}\kappa(x_i^l,x_j^l)s_p(x_j^l)^T)+(\nabla_{x_i^l}\nabla_{x_r^l}\kappa(x_i^l,x_r^l))^T(\nabla_{x_i^l}\kappa(x_i^l,x_j^l)s_p(x_j^l)^T)\right)\\
&= \text{Tr}\left(s_p(x_r^l)\nabla_{x_i^l}\kappa(x_i^l,x_r^l)^T\nabla_{x_i^l}\kappa(x_i^l,x_j^l)s_p(x_j^l)^T+(\nabla_{x_i^l}\nabla_{x_r^l}\kappa(x_i^l,x_r^l))^T(\nabla_{x_i^l}\kappa(x_i^l,x_j^l)s_p(x_j^l)^T)\right)\\
&= \text{Tr}\left(4\gamma^2\kappa(x_i^l,x_r^l)\kappa(x_i^l,x_j^l)s_p(x_r^l)\delta_{i,r}^T\delta_{i,j}s_p(x_j^l)^T-4\gamma^2\kappa(x_i^l,x_r^l)\kappa(x_i^l,x_j^l)\big(I-2\gamma\delta_{i,r}\delta_{i,r}^T\big)\delta_{i,j}s_p(x_j^l)^T\right)\\
&= \text{Tr}\left(4\gamma^2\kappa(x_i^l,x_r^l)\kappa(x_i^l,x_j^l)\big(s_p(x_r^l)\delta_{i,r}^T-I+2\gamma\delta_{i,r}\delta_{i,r}^T\big)\delta_{i,j}s_p(x_j^l)^T\right)\\
&= 4\gamma^2\kappa(x_i^l,x_r^l)\kappa(x_i^l,x_j^l)\big(\delta_{i,r}^T s_p(x_r^l)-d+2\gamma\|\delta_{i,r}\|^2\big)s_p(x_j^l)^T\delta_{i,j}
\end{aligned}
$$

Term (1b) $= \text{Tr}(D_{i,r}^T D_{i,j}+C_{i,r}^T D_{i,j})$

$$
\begin{aligned}
&= \text{Tr}\left((\nabla_{x_i^l}\nabla_{x_r^l}\kappa(x_i^l,x_r^l))^T(\nabla_{x_i^l}\nabla_{x_j^l}\kappa(x_i^l,x_j^l))+(\nabla_{x_i^l}\kappa(x_i^l,x_r^l)s_p(x_r^l)^T)^T(\nabla_{x_i^l}\nabla_{x_j^l}\kappa(x_i^l,x_j^l))\right)\\
&= \text{Tr}\left(4\gamma^2\kappa(x_i^l,x_r^l)\kappa(x_i^l,x_j^l)\big(I-2\gamma\delta_{i,r}\delta_{i,r}^T\big)\big(I-2\gamma\delta_{i,j}\delta_{i,j}^T\big)-4\gamma^2\kappa(x_i^l,x_r^l)\kappa(x_i^l,x_j^l)s_p(x_r^l)\delta_{i,r}^T\big(I-2\gamma\delta_{i,j}\delta_{i,j}^T\big)\right)\\
&= \text{Tr}\left(4\gamma^2\kappa(x_i^l,x_r^l)\kappa(x_i^l,x_j^l)\big(I-2\gamma\delta_{i,r}\delta_{i,r}^T-s_p(x_r^l)\delta_{i,r}^T\big)\big(I-2\gamma\delta_{i,j}\delta_{i,j}^T\big)\right)\\
&= 4\gamma^2\kappa(x_i^l,x_r^l)\kappa(x_i^l,x_j^l)\big(d-\delta_{i,r}^T s_p(x_r^l)-2\gamma|\delta_{i,r}|^2\big)\big(d-2\gamma|\delta_{i,j}|^2\big)
\end{aligned}
$$

Adding these sub-terms together

**Term ①** $= \dfrac{1}{M^2}\displaystyle\sum_{\substack{j=0\\j\neq i}}^{M-1}\sum_{\substack{r=0\\r\neq i}}^{M-1}4\gamma^2\kappa(x_i^l,x_r^l)\kappa(x_i^l,x_j^l)\big(\delta_{i,r}^T s_p(x_r^l)-d+2\gamma\|\delta_{i,r}\|^2\big)s_p(x_j^l)^T\delta_{i,j}$

$$
+4\gamma^2\kappa(x_i^l,x_r^l)\kappa(x_i^l,x_j^l)\big(d-\delta_{i,r}^T s_p(x_r^l)-2\gamma\|\delta_{i,r}\|^2\big)\big(d-2\gamma\|\delta_{i,j}\|^2\big)
$$

$$= \frac{1}{M^2} \sum_{\substack{j=0 \\ j \neq i}}^{M-1} \sum_{\substack{r=0 \\ r \neq i}}^{M-1} 4\gamma^2 \kappa(x_i^l, x_r^l) \kappa(x_i^l, x_j^l) \Big( \delta_{i,r}^T s_p(x_r^l) - d + 2\gamma \|\delta_{i,r}\|^2 \Big) \Big( s_p(x_j^l)^T \delta_{i,j} - d + 2\gamma \|\delta_{i,j}\|^2 \Big)$$

**Term ②** $= \text{Tr}(B_i^T B_i)$

$$= \text{Tr} \left( \frac{1}{M^2} \nabla_{x_i^l} s_p(x_i^l) \Big( \nabla_{x_i^l} s_p(x_i^l) \Big)^T \right)$$

$$= \frac{1}{VM^2} \sum_{t=1}^{V} v_t^T \nabla_{x_i^l} s_p(x_i^l) \Big( \nabla_{x_i^l} s_p(x_i^l) \Big)^T v_t$$

$$= \frac{1}{VM^2} \sum_{t=1}^{V} \Big\| \nabla_{x_i^l} \Big( v_t^T s_p(x_i^l) \Big) \Big\|^2$$

**Term ③** $= \text{Tr}(B_i^T A_i)$

$$\approx \frac{1}{V} \sum_{t=1}^{V} v_t^T \left( \frac{1}{M^2} \sum_{\substack{j=0 \\ j \neq i}}^{M-1} \Big[ \underbrace{-2\gamma \nabla_{x_i^l} s_p(x_i^l) \delta_{i,j} s_p(x_j^l)^T}_{E_{i,j}} + \underbrace{2\gamma \nabla_{x_i^l} s_p(x_i^l)(I - 2\gamma \delta_{i,j} \delta_{i,j}^T)}_{F_{i,j}} \Big] \kappa(x_i^l, x_j^l) \right) v_t$$

Using Hutchinson's Trace Estimator Hutchinson (1989)

$$\approx \frac{1}{V} \sum_{t=1}^{V} \frac{1}{M^2} \sum_{\substack{j=0 \\ j \neq i}}^{M-1} \kappa(x_i^l, x_j^l) \Big[ v_t^T E_{i,j} v_t + v_t^T F_{i,j} v_t \Big]$$

$$\approx \frac{1}{VM^2} \sum_{t=1}^{V} \sum_{\substack{j=0 \\ j \neq i}}^{M-1} \kappa(x_i^l, x_j^l) \Big[ -2\gamma (v_t^T \nabla_{x_i^l} s_p(x_i^l))(\delta_{i,j}^T v_t) s_p(x_j^l)^T + 2\gamma (v_t^T \nabla_{x_i^l} s_p(x_i^l))(v_t - 2\gamma(\delta_{i,j}^T v_t)\delta_{i,j}) \Big]$$

By combining **Terms ①**, **②** and **③**, we obtain:

$$(1) + (2) + (3) = \frac{1}{M^2} \sum_{j=0}^{M-1} \sum_{\substack{j \neq i}}^{} \sum_{r=0}^{M-1} \sum_{\substack{r \neq i}}^{} 4\gamma^2 \kappa(x_i^l, x_r^l) \kappa(x_i^l, x_j^l) \Big( \delta_{i,r}^T s_p(x_r^l) - d + 2\gamma|\delta_{i,r}|^2 \Big) \Big( s_p(x_j^l)^T \delta_{i,j} - d + 2\gamma|\delta_{i,j}|^2 \Big)$$

$$+ \frac{2}{M^2} |\nabla_{x_i^l} s_p(x_i^l)|^2$$

$$+ \frac{1}{VM^2} \sum_{t=1}^{V} \sum_{\substack{j=0 \\ j \neq i}}^{M-1} \kappa(x_i^l, x_j^l) \Big[ -2\gamma (v_t^T \nabla_{x_i^l} s_p(x_i^l))(\delta_{i,j}^T v_t) s_p(x_j^l)^T + 2\gamma (v_t^T \nabla_{x_i^l} s_p(x_i^l))(v_t - 2\gamma(\delta_{i,j}^T v_t)\delta_{i,j}) \Big]$$

# E   MOTIVATION: LEARNING THE SVGD LEARNING RATE (SEC. 3.2-STEP-SISE)

Learning the kernel bandwidth alone is generally insufficient to ensure convergence of the entropy term. Specifically, the expectation $\mathbb{E}_{x^l \sim q^l}[\epsilon \operatorname{Tr}(\nabla_{x^l} \phi(x^l))]$ does not necessarily vanish as $l \to \infty$. We show, via a Taylor expansion around $0$, that this cumulative trace term corresponds to a 8$^{\text{th}}$-degree polynomial whose convergence to zero requires the existence of at least one real root. However, the coefficients of this polynomial depend on the particle positions and are not guaranteed to yield a real root during training, making this condition both non-trivial and fragile.

For conciseness, we introduce three new quantities:

$$\kappa_j = \kappa(x^l, x_j^l), \quad \delta_j = x^l - x_j^l, \quad \text{and} \quad s_j = \nabla_{x_j^l} \log p(x_j^l).$$

*Proof.* According to Sec. C.2, if $\kappa$ is the RBF kernel, then:

$$\operatorname{Tr}(\nabla_{x^l} \phi(x^l)) = \frac{1}{M} \operatorname{Tr}\left(\nabla_{x^l}^2 \log p(x^l)\right) + \frac{1}{M\sigma^2} \sum_{\substack{j=0 \\ x_j^l \neq x^l}}^{M-1} k_j \left(d - \delta_j^\top s_j - \frac{1}{\sigma^2}||\delta_j||^2\right)$$

We approximate the RBF kernel using a Taylor expansion around $0$:

$$\exp\left(-\frac{1}{2\sigma^2}||\delta_j||^2\right) \approx 1 - \frac{1}{2\sigma^2}||\delta_j||^2 + \frac{1}{8\sigma^4}||\delta_j||^4$$

Then, we substitute in the formula above and obtain:

$$\operatorname{Tr}(\nabla_{x^l} \phi(x^l)) = \frac{1}{M} \operatorname{Tr}\left(\nabla_{x^l}^2 \log p(x^l)\right) + \frac{1}{M\sigma^2} \sum_{\substack{j=0 \\ x_j^l \neq x^l}}^{M-1} \left(1 - \frac{1}{2\sigma^2}||\delta_j||^2 + \frac{1}{8\sigma^4}||\delta_j||^4\right) \left(d - \delta_j^\top s_j - \frac{1}{\sigma^2}||\delta_j||^2\right)$$

Finally, we set $\operatorname{Tr}(\nabla_{x^l} \phi(x^l)) = 0$ and multiply by $\sigma^8$:

$$\frac{1}{M}\sigma^8 \operatorname{Tr}\left(\nabla_{x^l}^2 \log p(x^l)\right) + \frac{1}{M} \sum_{\substack{j=0 \\ x_j^l \neq x^l}}^{M-1} \left(\sigma^4 - \frac{1}{2}\sigma^2||\delta_j||^2 + \frac{1}{8}||\delta_j||^4\right) \left(\sigma^2 d - \sigma^2 \delta_j^\top s_j - ||\delta_j||^2\right) = 0$$

Since we have a polynomial of degree 8, we have 8 roots, not all of which are guaranteed to be real for $\sigma$. Thus, for convergence, $\epsilon$ needs to be flexible enough to converge to 0, ensuring that $\mathbb{E}_{x^l \sim q^l}[\epsilon \operatorname{Tr}(\nabla_{x^l} \phi(x^l))]$ converges to 0 as $l \to \infty$. $\qquad\square$

# F  A MORE EFFICIENT VARIANT OF STEIN'S IDENTITY

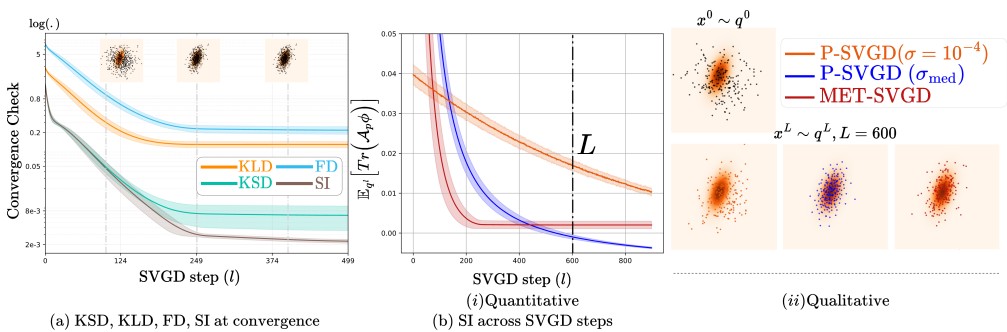

(a) KSD, KLD, FD, SI at convergence          (b) SI across SVGD steps

Figure 14: (a) SI shows the same convergence trend as other convergence metrics such as Fisher Divergence (FD) and Kernelized Stein Discrepancy (KSD). With SI being the tractable and computationally efficient metric. (b) SI can be used to check SVGD convergence across steps, for *MET-SVGD* and *P-SVGD*($\sigma_{\text{med}}, \sigma = 10^{-4}$).

**Proposition 3.4** Given $\phi(x)$ the *SVGD* velocity field with an RBF kernel from Eq. 1, a smooth and strictly positive density $p(x)$ with $\lim_{\|x\|\to\infty} p(x)\,\phi(x) = 0$, Stein's identity violation SI($q^l, p$) at every step $l$ of the *SVGD* update is evaluated as:

$$\text{SI}(q^l, p) = \sqrt{\mathbb{E}_{x^l}\left[\phi(x^l)^T \nabla_{x^l} \log p(x^l) + \text{Tr}(\nabla_{x^l}\phi(x^l))\right]} \tag{19}$$

*Proof.* Stein identity's violation can be measured with the kernelized Stein discrepancy (KSD) of Liu et al. (2016), which is defined as

$$\mathbb{S}(q^l, p) := \max_{\phi \in \mathcal{H}^D} \left(\mathbb{E}_{x^l \sim q^l}\left[\text{Tr}(\mathcal{A}_p \phi(x^l))\right]\right)^2 \quad \text{s.t. } \|\phi\|_{\mathcal{H}^D} \leq 1,$$

where the Stein operator is $\mathcal{A}_p \phi(x) = \phi(x)\nabla_x \log p(x)^\top + \nabla_x \phi(x)$.

We show in Sec. A.8 that, in the RKHS $\mathcal{H}^D$, the maximizer admits the closed form $\phi_{q^l, p}(\cdot) = \phi^*_{q^l, p}(\cdot)/\|\phi^*_{q^l, p}(\cdot)\|_{\mathcal{H}^D}$, with

$$\phi^*_{q^l, p}(\cdot) = \mathbb{E}_{x^l \sim q^l}\left[k(x^l, \cdot)\,\nabla_{x^l} \log p(x^l) + \nabla_{x^l} k(x^l, \cdot)\right],$$

and the optimal value satisfies $\mathbb{S}(q^l, p) = \|\phi^*_{q^l, p}\|^2_{\mathcal{H}^D}$. We, therefore, set SI($q^l, p$) = $\|\phi^*_{q^l, p}\|_{\mathcal{H}^D}$.

$$\begin{aligned}
\mathbb{S}(q^l, p) &= \langle \phi^*_{q^l, p}(\cdot), \phi^*_{q^l, p}(\cdot) \rangle_{\mathcal{H}^D} \\
&\overset{(i)}{=} \left\langle \mathbb{E}_{x^l \sim q^l}\left[k(x^l, \cdot)\,\nabla_{x^l} \log p(x^l) + \nabla_{x^l} k(x^l, \cdot)\right],\ \phi^*_{q^l, p}(\cdot)\right\rangle_{\mathcal{H}^D} \\
&\overset{(ii)}{=} \mathbb{E}_{x^l \sim q^l}\left[\left\langle k(x^l, \cdot)\,\nabla_{x^l} \log p(x^l),\ \phi^*_{q^l, p}(\cdot)\right\rangle_{\mathcal{H}^D} + \left\langle \nabla_{x^l} k(x^l, \cdot),\ \phi^*_{q^l, p}(\cdot)\right\rangle_{\mathcal{H}^D}\right] \\
&\overset{(iii)}{=} \mathbb{E}_{x^l \sim q^l}\left[\left(\phi^*_{q^l, p}(x^l)\right)^\top \nabla_{x^l} \log p(x^l) + \text{Tr}\left(\nabla_{x^l} \phi^*_{q^l, p}(x^l)\right)\right].
\end{aligned}$$

Taking the square root yields

$$\text{SI}(q^l, p) = \sqrt{\mathbb{E}_{x^l \sim q^l}\left[\left(\phi^*_{q^l, p}(x^l)\right)^\top \nabla_{x^l} \log p(x^l) + \text{Tr}\left(\nabla_{x^l} \phi^*_{q^l, p}(x^l)\right)\right]}.$$

($i$) substitute the closed-form representation of the optimal perturbation $\phi^*_{q^l, p}$ into the first argument of the inner product

($ii$) apply linearity of the inner product and of the expectation: $\langle \mathbb{E}[f_x], g \rangle = \mathbb{E}[\langle f_x, g \rangle]$

$(iii)$ use the reproducing property for vector-valued RKHS, $\langle k(x,\cdot)v, \phi(\cdot)\rangle_{\mathcal{H}^D} = v^\top \phi(x)$, and its derivative counterpart, $\langle \nabla_x k(x,\cdot), \phi(\cdot)\rangle_{\mathcal{H}^D} = \mathrm{Tr}(\nabla_x \phi(x))$, which together convert RKHS inner products into the function-space terms appearing inside the expectation

$\square$

# G    METROPOLIS HASTINGS AUGMENTED ENTROPY

We first prove that *MET-SVGD* converges in total variation to the target distribution. Then, provide derivations for (1) the acceptance probability (Proposition 3.5), and (2) MH-augmented *SVGD* density (Proposition 3.6).

## G.1    MET-SVGD CONVERGENCE

To transform the *SVGD* chain into a Markov chain with asymptotic convergence guarantees (strong convergence), the *SVGD* proposal must be irreducible and aperiodic (App. B.5-B.6). This is, however, not satisfied by the deterministic *SVGD* transformation.

One approach to achieve these properties is to inject Gaussian noise. This occurs naturally when the expectation in the *SVGD* update is approximated independently for each particle using its own mini-batch. By the Central Limit Theorem, the empirical mean used to approximate the expectation converges in distribution to a Gaussian. Consequently, the stochasticity introduced by this sampling scheme is equivalent to injecting Gaussian noise into the *SVGD* update, which ensures irreducibility and aperiodicity of the resulting Markov chain. One may also inject Gaussian noise explicitly. However, once additive noise is introduced, the update map is no longer invertible. Since invertibility is required to apply the change-of-variables formula to relate densities before and after the update, such a stochastic *SVGD* transformation cannot rely on this formula.

Instead, we construct an involution associated with the *SVGD* transformation $f$ by extending the state space to $(x, r)$, where $r \in \{-1, 1\}$ is a Rademacher random variable:

$$G(x, r) = \begin{cases} (x, r = +1) \mapsto (f(x), r = -1), \\ (x, r = -1) \mapsto (f^{-1}(x), r = +1). \end{cases}$$

This construction guarantees reversibility (*i.e.*, detailed balance holds) and also yields an irreducible and aperiodic chain, as the auxiliary variable $r$ prevents immediate backtracking and enables exploration of the full state space. These dynamics are analogous to those used in Hamiltonian Monte Carlo Betancourt (2017).

When $r = 1$, the proposal is $\tilde{x}^l = f(x^{l-1})$, and the conditional MH-augmented *SVGD*-induced density is:

$$q^{\text{MH},l}(x^l | r^l = 1) = \alpha^l q^{\text{MH},l-1}(x^{l-1}) \left| \det\left(I + \epsilon \nabla_{x^{l-1}} \phi(x^{l-1})\right) \right|^{-1} + (1 - \alpha^l) q^{\text{MH},l-1}(x^{l-1}).$$

However, when $r = -1$, we run the *SVGD* chain backwards. Therefore, the proposal is $\tilde{x}^l = f^{-1}(x^{l-1})$, and the conditional MH-augmented *SVGD*-induced density is:

$$q^{\text{MH},l}(x^l | r^l = -1) = \alpha^l q^{\text{MH},l-1}(x^{l-1}) \left| \det \nabla_{x^{l-1}} f^{-1}(x^{l-1}) \right|^{-1} + (1 - \alpha^l) q^{\text{MH},l-1}(x^{l-1})$$

$$\overset{(i)}{=} \alpha^l q^{\text{MH},l-1}(x^{l-1}) \left| \det\left(I + \epsilon \nabla_{x^{l-1}} \phi(x^{l-1})\right) \right| + (1 - \alpha^l) q^{\text{MH},l-1}(x^{l-1}).$$

$(i)$  by the fact that the jacobian of the inverse $f^{-1}$ is the inverse of the jacobian of $f$

We combine both expressions as:

$$q^{\text{MH},l}(x^l | r^l) = \alpha^l q^{\text{MH},l-1}(x^{l-1}) \left| \det\left(I + \epsilon \nabla_{x^{l-1}} \phi(x^{l-1})\right) \right|^{-r^l} + (1 - \alpha^l) q^{\text{MH},l-1}(x^{l-1}).$$

The reversibility guarantees from this construction depend on the acceptance probability $\alpha^l$ satisfying:

$$\alpha^l(x^{l-1}, r^l) p(x^{l-1}) p_r(r^l) = \alpha^l(G(x^{l-1}, r^l)) p(\tilde{x}^l) p_r(-r^l) | \det\left(I + \epsilon \nabla_{x^{l-1}} \phi(x^{l-1})\right) |,$$

where $\tilde{x}^l = f^{r^l}(x^{l-1})$ is the proposal. This is the case when $\alpha^l$ is defined to be the MH ratio:

$$\alpha^l = \min\left(1, \frac{\bar{p}(\tilde{x}^l) p_r(-r^l)}{\bar{p}(x^{l-1}) p_r(r^l)} \left| \det \nabla_{x^{l-1}} f^{r^l}(x^{l-1}) \right| \right).$$

Since we chose $r^l$ to be a Rademacher random variable for which $p_r(r^l) = p_r(r^{l-1}) = \frac{1}{2}$, we obtain:

$$\alpha^l = \min\left(1, \frac{\bar{p}(\tilde{x}^l)}{\bar{p}(x^{l-1})}\left| \det \nabla_{x^{l-1}} f^{r^l}(x^{l-1})\right|\right). \tag{20}$$

### G.2  ACCEPTANCE PROBABILITY (PROPOSITION 3.5)

**Proposition 3.5** Let $p(x) = \bar{p}(x)/Z$ be a strictly positive and continuously differentiable density on $\mathcal{X} \subset \mathbb{R}^d$, $\phi(x)$ the *SVGD* velocity field with an RBF kernel from Eq. 1 satisfying the step-size bound in Corr. 3.3 and $r^l \in \{-1, 1\}$ the Rademacher auxiliary variable selecting the forward ($r^l = 1$) or backward ($r^l = -1$) *SVGD* update in *MET-SVGD*. The log-likelihood of the MH acceptance probability for a *MET-SVGD* update of a particle $x^{l-1}$ is:

$$\log \alpha^l = \min\left[0, \log \bar{p}(\tilde{x}^l) - \log \bar{p}(x^{l-1}) + r^l \epsilon^l \mathrm{Tr}(\nabla_{x^{l-1}} \phi(x^{l-1})) + \mathcal{O}\left((\epsilon^l/\epsilon_{\mathrm{UB}}^l)^2\right)\right].$$

*Proof.* Apply log to the MH ratio in Eq. 20:

$$\log\left(\frac{\bar{p}(\tilde{x}^l)}{\bar{p}(x^{l-1})}|\det \nabla_{x^{l-1}} f^{r^l}(x^{l-1})|\right) \overset{(i)}{=} \log\left(\frac{\bar{p}(\tilde{x}^l)}{\bar{p}(x^{l-1})}\right) + r^l|\det \nabla_{x^{l-1}} f(x^{l-1})|$$

$$= \log\left(\frac{\bar{p}(\tilde{x}^l)}{\bar{p}(x^{l-1})}\right) + r^l \log\left|\det\left(I + \epsilon^l \nabla_{x^{l-1}}\phi(x^{l-1})\right)\right|$$

    ($i$) by the fact that the jacobian of the inverse $f^{-1}$ is the inverse of the jacobian of $f$ (for $r^l = -1$)

Using the first-order approximation $\log\left|\det(I + \epsilon^l \nabla_{x^{l-1}}\phi(x^{l-1}))\right| = \epsilon\,\mathrm{Tr}(\nabla_{x^{l-1}}\phi(x^{l-1})) + \mathcal{O}\left((\epsilon^l/\epsilon_{\mathrm{UB}}^l)^2\right)$ from Eq. 18, we obtain:

$$\log\left(\frac{\bar{p}(\tilde{x}^l)}{\bar{p}(x^{l-1})}\right) + r^l \epsilon^l \mathrm{Tr}(\nabla_{x^{l-1}}\phi(x^{l-1})) + \mathcal{O}\left((\epsilon^l/\epsilon_{\mathrm{UB}}^l)^2\right)$$

The error term $\mathcal{O}\left((\epsilon^l/\epsilon_{\mathrm{UB}}^l)^2\right)$ is explained in Sec. C.1. $\qquad\square$

### G.3  MH-AUGMENTED SVGD DENSITY (PROPOSITION 3.6)

**Proposition 3.6** Under the setup of Prop. 3.5, the MH-augmented density at step $l$ is computed as: $q_\theta^{\mathrm{MH},l}(x^l) = \frac{1}{2}q_\theta^{\mathrm{MH},l}(x^l|r^l = 1) + \frac{1}{2}q_\theta^{\mathrm{MH},l}(x^l|r^l = -1)$, where for $c \in \{-1, 1\}$:

$$q_\theta^{\mathrm{MH},l}(x^l|r^l = c) = \alpha_\theta^l q_\theta^{\mathrm{MH},l-1}(x^{l-1})\left|\det\left(I + \epsilon_{\theta_3}\nabla_{x^{l-1}}\phi_\theta(x^{l-1})\right)\right|^{-c} + (1 - \alpha_\theta^l)q_\theta^{\mathrm{MH},l-1}(x^{l-1}),$$

*Proof.* Simply marginalize over $r^l$:

$$q_\theta^{\mathrm{MH},l}(x^l) \overset{(i)}{=} \sum_{c\in\{-1,1\}} p_r(r^l = c)q_\theta^{\mathrm{MH},l}(x^l|r^l = c)$$

$$\overset{(ii)}{=} \sum_{c\in\{-1,1\}} \frac{1}{2}q_\theta^{\mathrm{MH},l}(x^l|r^l = c),$$

    ($i$) $p_r$ is the Rademacher distribution

    ($ii$) all outcomes under $p_r$ are equally likely, hence $p_r(r^l = c) = \frac{1}{2}$

$\qquad\square$

# H    ADDITIONAL RESULTS ON ENTROPY ESTIMATION

In this section, we provide implementation details for the toy experiments as well as some additional experiments.

We adopt the following conventions:

- We write $\sigma = \sigma_{\theta_2}^l$ and $\epsilon = \epsilon_{\theta_3}^l$ to indicate that we are learning a free parameter per step. For example, in Pytorch, this is achieved by instantiating a `nn.Parameter` object.

- We write $\sigma = \text{GNN}(\{x_i^l\}_{i=0}^{M-1}; h, \theta_2)$ to indicate that $\sigma$ is parameterized by a graph neural network with the layout described in Tab. 4.

- We write $\epsilon = \min(\epsilon_{\theta_3}^0, \epsilon_{\theta_3}^0 d_{\theta_3}^{l/s_{\theta_3}})$ to indicate that the step-size is defined as an exponentially decaying function of the step $l$, controlled by the initial value $\epsilon_{\theta_3}^0$, the decay factor $d_{\theta_3}$, and the scaling constant $s_{\theta_3}$, all of which being free parameters.

| Step | Operation | Description |
|------|-----------|-------------|
| 1 | Linear layer (output $h$) | Embedding layer: $x_i^l \rightarrow W_1 x_i^l$ |
| 2 | Mean aggregation | Aggregate node embeddings: $\bar{x}^l = \frac{1}{M} \sum_{i=0}^{M-1} W_1 x_i^l$ |
| 3 | ReLU | Apply nonlinearity |
| 4 | Linear layer (output $h$) | Transform aggregated embedding |
| 5 | ReLU | Apply second nonlinearity |
| 6 | Linear layer (output 1) | Map to scalar |
| 7 | $\exp(\cdot)$ | Ensure positivity |

Table 4: Architecture of the graph neural network denoted by $\text{GNN}(\{x_i^l\}_{i=0}^{M-1}; h, \theta_2)$.

## H.1    GAUSSIAN TARGETS

### H.1.1    FIG. 2A, LIMITATION 1: P-SVGD'S SENSITIVITY TO RBF KERNEL BANDWIDTH

**Implementation Details** are reported in Tab. 5.

| Algorithm | Parameter | Value |
|-----------|-----------|-------|
| *LD / P-SVGD / MET-SVGD* | Target $p$ | $p = \mathcal{N}([-0.69, 0.8], [[1.13, 0.82], [0.82, 3.39]])$ |
| | Number of Particles $M$ | 200 |
| | Number of Steps $L$ | 1500 |
| | Initial Distribution $q^0$ | $\mathcal{N}(0, 6I)$ |
| *LD / P-SVGD* | Step-Size | $\epsilon = 0.1$ |
| *P-SVGD* | Kernel Bandwidth | $\sigma \in \{1, 5, \sigma_{\text{med}}\}$ |
| *MET-SVGD* | Step-Size | $\epsilon = \min(\epsilon_{\theta_3}^0, \epsilon_{\theta_3}^0 d_{\theta_3}^{l/s_{\theta_3}})$ |
| | Kernel Bandwidth | $\sigma = \text{GNN}(\{x_i^l\}_{i=0}^{M-1}; 128, \theta_2)$ (Tab. 4) |
| | **Training Parameters** | |
| | Optimizer | Adam |
| | Learning Rate | $5 \cdot 10^{-3}$ |
| | Epochs | 300 |
| | Loss | KL Divergence |
| Resources | GPU | Tesla V100-SXM2-32GB |
| | RAM | 2 GB |
| | Per-epoch Runtime | 2.6s |

Table 5: Implementation details for the setup in Fig. 2a.

**Ablation Study** is reported in Fig. 15. Learning the kernel bandwidth and the step-size is what led to the largest improvements.

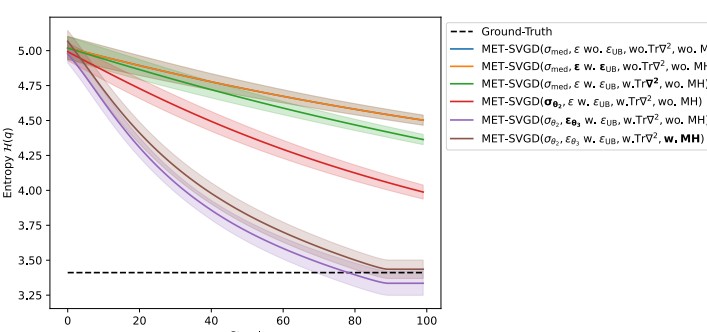

Figure 15: Ablation study for the setup in Fig. 2a. Note that the first two configurations (w. and wo. $\epsilon_{\mathrm{UB}}$) are overlapping as the selected $\epsilon$ satisfies the upper bound constraint. The configuration with MH yields the best results.

### H.1.2 FIG. 2B, LIMITATION 2: P-SVGD'S POOR CONVERGENCE TO NON-SMOOTH TARGETS

**Implementation Details** are reported in Tab. 6.

| Algorithm | Parameter | Value/Design |
|---|---|---|
| | Target $p$ | Smooth distribution: $\mathcal{N}(0, I_2)$ |
| | | Non-smooth distribution: |
| | | $\mathcal{N}(0, \Sigma)$ with $\Sigma = \begin{pmatrix} 0.505 & 0.495 \\ 0.495 & 0.505 \end{pmatrix}$ |
| | Number of Particles $M$ | 100 |
| | Number of Steps $L$ | 110 |
| | Initial Distribution $q^0$ | $\mathcal{N}(0, \sqrt{3}I)$ |
| *P-SVGD* | Kernel Bandwidth | $\sigma = \sigma_{\mathrm{med}}$ |
| | Step-Size | $\epsilon = 0.1$ |
| *MET-SVGD* | Kernel Bandwidth | $\sigma = \sigma_{\theta_2}^l$ |
| | Step-Size | $\epsilon = \min(\epsilon_{\theta_3}^0, \epsilon_{\theta_3}^0 d_{\theta_3}^{l/s_{\theta_3}})$ |
| Training Parameters | Optimizer | Adam |
| | Learning Rate | $10^{-3}$ |
| | Epochs | 200 |

Table 6: Implementation details for the setup in Fig. 2b.

**Ablation Study** is reported in in Fig. 16. Though learning the kernel bandwidth and the step-size helped, the best result was obtained by incorporating MH.

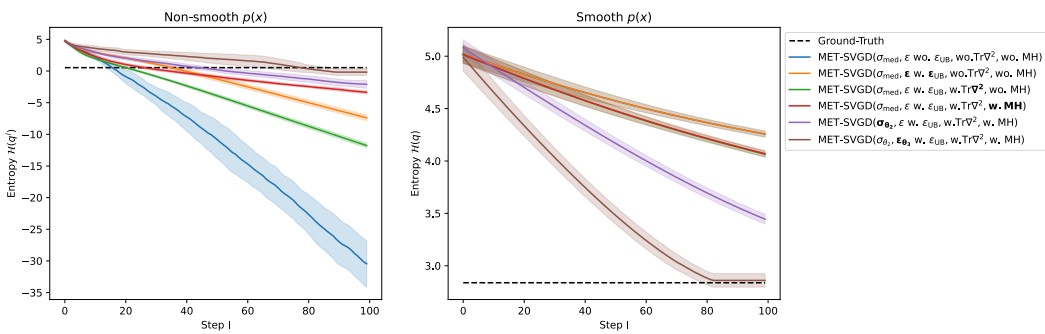

Figure 16: Ablation study for the setup in Fig. 2b. [Left] Non-smooth target. [Right] Smooth target. For the non-smooth target, MH significantly helps convergence to the target.

### H.1.3 FIG. 2D, LIMITATION 4: P-SVGD'S POOR SCALABILITY TO HIGH DIMENSIONAL SPACES

**Implementation Details** are reported in Tab. 7.

| Parameter | Value |
|---|---|
| Target distribution | $p = \mathcal{N}(0, I_d), \; d \in \{100, 200, \ldots, 1000\}$ |
| Initial distribution | $\mathcal{N}(2\mathbb{1}, 2I_d)$ |
| **SVGD Parameters** | |
| Number of Particles | $M = 100$ |
| Number of Steps | $L = 100$ |
| Kernel Bandwidth | $LD: \sigma = 0$ |
| | $P\text{-}SVGD: \sigma = \sigma_{\text{med}}$ |
| | $MET\text{-}SVGD: \sigma = \sigma_{\theta_2}^l$ |
| Step-Size | $LD: \epsilon = \epsilon_{\theta_3}^l$ |
| | $P\text{-}SVGD: \epsilon = 0.1$ |
| | $MET\text{-}SVGD: \epsilon = \epsilon_{\theta_3}^l$ |
| **Training Parameters** | |
| Optimizer | Adam |
| Learning rate | $10^{-2}$ |
| Epochs | 500 |

Table 7: Implementation details for the setup in Fig. 2d.

In Fig. 17, we visualize the learnt *SVGD* step-size for the setup in Fig. 2d. The target is a $d$-dimensional multivariate Gaussian $p(x) = \mathcal{N}(x; 0, I_d)$. For each method, 100 particles are initialized from $\mathcal{N}(x; 2\mathbb{1}, 2I_d)$, where $\mathbb{1} \in \mathbb{R}^d$ denotes the vector of ones. This is a standard benchmark illustrating the diminishing variance issue of *SVGD*. We show that *MET-SVGD* outperforms other baselines in high dimensional spaces as measured by the entropy and the variance across dimensions (Fig. 17.a and Fig. 17.b). The *SVGD* repulsive force is large during the first updates, preventing the particles from collapsing, unlike other baselines (Fig. 17.c). *P-SVGD* is not scalable due to a missing term in its entropy formula, which is subsequently fixed in the *MET-SVGD* update (Sec. 3.3).

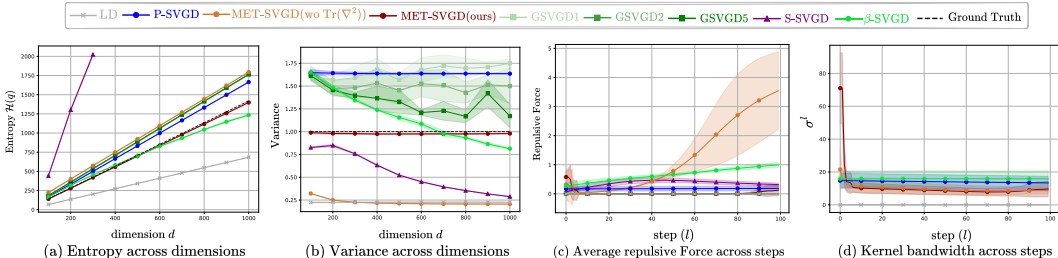

(a) Entropy across dimensions    (b) Variance across dimensions    (c) Average repulsive Force across steps    (d) Kernel bandwidth across steps

Figure 17: *MET-SVGD* achieves higher accuracy on both (a) entropy estimation and (b) marginal variance than baselines across dimensions, which is explained by the difference in (c) the trend of the repulsive force across steps, and driven by different (d) kernel bandwidth trends.

**Ablation Study** is reported in in Fig. 18. Adding the trace term is what led to the largest improvements.

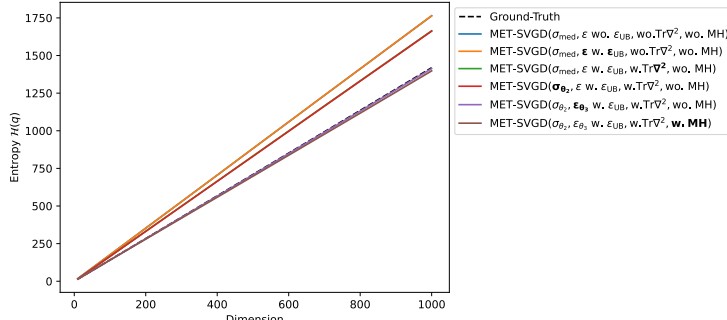

Figure 18: Ablation study for the setup in Fig. 2d. Note that the first two configs (w. and wo. $\epsilon_{\text{UB}}$) are overlapping as the selected $\epsilon$ satisfies the upper bound constraint. Also, since the distribution is smooth, MH didn't lead to significant improvements.

### H.1.4 FIG. 13: BILINEAR KERNEL

**Implementation Details** are reported in Tab. 8.

| Parameter | Figure 13 |
|---|---|
| Number of Steps $L$ | 1000 |
| **Kernel** | |
| Architecture | $C_{\theta_2} = \text{GNN}(\{x_i^l\}_{i=0}^{M-1}; 128, \theta_2)$ (Tab. 4) |
| Kernel Type | Bilinear: $\frac{x_i^T x_j}{C_{\theta_2}^2} + 1$ |
| **Step-Size** | |
| Architecture | $\epsilon = \min\left(\epsilon_{\theta_3}^0, \epsilon_{\theta_3}^0 d_{\theta_3}^{l/s_{\theta_3}}\right)$ |
| **Training Parameters** | |
| Optimizer | Adam |
| Learning Rate | $5 \cdot 10^{-3}$ |
| Epochs | 300 |

Table 8: Implementation details for the setup in Fig. 13

### H.1.5 ACCELERATING CONVERGENCE

In Fig. 19, we show that, for the setup of Fig. 14, convergence can be accelerated either by (1) adding a regularization on the decay rate in the optimization objective (Sec. 3.2) or (2) randomizing the maximum number of steps during training (Eq. 5).

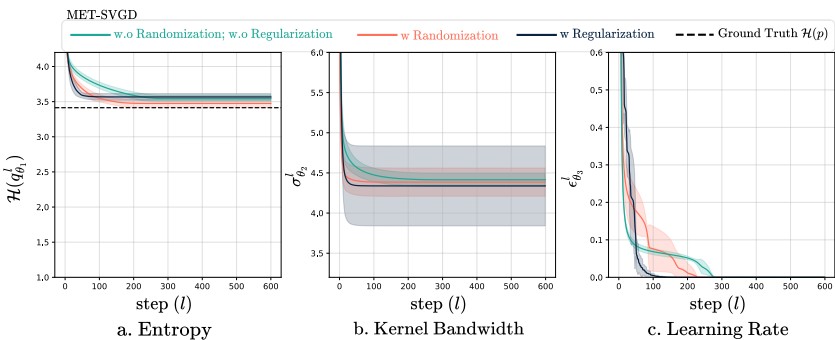

Figure 19: Accelerated convergence via regularization and number of *SVGD* steps randomization for the same *MET-SVGD* setup in Tab. 7

### H.2 GAUSSIAN MIXTURE MODELS

The density of GMM with $K$ components is given by:

$$p(x) = \sum_{i=1}^{K} \omega_i \mathcal{N}(x; \mu_i, C_i),$$

where $\{\omega_i\}_{i=1}^K$ are non-negative weighting coefficients such that $\sum_i \omega_i = 1$, and $\mathcal{N}(x; \mu_i, C_i)$ is a gaussian density with mean $\mu_i$ and covariance $C_i$. Note that the entropy generally cannot be calculated in closed form for GMM due to the logarithm of a sum of exponential functions (except for the special case of a single Gaussian density). We derive two bounds to assess the performance of the different baselines.

- We start by deriving the **lower bound**, *i.e.*, we show that:

$$\mathcal{H}(p) \geq -\sum_{i=1}^{K} \omega_i \log\left[\sum_{j=1}^{K} \omega_j \mathcal{N}(\mu_i; \mu_j, C_i + C_j)\right]$$

*Proof.*

$$\mathcal{H}(p) = -\sum_{i=1}^{K} \omega_i \int_{\mathbb{R}^N} \mathcal{N}(x; \mu_i, C_i) \log p(x) dx$$

$$\geq -\sum_{i=1}^{K} \omega_i \log \Big[ \int_{\mathbb{R}^N} \mathcal{N}(x; \mu_i, C_i) p(x) dx \Big] \quad \text{by Jensen's inequality}$$

$$\geq -\sum_{i=1}^{K} \omega_i \log \Big[ \int_{\mathbb{R}^N} \mathcal{N}(x; \mu_i, C_i) \sum_{j=1}^{K} \omega_j \mathcal{N}(x; \mu_j, C_j) dx \Big]$$

$$\geq -\sum_{i=1}^{K} \omega_i \log \Big[ \sum_{j=1}^{K} \omega_j \mathcal{N}(\mu_i; \mu_j, C_i + C_j) \Big]$$

$\square$

- Next, we derive the **upper bound**, *i.e.*, we show that

$$\mathcal{H}(p) \leq \mathcal{H}(\mathcal{N}(\mu_{\mathrm{G}}, C_{\mathrm{G}})), \quad \text{with} \begin{cases} \mu_{\mathrm{G}} = \sum_{i=1}^{L} \omega_i \mu_i \\ C_{\mathrm{G}} = \sum_{i=1}^{L} \omega_i (C_i + \mu_i \mu_i^T) - \sum_{i=1}^{L} \sum_{j=1}^{L} \omega_i \omega_j \mu_i \mu_j^T \end{cases}$$

*Proof.* Consider a Gaussian mixture model $p(x) = \sum_{i=1}^{L} \omega_i \mathcal{N}(x; \mu_i, C_i)$. The mean is $\mu = \sum_{i=1}^{L} \omega_i \mu_i$. The covariance can be computed as:

$$C = \mathbb{E}[(x - \mu)(x - \mu)^T] = \mathbb{E}[xx^T] - \mu\mu^T = \int_x \left( \sum_{i=1}^{L} \omega_i \mathcal{N}(x; \mu_i, C_i) \right) xx^T dx - \mu\mu^T$$

$$= \sum_{i=1}^{L} \omega_i \int_x xx^T \mathcal{N}(x; \mu_i, C_i) dx - \mu\mu^T$$

For a single Gaussian component, $C_i = \mathbb{E}[(x - \mu_i)(x - \mu_i)^T] = \mathbb{E}[xx^T] - \mu_i \mu_i^T$, which means $\mathbb{E}_{x \sim \mathcal{N}(x; \mu_i, C_i)}[xx^T] = C_i + \mu_i \mu_i^T$. Substituting this in the above:

$$C = \sum_{i=1}^{L} \omega_i (C_i + \mu_i \mu_i^T) - \mu\mu^T$$

$$= \sum_{i=1}^{L} \omega_i (C_i + \mu_i \mu_i^T) - \left( \sum_{i=1}^{L} \omega_i \mu_i \right) \left( \sum_{j=1}^{L} \omega_j \mu_j^T \right) = \sum_{i=1}^{L} \omega_i (C_i + \mu_i \mu_i^T) - \sum_{i,j=1}^{L} \omega_i \omega_j \mu_i \mu_j^T = C_G$$

Since a Gaussian distribution maximizes entropy among all distributions with the same mean and covariance, we have $\mathcal{H}(p) \leq \mathcal{H}(\mathcal{N}(\mu_{\mathrm{G}}, C_{\mathrm{G}}))$. $\square$

### H.2.1 Fig. 2c, Limitation 3: P-SVGD Suffers from Mode Collapse

**Implementation Details** are reported in Tab. 9.

We show that the divergence control heuristic based on eliminating particles further than 3 standard deviations of the initial distribution mean exacerbates mode collapse, which is already an issue when using the reverse KL-divergence.

**Performance.** *P-SVGD* without steps, which is essentially traditional VI, collapses to one mode and is not able to recover the true entropy. *P-SVGD* with $L = 140$ steps only recovers two out of the three modes: the divergence control heuristic prevents particles from reaching the lower mode. *MET-SVGD* on the other hand recovers all three modes and accurately estimates the entropy of the target.

| Algorithm | Parameter | Value/Design |
|---|---|---|
| | Target $p$ | GMM with 3 components: $\mu_1 = (-4.0, 5.0)$, $\Sigma_1 = 1.5I_2$ $\mu_2 = (3.0, 3.0)$, $\Sigma_2 = I_2$ $\mu_3 = (-4.0, -15.0)$, $\Sigma_3 = 2I_2$ |
| | Number of Particles $M$ | 100 |
| | Number of Steps $L$ | 140 |
| | Initial Distribution $q^0$ | $\mathcal{N}(0, 4I)$ |
| P-SVGD | Kernel Bandwidth | $\sigma = \sigma_{\mathrm{med}}$ |
| | Step-Size | $\epsilon = 0.5$ |
| MET-SVGD | Kernel Bandwidth | $\sigma = \sigma_{\theta_2}^l$ |
| | Step-Size | $\epsilon = \epsilon_{\theta_3}^l$ |
| Training Parameters | Optimizer | Adam |
| | Learning Rate | $10^{-1}$ |
| | Epochs | 200 |

Table 9: Implementation details for the setup in Fig. 2c.

**Ablation Study** is reported in in Fig. 20.

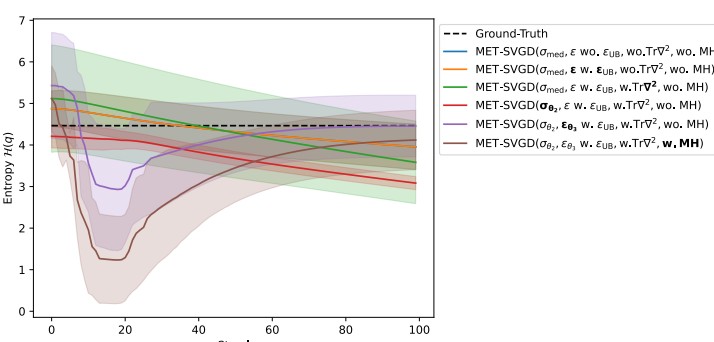

Figure 20: Ablation study for the setup in Fig. 2c.

## H.2.2 Fig. 6b: Experiment on 2D GMM with moving component

**Implementation Details** are reported in Tab. 10.

**Performance.** In Fig. 21, we show that *P-SVGD* has poor entropy estimation accuracy, even falling below a lower bound on the target entropy. This behavior can be explained by the fact that the optimal $\sigma_{\theta_2}$ exhibits a trend that deviates significantly from $\sigma_{\mathrm{med}}$, and that the optimal $\epsilon_{\theta_3}$ consistently converges to 0 across all *MET-SVGD* configurations, rather than remaining constant as in *P-SVGD* (Fig. 22).

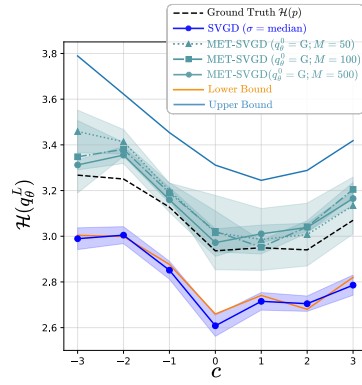

Figure 21: *MET-SVGD* significantly outperforms *P-SVGD*. Increasing the number of particles reduces the variances. Results are reported on 5 different seeds.

| Parameter | P-SVGD | MET-SVGD |
|---|---|---|
| Distribution type | | GMM with 5 components |
| Fixed components | | $\mu_1 = (0.0, 0.0),\ \Sigma_1 = 0.16 I_2$ |
| | | $\mu_2 = (3.0, 2.0),\ \Sigma_2 = I_2$ |
| | | $\mu_3 = (1.0, -0.5),\ \Sigma_3 = 0.5 I_2$ |
| | | $\mu_4 = (2.5, 1.5),\ \Sigma_4 = 0.5 I_2$ |
| Mobile component | | $\mu_5 = (c, c),\ \Sigma_5 = 0.5 I_2,\ c \in [-3, 3]$ |
| Dimension | | $d = 2$ |
| Number of particles | | $M \in \{50, 100, 500\}$ |
| Number of iterations | | $L = 1500$ |
| **Initial Distribution Settings** | | |
| Setting 1 | | $\mathcal{N}(0, I_2)$ |
| Setting 2 | | $\mathrm{GMM}(K = 10)$ with $\mu_k \sim \mathcal{U}([-4, 4]^2),\ \Sigma_k = I_2\ \forall k$ |
| **Algorithm-Specific Parameters** | | |
| Kernel bandwidth | $\sigma \in \{1, 5, \mathrm{median}\}$ | $\sigma = \mathrm{GNN}(\{x_i^l\}_{i=0}^{M-1}; 8, \theta_2)$ |
| Step-size | $\epsilon = 0.1$ | $\epsilon_{\theta_3}^l = \min(\epsilon_{\theta_3}^0, \epsilon_{\theta_3}^0 d^{l/s\theta_3})$ |
| **Training Parameters** | | |
| Optimizer | | Adam |
| Learning rate | | $5 \times 10^{-3}$ |
| Epochs | | 300 |

Table 10: Implementation details for the setup in Fig. 6b.

### H.3 HIGH DIMENSIONAL GMM

The goal of this experiment is to further assess the scalability of *MET-SVGD*.

The target distribution is a mixture of four D-dimensional Gaussian distributions $p(x) = \sum_{k=1}^4 0.25 \mathcal{N}(x, \mu_k, I_d)$ with uniform mixture ratios. The first two coordinates of the mean vectors are equally spaced on a circle, while the other coordinates are set to 0 (Fig. 23.A-D). Particles are initialized from $\mathcal{N}(0, I_d)$ and only the first two dimensions need to be learned.

**Implementation Details** are reported in Tab. 11.

| Parameter | Value |
|---|---|
| Target Distribution | $p(x) = \sum_{k=1}^4 0.25 \mathcal{N}(x, \mu_k, I_d)$ |
| Initial Distribution | $q^0 = \mathcal{N}(0, I)$ |
| **Default SVGD Parameters** | |
| Step-Size | P-SVGD: $\epsilon = 0.1$ |
| | MET-SVGD: $\epsilon = \min(\epsilon_{\theta_3}^0, \epsilon_{\theta_3}^0 d_{\theta_3}^{l/s\theta_3})$ |
| Number of Steps | $L = 100$ |
| Number of Particles | $M = 100$ |
| Kernel Bandwidth | P-SVGD: $\sigma = \sigma_{\mathrm{med}}$ |
| | MET-SVGD: $\sigma = \sigma_{\theta_2}^l$ |
| **Training Parameters** | |
| Optimizer | Adam |
| Learning Rate | $10^{-2}$ |
| Epochs | 500 |

Table 11: Implementation details for the setup in Fig. 23

**Performance.** In Fig. 23, we show that *MET-SVGD* significantly outperforms *P-SVGD* and other *SVGD* variants in terms of entropy estimation accuracy. Also, unlike *P-SVGD*, *MET-SVGD* is able to recover all four modes.

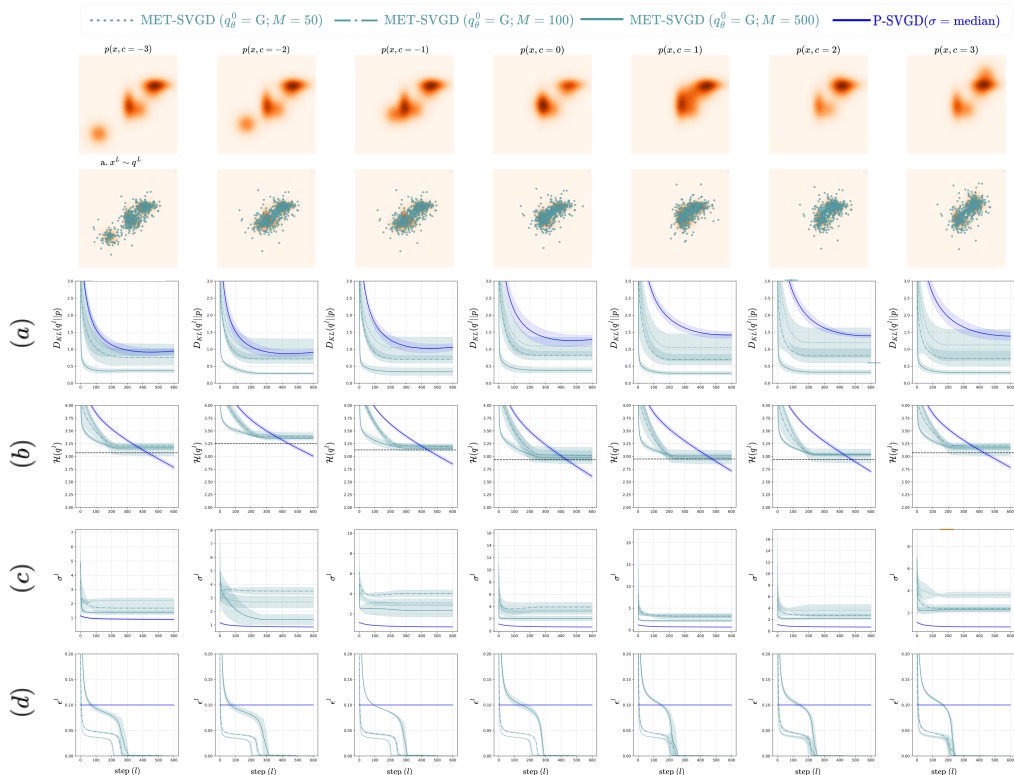

Figure 22: Results on entropy estimation for different $c$ configurations. *MET-SVGD* significantly outperforms *P-SVGD* in terms of (a) KLD minimization and (b) entropy estimation. This is explained by the fact that it is able to learn a more optimal (c) kernel bandwidth trend than the median heuristic, as well as a more optimal (d) step-size trend than a constant one.

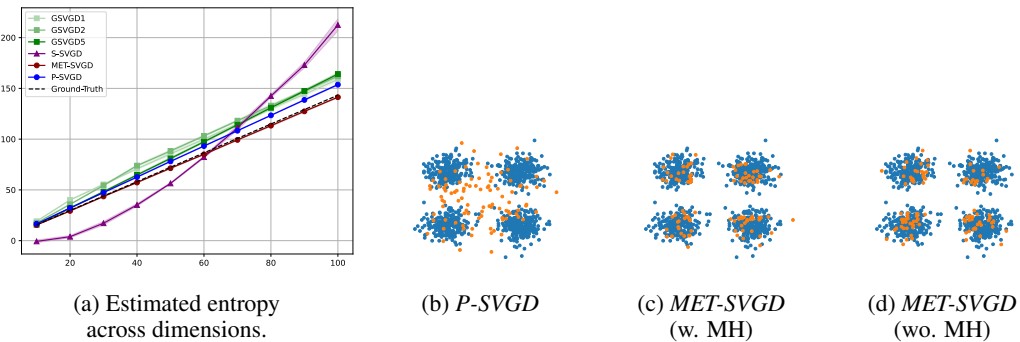

(a) Estimated entropy across dimensions.

(b) *P-SVGD*

(c) *MET-SVGD* (w. MH)

(d) *MET-SVGD* (wo. MH)

Figure 23: Scalability results. Target is a high-dimensional GMM. *MET-SVGD* successfully recovers the entropy of the low-dimensional GMM, as well as all four modes, unlike *P-SVGD*.

## I    ADDITIONAL RESULTS: ENERGY BASED MODELS

**Proposition** (Sec. 4.2). *Training EBMs* $p_\phi(x) = \bar{p}_\phi(x)/Z_\phi$ *via maximum likelihood* $(\mathcal{L}_{ebm}(\phi) = -\mathbb{E}_{x \sim p_d}[\log p_\phi(x)])$ *is intractable due to the partition function* $Z_\phi$. *When the sampler has a tractable distribution* $q_\theta$, *a tight lower bound can be computed as:* $\mathcal{L}_{ELBO}(\phi, \theta) = \mathbb{E}_{x \sim q_\theta}[\log \bar{p}_\phi(x)] - \mathbb{E}_{x \sim p_d}[\log \bar{p}_\phi(x)] + \mathcal{H}(q_\theta)$ *with* $p_d$ *being the data distribution,*

*Proof.* Given:

$$\mathcal{L}_{\text{ebm}}(\phi) = -\mathbb{E}_{x \sim p_d}[\log p_\phi(x)] = -\mathbb{E}_{x \sim p_d}[\log \bar{p}_\phi(x)] + \log Z(\phi).$$

We bound the partition function using the KL-divergence:

$$\log Z(\phi) \geq \log Z(\phi) - D_{\text{KL}}(q_\theta(x) \| p_\phi(x)) \geq \log Z(\phi) + \int_x q_\theta(x) \log \frac{p_\phi(x)}{q_\theta(x)} dx$$

$$\geq \log Z(\phi) + \int_x q_\theta(x) \log \frac{\frac{\bar{p}_\phi(x)}{Z(\phi)}}{q_\theta(x)} dx$$

$$\geq \log Z(\phi) + \int_x q_\theta(x) \log \bar{p}_\phi(x) dx - \int_x q_\theta(x) \log Z(\phi) dx - \int_x q_\theta(x) \log q_\theta(x) dx$$

$$\geq \log Z(\phi) + \mathbb{E}_{x \sim q_\theta}[\log \bar{p}_\phi(x)] - \log Z(\phi) + \mathcal{H}(q_\theta) \geq \mathbb{E}_{x \sim q_\theta}[\log \bar{p}_\phi(x)] + \mathcal{H}(q_\theta).$$

Substituting back into the MLE objective:

$$\mathcal{L}_{\text{ebm}}(\phi) = -\mathbb{E}_{x \sim p_d}[\log \bar{p}_\phi(x)] + \log Z(\phi)$$
$$\geq -\mathbb{E}_{x \sim p_d}[\log \bar{p}_\phi(x)] + \mathbb{E}_{x \sim q_\theta}[\log \bar{p}_\phi(x)] + \mathcal{H}(q_\theta) = \mathcal{L}_{\text{ELBO}}(\phi, \theta).$$

$\square$

## I.1 SYNTHETIC EXPERIMENT: MOON DISTRIBUTION

We evaluate *MET-SVGD* on the Moon dataset (Rezende & Mohamed, 2015).

**Implementation Details** are reported in Tab. 12. We write $\sigma = \sigma_{\theta_2}^l$ to indicate that we are learning a free parameter per step. For example, in Pytorch, this is achieved by instantiating a `nn.Parameter` object. Moreover, we write $\epsilon = \min(\epsilon_{\theta_3}^0, \epsilon_{\theta_3}^0 d_{\theta_3}^{l/s_{\theta_3}})$ to indicate that the step-size is defined as an exponentially decaying function of the step $l$, controlled by the initial value $\epsilon_{\theta_3}^0$, the decay factor $d_{\theta_3}$, and the scaling constant $s_{\theta_3}$, all of which being free parameters.

| Parameter | Value |
|---|---|
| Target Distribution | $p_\phi(x) = \frac{\exp f_\phi(x)}{Z_\phi}$ |
| | $f_\phi(x) = \text{MLP}_\phi(128, \text{Swish}, 128, \text{Swish}, 128, \text{Swish}, 1)$ |
| Initial Distribution | $q^0 = \mathcal{N}([0,0], 7I)$ |
| **Default SVGD Parameters** | |
| Step-Size | $\epsilon = \min(\epsilon_{\theta_3}^0, \epsilon_{\theta_3}^0 d_{\theta_3}^{l/s_{\theta_3}})$ |
| Number of Steps | $L = 100$ |
| Number of Particles | $m = 129$ |
| Kernel Bandwidth | $\sigma = \sigma_{\theta_2}^l$ |
| **Training Parameters** | |
| Optimizer | Adam |
| $\phi$ Learning rate | $10^{-3}$ |
| $\theta$ Learning rate | $10^{-2}$ |
| Iterations | 1250 |
| **Resources** | |
| GPU | Tesla V100-SXM2-32GB |
| RAM | 2 GB |
| Per-epoch runtime | 2.6 seconds |

Table 12: Implementation details for the setup in Fig. 24

**Performance.** In Fig. 24, we report the Maximum Mean Discrepancy (MMD) (Gretton et al., 2012) and present samples generated by *MET-SVGD* trained with and without an MH correction on the

Moons dataset under varying smoothness conditions. The configuration with MH provides substantial benefits in the least smooth regime (where the target distribution resembles three nearly piecewise-linear clusters), yielding the lowest MMD scores and enabling the sampler to learn a highly non-smooth energy landscape that accurately reflects the underlying data distribution. Additionally, in Fig. 25, we compare our method against Masked Autoregressive Flows (Papamakarios et al., 2017). In contrast to *MET-SVGD*, the normalizing flow fails to approximate the target distribution.

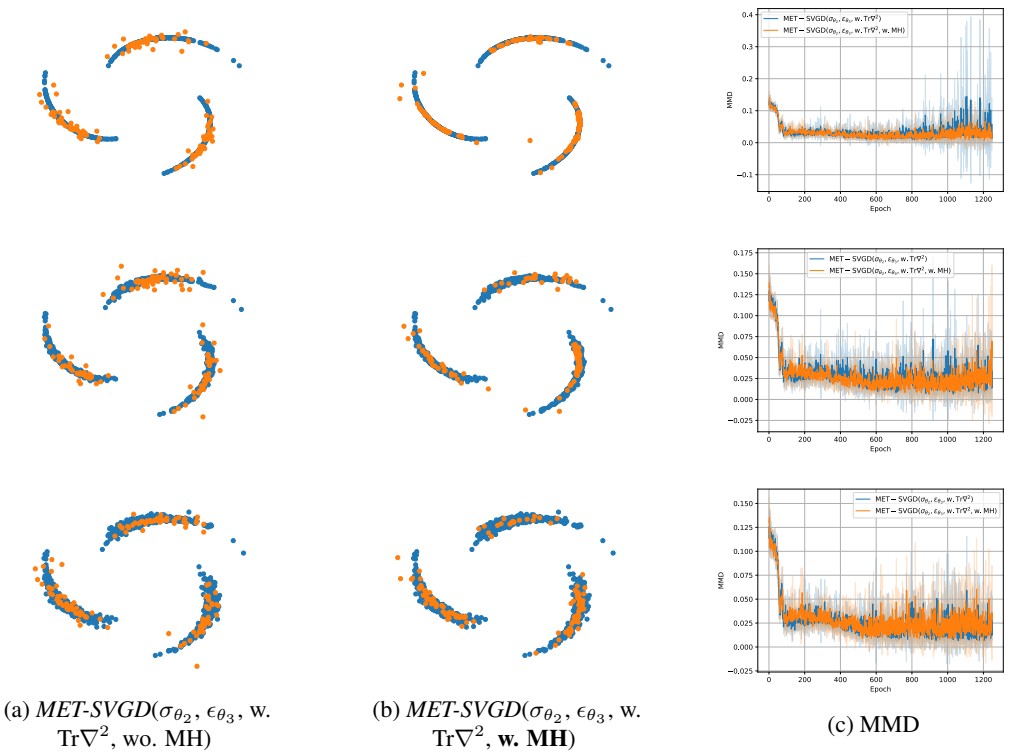

(a) *MET-SVGD*($\sigma_{\theta_2}$, $\epsilon_{\theta_3}$, w. Tr$\nabla^2$, wo. MH)

(b) *MET-SVGD*($\sigma_{\theta_2}$, $\epsilon_{\theta_3}$, w. Tr$\nabla^2$, **w. MH**)

(c) MMD

Figure 24: Samples and MMD score on the Moons dataset with varying smoothness degrees.

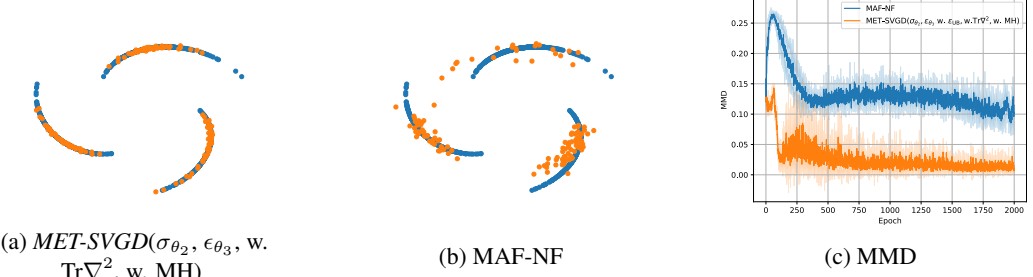

(a) *MET-SVGD*($\sigma_{\theta_2}$, $\epsilon_{\theta_3}$, w. Tr$\nabla^2$, w. MH)

(b) MAF-NF

(c) MMD

Figure 25: *MET-SVGD* and MAF-NF Samples and MMD score on the non-smooth version of the Moons dataset.

## I.2   FIG. 7: IMAGE GENERATION

**Implementation Details** are reported in Tab. 13.

All experiments were conducted on a single NVIDIA A100 80GB with 8GB allocated RAM; average runtime was around 6 seconds per iteration (processing one batch of data composed of 64 particles).

| Parameter | Value |
|---|---|
| Target distribution | $p_\theta(x) = \frac{\exp f_\theta(x)}{Z_\theta}$ |
| | $f_\theta(x)$ is a WideResnet(28,10) network |
| Initial distribution | $q^0$ is a replay buffer initialized using a GMM whose modes are based on the class-conditional means and covariances |
| **Default SVGD Parameters** | |
| Step-Size | *LD* and *P-SVGD*: $\epsilon = 64$ (divided by $M$ in the *SVGD* update formula) |
| | *MET-SVGD*: $\epsilon = \text{GNN}(\{x_i^l\}, \{\nabla_{x_i^l} \log p_\theta(x_i^l)\}; 1024, \theta_3)$ (Tab. **??**) |
| Number of steps | *LD*, *P-SVGD*, and *MET-SVGD* wo. $L_c$: $L = 5$ |
| Number of particles | $M = 64$ |
| Kernel bandwidth | *LD*: $\sigma = 0$ |
| | *P-SVGD*: $\sigma = \sigma_{\text{med}}$ |
| | *MET-SVGD*: $\sigma = \text{GNN}(\{x_i^l\}; 1024, \theta_2)$ (Tab. 14) |
| **Training Parameters** | |
| Optimizer | SGD |
| $\theta$ Learning rate | $10^{-1}$: 1000 iterations warm-up and decay at epochs 60, 120, and 180, with 0.2 decay rate |
| $\phi$ Learning rate | $10^{-4}$ |
| Epochs | 200 |
| **Resources** | |
| GPU | NVIDIA A100 80GB |
| RAM | 8 GB |
| Per-iteration runtime | 6 seconds |

Table 13: Implementation details for the setup in Fig. 7.

| Step | Operation | Description |
|---|---|---|
| 1 | Linear layer (output $h$) | Embedding layer: $x_i^l \rightarrow W_1 x_i^l$ |
| 2 | ReLU | Apply nonlinearity |
| 3 | Mean aggregation | Aggregate node embeddings: $\bar{x}^l = \frac{1}{M} \sum_{i=0}^{M-1} \text{ReLU}(W_1 x_i^l)$ |
| 4 | Linear layer (output $h$) | Transform aggregated embedding |
| 5 | ReLU | Apply second nonlinearity |
| 6 | Linear layer (output 1) | Map to scalar |
| 7 | $\exp(\cdot)$ | Ensure positivity |

Table 14: Architecture of $\text{GNN}(\{x_i^l\}_{i=0}^{M-1}; h, \theta_2)$ for the setup in Fig. 7.

We distinguish between two training procedures:

- *LD* is trained with contrastive divergence

$$\min_\phi \mathcal{L}_{\text{CD}}(\phi) = \min_\phi -\mathbb{E}_{x \sim p_d}[f_\phi(x)] + \mathbb{E}_{x \sim q}[f_\phi(x)] \tag{21}$$

- *P-SVGD* and *MET-SVGD* are trained adversarially by alternating between learning the sampler's parameters (Eq. 22) and minimizing the contrastive divergence (Eq. 21).

$$\min_\theta -\mathcal{L}_{\text{ELBO}}(\theta) = \max_\theta -\mathbb{E}_{x \sim p_d}[f_\phi(x)] + \mathbb{E}_{x \sim q_\theta}[f_\phi(x)] + \mathcal{H}(q_\theta), \tag{22}$$

**Performance.** We report the FID and Inception scores in Fig. 26. The best FID (lowest) is achieved when both the kernel bandwidth and the step size are learned. Comparable performance with improved scalability is obtained when using an adaptive number of steps $Lc$. In contrast, removing the trace term causes premature divergence, highlighting the critical role of this correction. Both *LD* and *P-SVGD* exhibit early divergence. Moreover, due to high rejection rates, the MH setup also diverges (Fig. 28). In Fig. 29, we show that the trends of the learned optimal kernel bandwidth and step-size are different from those of *P-SVGD*, which explains the discrepancy in performance. Fig. 27 shows that the resulting performs gains from learning the kernel bandwidth and the step-size are explained by the fact that the optimal value for these parameters induces smoother energy landscapes, which facilitates sampling from the target.

| Step | Operation | Description |
|------|-----------|-------------|
| 1.a | Linear layer followed by ReLU (output $h$) | First particle embedding layer: $x_i^l \rightarrow \text{ReLU}(W_1 x_i^l)$ |
| 1.b | Linear layer followed by ReLU (output $h$) | First score embedding layer: $\nabla_{x_i^l} \log p(x_i^l) \rightarrow \text{ReLU}(\tilde{W}_1 \nabla_{x_i^l} \log p(x_i^l))$ |
| 2.a | Linear layer (output $h$) | Second particle embedding layer |
| 2.b | Linear layer (output $h$) | Second score embedding layer |
| 3 | Embedding aggregation layer followed by ReLU (output $h$) | Sum particle and score embeddings |
| 4 | Mean aggregation (output $h$) | Average embeddings over the batch dimension |
| 5 | Linear layer followed by ReLU (output $h$) | |
| 6 | Linear layer followed by ReLU (output $h$) | |
| 7 | Linear layer (output 1) | Map to scalar |
| 8 | $\exp(\cdot)$ | Ensure positivity |

Table 15: Architecture of $\text{GNN}(\{x_i^l\}, \{\nabla_{x_i^l} \log p_\theta(x_i^l)\}; h, \theta_3)$ for the setup in Fig. 7.

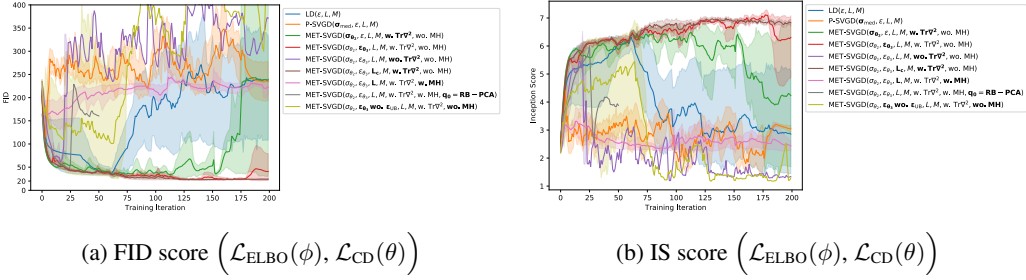

(a) FID score $\left( \mathcal{L}_{\text{ELBO}}(\phi), \mathcal{L}_{\text{CD}}(\theta) \right)$  (b) IS score $\left( \mathcal{L}_{\text{ELBO}}(\phi), \mathcal{L}_{\text{CD}}(\theta) \right)$

Figure 26: EBM Results. We report the FID and IS scores across training iterations. $\text{Tr} \nabla^2$ stands for including the trace of Hessian term and $q_0 = \text{RB-PCA}$ for initializing the replay buffer with samples obtained via a linear combination of the principal components of the data samples.

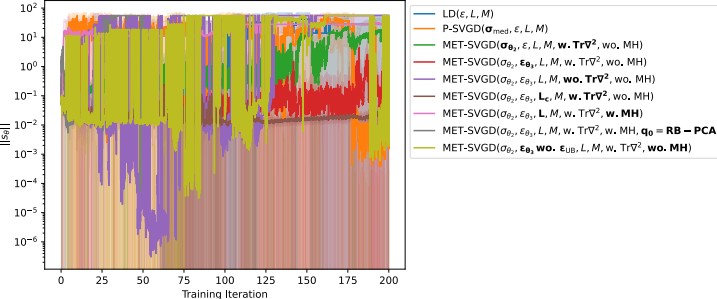

Figure 27: Smoothness of the learn EBM as measured by $||\nabla_x f_\theta(x)||_2$ throughout training iterations. The configurations with the best FID and Inception scores (Fig. 26) exhibit smoother landscapes.

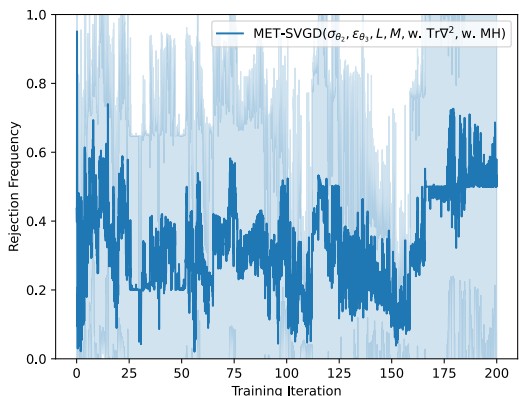

Figure 28: MH rejection rate across training iterations for the *MET-SVGD*($\sigma_{\theta_2}, \epsilon_{\theta_3}, L, M$, w.Tr $\nabla^2$, wo. MH) configuration.

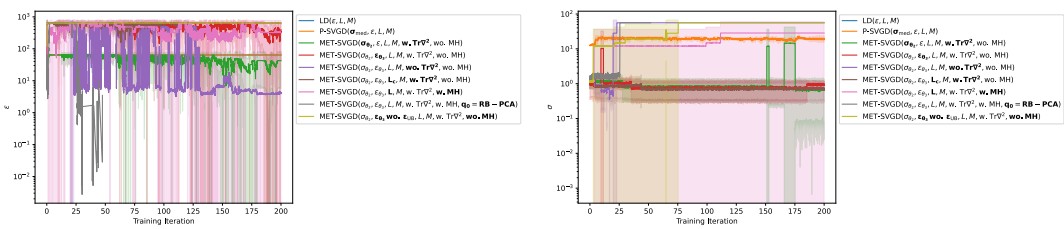

(a) Step-size $\epsilon_{\theta_3}^l$ across training iterations.  (b) Kernel bandwidth $\sigma_{\theta_2}^l$ across training iterations.

Figure 29: Learnt kernel bandwidth and step-size across training iterations. The trends exhibited by the learnt kernel bandwidth and step-size in the *MET-SVGD* configuration are significantly different from *P-SVGD*'s $\sigma_{\text{med}}$ and constant step-size.

| Configuration | FID | Stability |
|---|---|---|
| $LD(\epsilon, L, M)$ | $28.5 \pm 6.9$ | 59.4 |
| $P\text{-}SVGD(\sigma_{\text{med}}, \epsilon, L, M)$ | $104. \pm 17.$ | 54.5 |
| $MET\text{-}SVGD(\sigma_{\theta_2}, \epsilon, L, M, \text{w. Tr}\nabla^2, \text{wo. MH})$ | $28.1 \pm 5.1$ | 76.1 |
| $MET\text{-}SVGD(\sigma_{\theta_2}, \epsilon_{\theta_3}, L, M, \text{w. Tr}\nabla^2, \text{wo. MH})$ | $20.5 \pm 0.4$ | 59.4 |
| $MET\text{-}SVGD(\sigma_{\theta_2}, \epsilon_{\theta_3}, L, M, \text{wo. Tr}\nabla^2, \text{wo. MH})$ | $54.4 \pm 6.5$ | 16.7 |
| $MET\text{-}SVGD(\sigma_{\theta_2}, \epsilon_{\theta_3}, L_c, M, \text{w. Tr}\nabla^2, \text{wo. MH})$ | $21.7 \pm 0.5$ | 0.846 |
| $MET\text{-}SVGD(\sigma_{\theta_2}, \epsilon_{\theta_3}, L, M, \text{w. Tr}\nabla^2, \text{w. MH})$ | $133. \pm 36.$ | 20.2 |
| $MET\text{-}SVGD(\sigma_{\theta_2}, \epsilon_{\theta_3} \text{ wo. } \epsilon_{\text{UB}}, L, M, \text{w. Tr}\nabla^2, \text{wo. MH})$ | $60.3 \pm 47.2$ | 39.6 |

Table 16: FID and Stability for CIFAR10 image generation with EBMs.

**Qualitative Results.** In Fig. 30, we visualize generated images sampled from the different models.

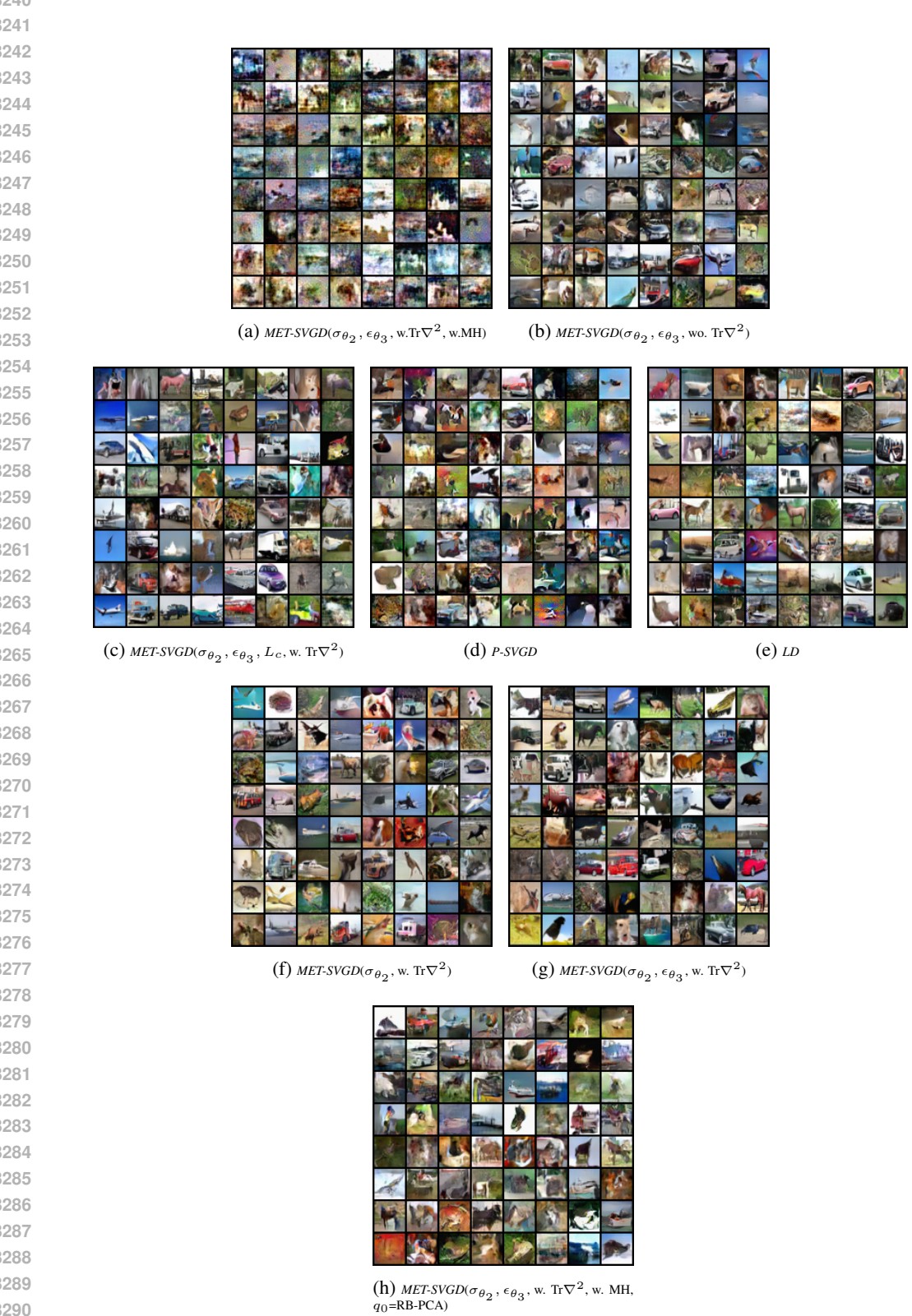

(a) *MET-SVGD*($\sigma_{\theta_2}$, $\epsilon_{\theta_3}$, w.Tr$\nabla^2$, w.MH)   (b) *MET-SVGD*($\sigma_{\theta_2}$, $\epsilon_{\theta_3}$, wo. Tr$\nabla^2$)

(c) *MET-SVGD*($\sigma_{\theta_2}$, $\epsilon_{\theta_3}$, $L_c$, w. Tr$\nabla^2$)   (d) *P-SVGD*   (e) *LD*

(f) *MET-SVGD*($\sigma_{\theta_2}$, w. Tr$\nabla^2$)   (g) *MET-SVGD*($\sigma_{\theta_2}$, $\epsilon_{\theta_3}$, w. Tr$\nabla^2$)

(h) *MET-SVGD*($\sigma_{\theta_2}$, $\epsilon_{\theta_3}$, w. Tr$\nabla^2$, w. MH, $q_0$=RB-PCA)

Figure 30: Image generation using EBMs across different configurations.

# J   ADDITIONAL RESULTS: MAXENTR RL (FIG. **??**)

**Implementation Details** are reported in Tab. 17.

| Category | Hyperparameter | Value |
|---|---|---|
| Training | Optimizer | Adam |
| | Actor / Critic learning rate | $10^{-4}$ for Humanoid, $10^{-3}$ for all other environments |
| | Batch size | 100 |
| Critic Deepnet | Hidden layers | 2 |
| | Hidden units per layer | 256 |
| | Nonlinearity | ELU |
| RL | Target smoothing coefficient | 0.005 |
| | Discount $\gamma$ | 0.99 |
| | Target update interval | 1 |
| | Entropy weight $\alpha$ | 0.2 |
| | Replay buffer size $|\mathcal{D}|$ | $10^6$ |
| SVGD | Initial distribution | $q_0 = \mathcal{N}(\mu_{\theta_1}(s_t), \mathrm{diag}(\sigma_{\theta_1}(s_t)))$, where $\mu_{\theta_1}$ and $\sigma_{\theta_1}$ are a 3 hidden layers MLP with 256 units per layer and ELU as an activation function |
| | Number of steps | $L = 3$ |
| | Number of particles | $M \in \{10, 64\}$ |
| | Kernel bandwidth | *P-SVGD*: $\sigma = \sigma_{\mathrm{med}}$ |
| | | *MET-SVGD*: $\sigma = \mathrm{GNN}_1(s_t, \{x_i^l\}, \{\nabla_{x_i^l} \log p(x_i^l)\}; 256, \theta_2)$ (Tab. 18) |
| | Step-size | *P-SVGD*: $\epsilon = 0.1$ |
| | | *MET-SVGD*: $\epsilon = \mathrm{GNN}_2(s_t, \{x_i^l\}, \{\nabla_{x_i^l} \log p(x_i^l)\}; 256, \theta_3)$ (Tab. 19) |

Table 17: Implementation details for the setup in Fig. **??**.

| Step | Operation | Description |
|---|---|---|
| 1.a | Linear layer followed by ReLU (output $h$) | First particle difference embedding layer: $x_i^l - x_j^l \to \mathrm{ReLU}(W_1(x_i^l - x_j^l))$ |
| 1.b | Linear layer followed by ReLU (output $h$) | First score difference embedding layer: $\nabla_{x_i^l} \log p(x_i^l) - \nabla_{x_j^l} \log p(x_j^l) \to \mathrm{ReLU}(\tilde{W}_1(\nabla_{x_i^l} \log p(x_i^l) - \nabla_{x_j^l} \log p(x_j^l)))$ |
| 1.c | Linear layer followed by ReLU (output $h$) | First state embedding layer: $s_t \to \mathrm{ReLU}(\bar{W}_1 s_t)$ |
| 2.a | Linear layer (output $h$) | Second particle difference embedding layer |
| 2.b | Linear layer (output $h$) | Second score difference embedding layer |
| 2.c | Linear layer (output $h$) | Second state embedding layer |
| 3 | Embedding aggregation layer followed by ReLU (output $h$) | Sum particle difference, score difference, and state embeddings |
| 4 | Mean aggregation (output $h$) | Average embeddings over the batch dimension |
| 5 | Linear layer followed by ReLU (output $h$) | |
| 6 | Linear layer (output 1) | Map to scalar |
| 7 | $\exp(\cdot)$ | Ensure positivity |

Table 18: Architecture of $\mathrm{GNN}_1(s_t, \{x_i^l\}, \{\nabla_{x_i^l} \log p(x_i^l)\}; h, \theta_3)$ for the setup in Fig. **??**.

**Performance.** In Fig. 31a and Fig. 31b, we report the Inter Quantile Mean (IQM) return values averaged over 5 runs, where every run is the average of 10 evaluations of the policy.

In both of the Walker2d-v2 and Humanoid-v2 environments (Fig. 32), jointly learning the kernel bandwidth and restoring the trace term yields substantial performance gains over *P-SVGD*. In contrast, learning the step-size without the correction term underperforms all other configurations. This is due to the fact that the learned step-size in this setting becomes comparatively large, causing the

| Step | Operation | Description |
|------|-----------|-------------|
| 1.a | Linear layer followed by ReLU (output $h$) | First particle embedding layer: $x_i^l \rightarrow \text{ReLU}(W_1 x_i^l)$ |
| 1.b | Linear layer followed by ReLU (output $h$) | First score embedding layer: $\nabla_{x_i^l} \log p(x_i^l) \rightarrow \text{ReLU}(\tilde{W}_1 \nabla_{x_i^l} \log p(x_i^l))$ |
| 1.c | Linear layer followed by ReLU (output $h$) | First state embedding layer: $s_t \rightarrow \text{ReLU}(\bar{W}_1 s_t)$ |
| 2.a | Linear layer (output $h$) | Second particle embedding layer |
| 2.b | Linear layer (output $h$) | Second score embedding layer |
| 2.c | Linear layer (output $h$) | Second state embedding layer |
| 3 | Embedding aggregation layer followed by ReLU (output $h$) | Sum particle, score, and state embeddings |
| 4 | Mean aggregation (output $h$) | Average embeddings over the batch dimension |
| 5 | Linear layer followed by ReLU (output $h$) | |
| 6 | Linear layer followed by ReLU (output $h$) | |
| 7 | Linear layer (output 1) | Map to scalar |
| 8 | $\exp(\cdot)$ | Ensure positivity |

Table 19: Architecture of $\text{GNN}_2(s_t, \{x_i^l\}, \{\nabla_{x_i^l} \log p(x_i^l)\}; h, \theta_3)$ for the setup in Fig. **??**.

particles to drift into non-smooth regions of the landscape (Fig. 41). Using an adaptive number of steps significantly improves returns in the Humanoid-v2 environment, yielding the best overall performance, but provides no clear benefit in Walker2d-v2. This discrepancy is due to Humanoid-v2 having a more complex target landscape than Walker2d-v2. Incorporating an MH step provides a small improvement in Walker2d-v2 but not in Humanoid-v2, where it restricts exploration in an already highly complex landscape. The highest returns in Walker2d-v2 are achieved by combining MH with an $\epsilon$-greedy schedule, *i.e.*, high MH probability in later stages and lower probability early on, which preserves early exploration while improving late-stage exploitation.

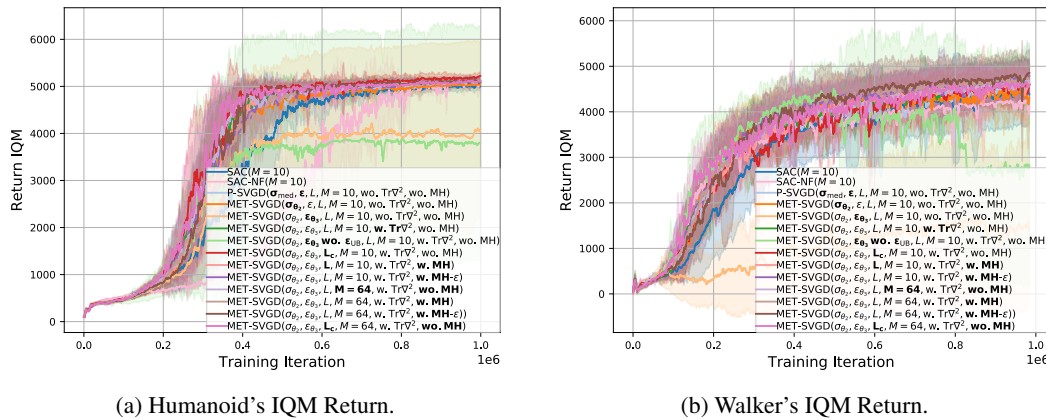

(a) Humanoid's IQM Return.  (b) Walker's IQM Return.

Figure 31: IQM return scores across training iterations. In both environments, *MET-SVGD* significantly outperforms all baselines.

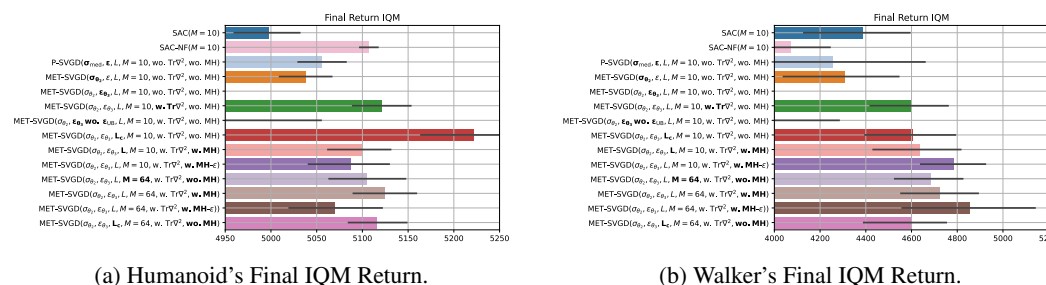

(a) Humanoid's Final IQM Return.

(b) Walker's Final IQM Return.

Figure 32: Final IQM return scores across environments. In both environments, the best *MET-SVGD* configuration significantly outperforms all baselines.

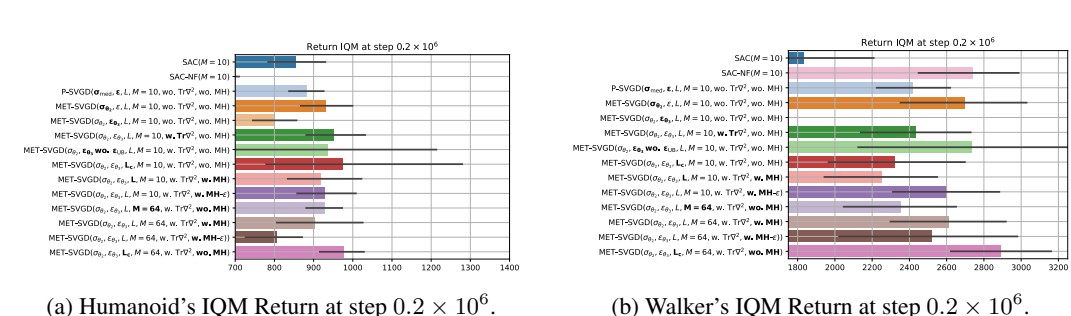

(a) Humanoid's IQM Return at step $0.2 \times 10^6$.

(b) Walker's IQM Return at step $0.2 \times 10^6$.

Figure 33: IQM return scores at step $0.2 \times 10^6$ across environments. In Walker2d-v2, though *MET-SVGD* has similar early returns to *SAC-NF* initially, it does not mode collapse as *SAC-NF* (Fig. 32).

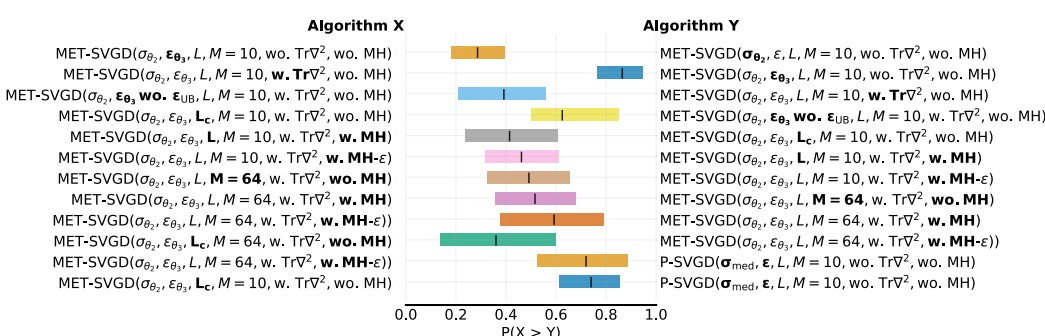

Figure 34: Probability of Improvement. *MET-SVGD* outperforms P-SVGD with a probability greater than 75%, and within *MET-SVGD* configurations, the largest performance improvement is achieved when the correction term is restored.

**Effect of $\sigma_{\theta_2}$ and $\epsilon_{\theta_3}$ (Humanoid-v2)**

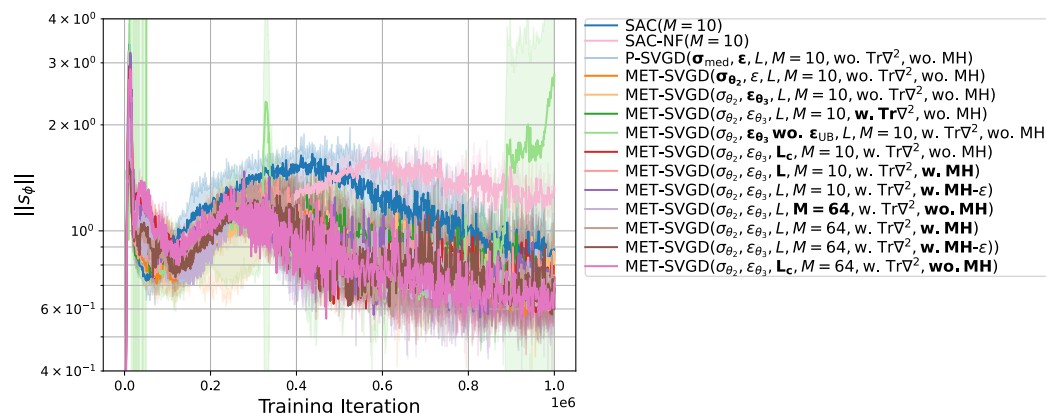

Figure 35: Average of $||s_\phi(a; s_t)||_2$ across training iterations in the Humanoid-v2 environment, where $s_\phi$ is the score function. Not incorporating the step-size bound when learning the step-size led to non-smoothness in the landscape and eventual divergence toward the end of the training.

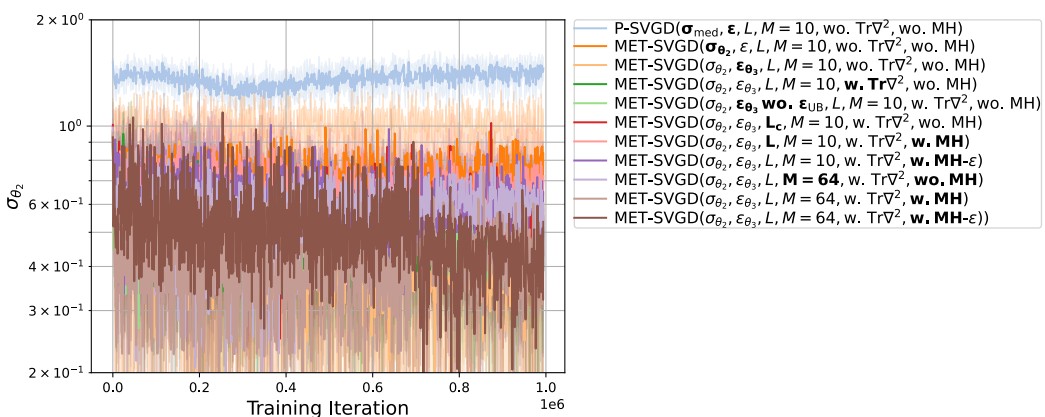

Figure 36: Kernel bandwidth across training iterations in the Humanoid-v2 environment. The optimal $\sigma_{\theta_2}$ and $\sigma_{\text{med}}$ show significantly different trends. Specifically, $\sigma_{\theta_2}$ is smaller than $\sigma_{\text{med}}$ by an order of magnitude, enabling more exploration.

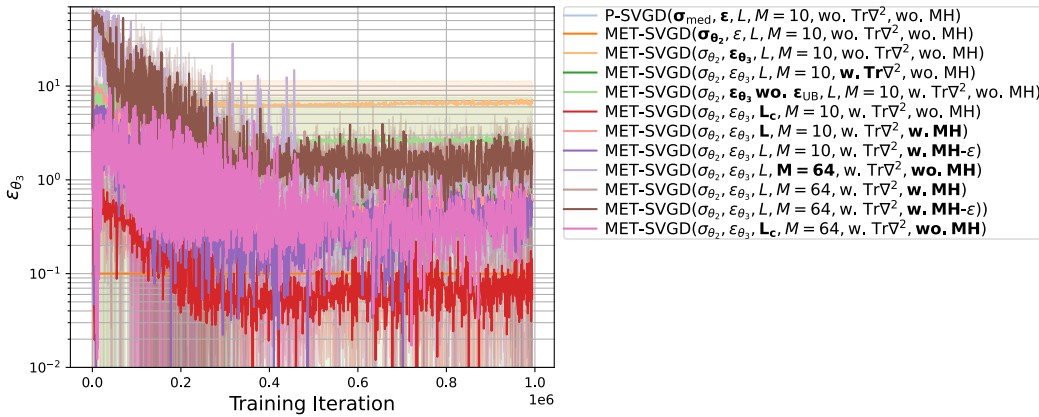

Figure 37: Step-size across training iterations in the Humanoid-v2 environment. The optimal $\epsilon_{\theta_3}$ is larger than the fixed $\epsilon$, except in the configuration with $L_c$. This enables faster convergence to the target.

**Effect of $\sigma_{\theta_2}$ and $\epsilon_{\theta_3}$ (Walker2d-v2)**

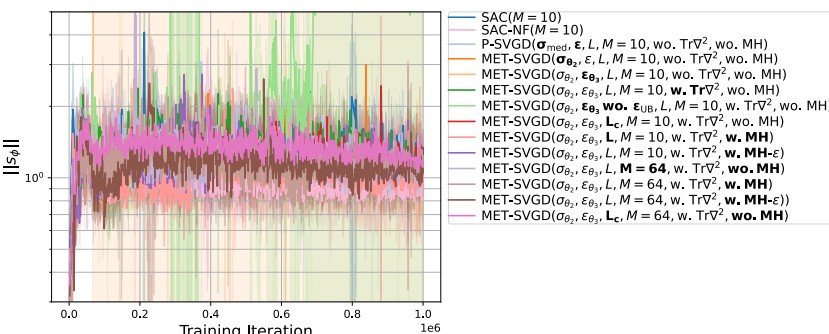

Figure 38: Average of $||s_\phi(a; s_t)||_2$ across training iterations in the Walker2d-v2 environment, where $s_\phi$ is the score function. The best *MET-SVGD* configuration (brown) leads to a smoother landscape compared to *P-SVGD*, facilitating convergence to the target.

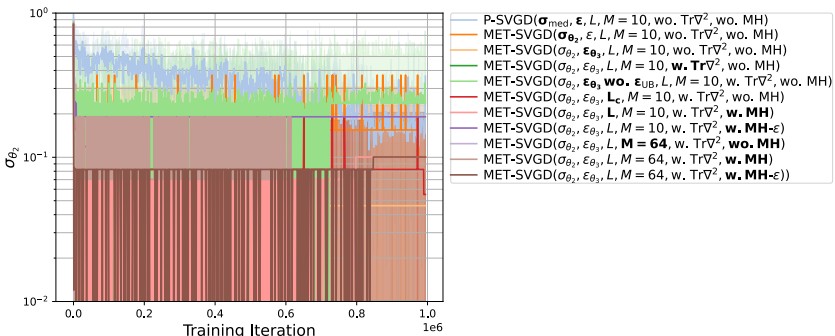

Figure 39: Kernel bandwidth across training iterations in the Walker2d-v2 environment. The optimal $\sigma_{\theta_2}$ and $\sigma_{\text{med}}$ show significantly different trends. Specifically, $\sigma_{\theta_2}$ is smaller than $\sigma_{\text{med}}$, enabling more exploration.

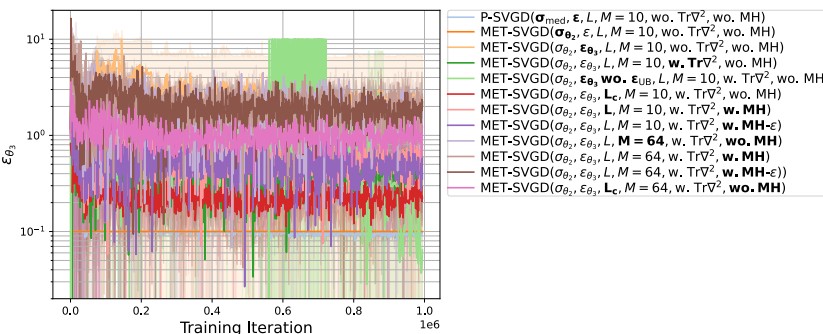

Figure 40: Step-size across training iterations in the Walker2d-v2 environment. For all *MET-SVGD* configurations, the optimal $\epsilon_{\theta_3}$ is larger than *P-SVGD*'s fixed $\epsilon$, leading to faster convergence to the target.

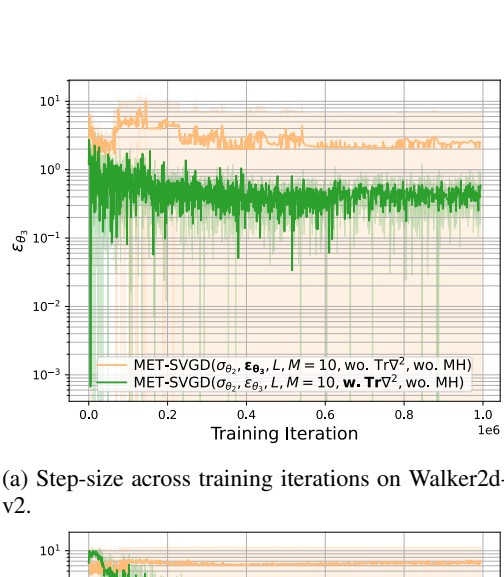

(a) Step-size across training iterations on Walker2d-v2.

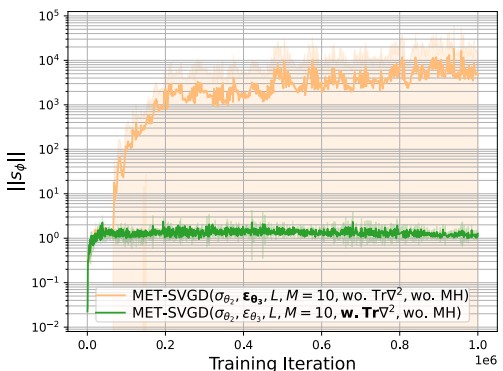

(b) Average of $||s_\phi(a; s_t)||_2$ across training iterations on Walker2d-v2, where $s_\phi$ is the score function.

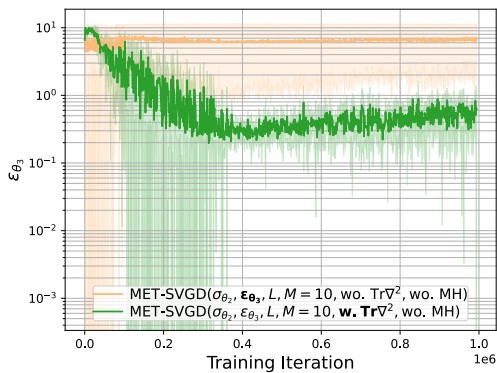

(c) Step-size across training iterations on Humanoid-v2.

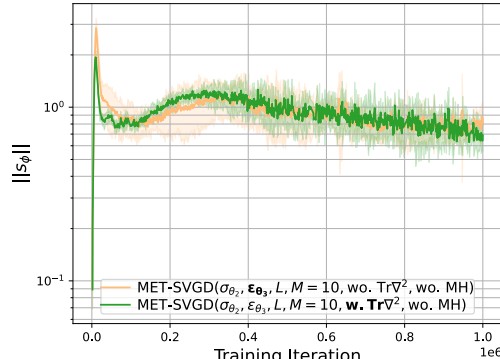

(d) Average of $||s_\phi(a; s_t)||_2$ across training iterations on Humanoid-v2, where $s_\phi$ is the score function.

Figure 41: Diagnosis of divergence in the configuration where the step-size is learned without the correction term. In Walker2d-v2, the large magnitude of the step-size in the configuration without the correction term led to learning a very non-smooth landscape.

**Stein Identity's Violation.**

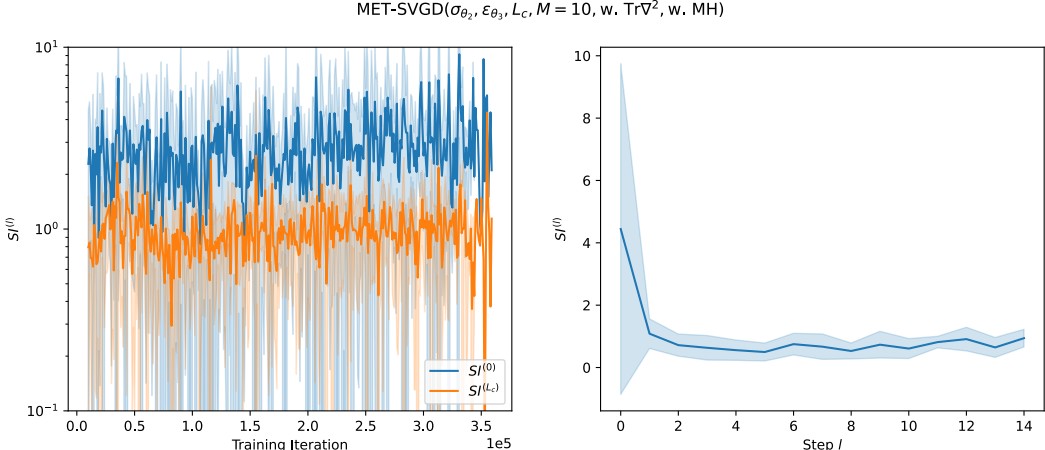

Figure 42: Stein's identity violation in Walker2d-v2. On the left, we show the evolution of Stein's identity violation at the initial (blue) and final (orange) steps over training iterations. On the right, we show the evolution of Stein's identity violation over *SVGD* steps during the latest training iteration.

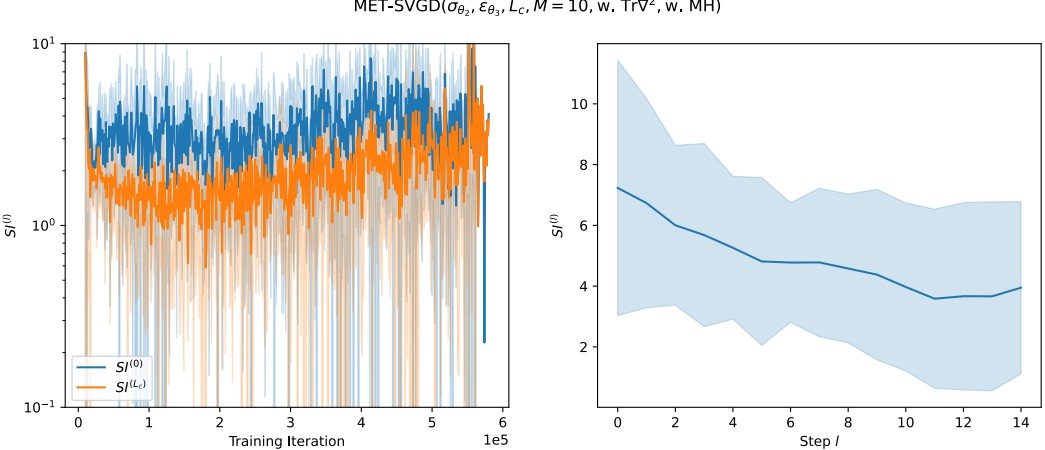

Figure 43: Stein's identity in Humanoid-v2. On the left, we show the stein identity at the initial (blue) and final (orange) steps over training iterations. On the right, we show the evolution of the stein identity over *SVGD* steps during the latest training iteration.

**MH Rejection Rate.**

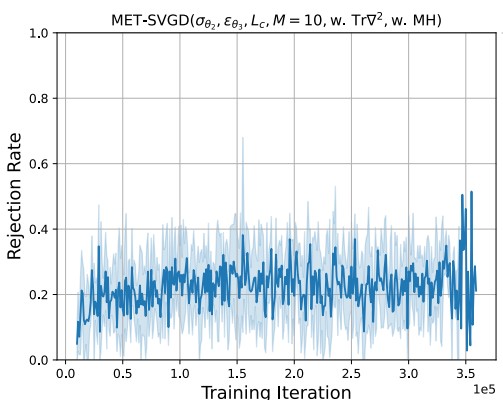

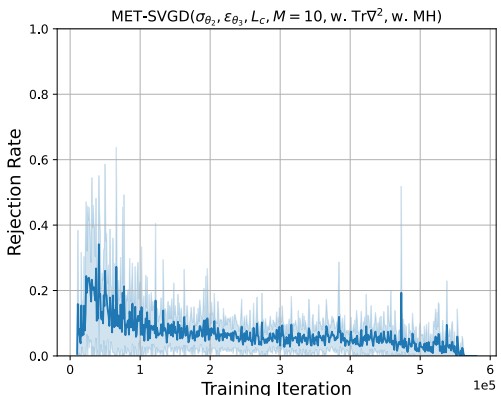

Figure 44: Metropolis-Hastings' rejection rate throughout training iterations on Walker2d-v2.

Figure 45: Metropolis-Hastings' rejection rate throughout training iterations on Humanoid-v2.

**Emprical distribution of $L_c$.**

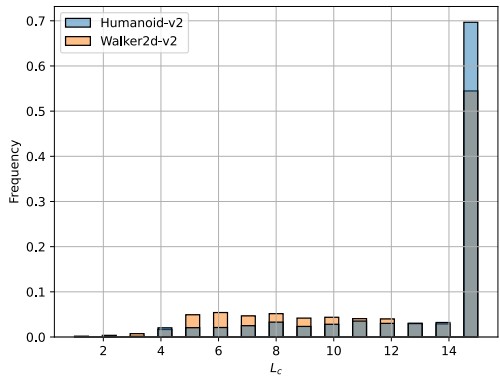

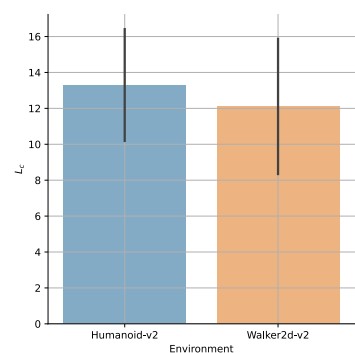

(a) Empirical distribution of $L_c$ collected across training iterations for Walker2d-v2 and Humanoid-v2.

(b) Average of $L_c$ across training iterations for Walker2d-v2 and Humanoid-v2.

Figure 46: Empirical distribution and average of $L_c$ across training iterations for Walker2d-v2 and Humanoid-v2. Humanoid-v2 requires more steps on average, which is consistent with its task's difficulty relative to Walker2d-v2.

