$$+ \underbrace{\operatorname{Tr}\left(\nabla_{x^l}^2\log p(x^l)\right)}_{③} + \mathcal{O}\left(\left(\epsilon/\epsilon_{\mathrm{UB}}^l\right)^2\right)\Bigg]$$

$$(14)$$

In the following, we denote by $()^{(k)}$ the $k$-th dimension of the vector.

**Term ①:**

$$\begin{aligned}
\operatorname{Tr}\left(\nabla_{x^l}(\kappa(x^l,x_j^l)\nabla_{x_j^l}s_p(x_j^l)^T)\right) &= \operatorname{Tr}\left(\nabla_{x^l}\kappa(x^l,x_j^l)(\nabla_{x_j^l}s_p(x_j^l))^T + \kappa(x^l,x_j^l)\nabla_{x^l}\nabla_{x_j^l}s_p(x_j^l)\right)\\
&= \sum_{t=1}^{d}\frac{\partial\kappa(x^l,x_j^l)}{\partial(x^l)^{(t)}}\frac{\partial s_p(x_j^l)}{\partial(x_j^l)^{(t)}} + 0\\
&= (\nabla_{x^l}\kappa(x^l,x_j^l))^T\nabla_{x_j^l}s_p(x_j^l)\\
&= -\frac{1}{\sigma^2}\kappa(x^l,x_j^l)(x^l-x_j^l)^\top\nabla_{x_j^l}s_p(x_j^l)
\end{aligned}$$

**Term ②:**

$$\begin{aligned}
\operatorname{Tr}\left(\nabla_{x^l}\nabla_{x_j^l}\kappa(x^l,x_j^l)\right) &= \operatorname{Tr}\left(\nabla_{x^l}\left(\frac{1}{\sigma^2}\kappa(x^l,x_j^l)(x^l-x_j^l)\right)\right)\\
&= \frac{1}{\sigma^2}\sum_{k=1}^{d}\left(\frac{\partial\kappa(x^l,x_j^l)}{\partial(x^l)^{(k)}}(x^l-x_j^l)^{(k)} + \kappa(x^l,x_j^l)\right)\\
&= \frac{1}{\sigma^2}\left(\nabla_{x^l}\kappa(x^l,x_j^l)^\top(x^l-x_j^l) + d\times\kappa(x^l,x_j^l)\right)\\
&= \frac{1}{\sigma^2}\left(\nabla_{x^l}\kappa(x^l,x_j^l)^\top(x^l-x_j^l) + d\times\kappa(x^l,x_j^l)\right)\\
&= -\frac{1}{\sigma^4}\times\kappa(x^l,x_j^l)\|x^l-x_j^l\|^2 + \frac{1}{\sigma^2}\times d\times\kappa(x^l,x_j^l)\\