# OpenReview forum: "Particles Don’t Care About Z: Towards Scaling Entropy Estimation of Unnormalized Densities"
_ICLR.cc/2026/Conference — Submitted to ICLR 2026_

### Official Review · Reviewer_Cp7b · 2025-10-27

**Soundness:** 3
**Presentation:** 4
**Contribution:** 3
**Rating:** 8
**Confidence:** 3

**Summary:**

The paper introduces MET-SVGD, a nonparametric method for entropy estimation that combines the kernelized Stein discrepancy with a Metropolis–Hastings correction. The approach is motivated by shortcomings in entropy estimation using P-SVGD. The authors identify a missing term in the log-determinant as the true source of divergence in P-SVGD—contrary to prior claims attributing it to violations of the invertibility assumption. To address this, the paper proposes a correction based on Hutchinson’s trace estimator, along with adaptive strategies for step size and kernel bandwidth, and a new convergence criterion.

**Strengths:**

- Careful analysis supports the claim that invertibility, per se, is not the cause of P-SVGD diverging.
- The computable Stein-identity–based convergence check is broadly useful beyond MET-SVGD.
- Empirical studies on both synthetic and real problems convincingly validate MET-SVGD claim to improve upon P-SVGD.
- The appendix (preliminaries and derivations) is clear and well organized.

**Weaknesses:**

- The article does not elaborate on the magnitude or interpretation of $L_C$ in the evaluation.

**Questions:**

## Questions
- What exactly is meant by *global* invertibility of the SVGD map? Does this refer to the composed transport $f$ being globally invertible?
- If Proposition 3.1 requires $\epsilon \nabla \phi(x^\ell)$ to be contractive (see Technicality below), what ensures that this assumption holds throughout optimization?
- In Figure 4, it is not clear that $L_C$ corresponds to the step-size $\to 0$ limit. Could you clarify the expected asymptotic behavior?
- In the experiments, do the MET-SVGD runs satisfy the Stein-identity (SI) convergence criterion in practice?

## Technicalities
- In Proposition 3.1, are you assuming that $\phi^\ell$ is contractive? Otherwise, it is unclear how item (iii) in the appendix derivation follows.
- Could you elaborate on the final equality on line 2066 and the assumptions required for it to hold?

---

> ### Author Response · Authors · 2025-11-29
>
> **Interpretation of $L_c$.** $L_c$ can be interpreted as the number of reasoning steps adaptively scaling with the difficulty of action selection, i.e., the complexity of $\pi(a|s)$. Intuitively, more complex or higher-dimensional action spaces require more reasoning steps.
>
> In Fig. 46, we include histograms of the empirical distribution of $L_c$ collected across training iterations for both the Walker-2d and Humanoid environments. As expected, the Humanoid task exhibits a noticeably larger $L_c$, reflecting its significantly higher dimensionality and more complex action-selection landscape. Note that 14 is the maximum number of allowed steps.
> ____________________________
> **Global invertibility.** We use global invertibility to refer to the standard invertibility: a function $f$ is globally invertible if and only if it's bijective on its entire domain. In contrast, a function is locally invertible if, around every point, it behaves like a reversible map, i.e., there exists a neighborhood on which $f$ is bijective. For instance a circle is locally invertible but not globally invertible. Because the SVGD-induced distribution is defined through the change-of-variables formula at every update step, each SVGD update must be invertible in order for the density transformation to be well defined. Ensuring invertibility at each step also guarantees that the composition of updates (the full transport after multiple iterations) is globally invertible.
> ____________________________
> **Prop. 3.1.** Corr. 3.3 provides an upper bound on the step size $\epsilon$ that ensures $\epsilon \nabla \phi$ is contractive. To guarantee that this condition holds throughout optimization, we enforce the bound at every iteration by truncating the learned step size: $\epsilon^l_{\theta_3}\leftarrow\min\left(\epsilon^l_{\theta_3},\epsilon^l_{\text{UB}}\right).$
> We added a clarifying sentence in the revised manuscript (L.345-346).
> In practice, this constraint turns out to be quite important: without it, the EBM loss may diverge (gray trace in Fig.7), and the RL returns can drop noticeably (light-green trace Fig.8).
> ____________________________
> **$\epsilon_{\theta_3}  \rightarrow 0 \Rightarrow SI(q^l_{\theta},p)$?.** Thank you for bringing this up. This is only valid if, additionally, $\nabla_{\epsilon_{\theta_3}} D_{KL}(q||p)$ evaluated at $\epsilon_{\theta_3}=0$ is $0$, as the negative gradient of KL divergence exactly equals the square Stein discrepancy when $\epsilon=0$. This follows from the SVGD derivation (Theorem 3.1 in (Liu & Wang, 2016)):
>
> **Theorem 3.1.** Let $T(x) = x + \epsilon \phi(x)$ and $q_{[T]}(z)$ be the density of $z = T(x)$ when $x \sim q(x)$. Then $\nabla_{\epsilon} \mathrm{KL} \left(q_{[T]} | p \right)$ evaluated at $\epsilon = 0$ is $-\mathbb{E}_{x \sim q} \left[ \mathrm{Tr} \big( \mathcal{A}_p \phi(x) \big) \right]$, where $ \mathcal{A}_p \phi(x) = \nabla_x \log p(x) \phi(x)^\top + \nabla_x\phi(x) $ is the Stein operator.
>
> We added this clarification to the paper (L. 347-349).
> ____________________________
> **Stein Identity in experiments.** We add the plots of the kernalized Stein discrepancy in Fig. 42 (Walker2d) and Fig. 43 (Humanoid). The right sub-plots visualize the starting ($l=0$, blue) and end value ($L_c=0$, orange) of SI throughout the training iterations. SI is consistently dropping with the number of steps throughout training. In particular, the left plot visualizes  SI across SVGD steps of the trained model.  While SI does not always converge exactly to zero, this is expected due to our stopping criteria:
> 1. we halt the SVGD update if SI increases (to prevent divergence), and
> 2. we cap the number of SVGD steps.
>
> Under these constraints, MET-SVGD reliably reduces SI throughout training, demonstrating approximate satisfaction of the Stein identity in practice.
> ____________________________
> **$\phi^l$ contractivity.** Thank you for pointing this out. We clarify that Prop. 3.1 does *not* assume that $\phi^l$ itself is contractive. The required condition is that the *update map* $ f^l = I + \epsilon \phi^l$ is contractive, which corresponds to the bound $\epsilon  \mathrm{Lip}(\phi^l) < 1$.
> Thus, $\phi^l$ only needs to be Lipschitz continuous; it is the scaled map $\epsilon \phi^l$ that must satisfy the contraction condition.
> We have updated the wording of Prop. 3.1 to eliminate the ambiguity.
> The proof that $\phi^l$ is Lipschitz continuous in case of the SVGD update is included in Appendix D.2.
> ____________________________
> **Proof of Proposition 3.4.** We elaborate in Appendix F.

---

### Official Review · Reviewer_TwdB · 2025-10-31

**Soundness:** 3
**Presentation:** 2
**Contribution:** 3
**Rating:** 6
**Confidence:** 3

**Summary:**

The paper introduces MET-SVGD, a principled extension of P-SVGD (Messaoud et al., 2024) for entropy estimation of distributions known only up to a normalization constant. The authors identify and correct fundamental algorithmic flaws in P-SVGD that limit its scalability to high dimensions, including: (1) misdiagnosed sensitivity to SVGD hyperparameters, (2) violation of global invertibility assumptions, (3) omission of a critical trace-of-Hessian term, and (4) suboptimal divergence control heuristics. MET-SVGD addresses these issues through several innovations: a unified sufficient condition for global invertibility and log-det approximation;  end-to-end learning of kernel bandwidth and step-size; adaptive determination of sampling steps via Stein Identity;  efficient restoration of the missing trace-of-Hessian term; and  replacement of heuristic divergence control with Metropolis-Hastings correction. The method provides both theoretical guarantees and empirical improvements.

**Strengths:**

The paper tackles the specific problem of estimating differential entropy for distributions known only up to a normalization constant, a critical challenge with broad theoretical and practical significance. While entropy estimation from samples has been extensively studied, the unnormalized density setting is central to important applications like Energy-Based Models and Maximum Entropy Reinforcement Learning, yet remains underexplored due to scalability challenges.

The paper effectively articulates the longstanding challenge of entropy estimation for unnormalized densities and clearly identifies limitations in prior work (P-SVGD). It provide rigorous mathematical analysis of P-SVGD's limitations and derive principled solutions with formal guarantees (Propositions 3.1-3.6). The unified condition for global invertibility (Corollary 3.3) is particularly elegant. The evaluation spans multiple domains (Gaussian/GMM entropy estimation, EBM training, MaxEnt RL) with thorough ablation studies demonstrating each component's contribution. Results consistently show substantial improvements over baselines.

**Weaknesses:**

While the paper provides a thorough comparison with SVGD-based methods, it offers minimal comparison to alternative entropy estimation approaches, such as normalizing flows or other MCMC variants capable of handling unnormalized densities.

Additionally, I strongly recommend including numerical experiments using data from known models, along with well-designed ablation studies that examine the effects of model complexity and dimensionality.

The paper is quite dense and technical. It would be helpful to include a dedicated section or subsection that clearly explains the proposed method and its algorithm/implementation.

**Questions:**

How does MET-SVGD compare to non-SVGD approaches for entropy estimation from unnormalized densities, such as normalizing flows or other MCMC variants?
Could you quantify the MH acceptance rate in your CIFAR-10 experiments (e.g., average rate across training)? What strategies might mitigate high rejection rates for complex high-dimensional targets?
How should a user choose the number of particles M based on problem dimensionality or complexity? Are there empirical or theoretical guidelines for this parameter?

---

> ### Author Response · Authors · 2025-11-29
>
> We thank the reviewer for the positive feedback on our method and address the remaining concerns in the following.
> ___________________________
> **Comparison to Normalizing Flows (NF).**
> We include normalizing flows as baselines in the EBM and RL experiments in Fig. 7 (NF-Glow) and Fig. 8 (SAC-NF), respectively. Across all settings, flow baselines underperform MET-SVGD: they diverge in the EBM task (similarly to MH; due to poor early samples) and exhibit strong mode collapse in the RL task, leading to limited exploration and hence low returns (Fig. 32b shows SAC-NF saturating early during training). This outcome is consistent with known challenges in training flows, i.e. training instability and mode collapse. Crucially, unlike normalizing flows, both P-SVGD and MET-SVGD explicitly leverage the unnormalized target density $\bar{p}$ through $\nabla \log\bar{p}$ (Eq. 2) which provides better guidance than relying on log-likelihood optimizations and more flexibility than flow models' constrained architectures. We have updated the experimental section to include these insights.
>
> Finally, as illustrated in Fig. 11, MET-SVGD can be viewed as a flow model with a full-rank Jacobian and an adaptive number of layers.
> ___________________________
> **Comparison to MCMC methods.** We actually compare to Langevin dynamics, an MCMC variant, in Fig. 2d (grey curve). We show that it's more sensitive to the trace approximation as the trace term in P/MET-SVGD derived distributions is scaled by the number of particles $M$ (L.418-420 in the updated paper).
> ___________________________
> **Experiments with known models.** We do include extensive experiments on known, analytically tractable targets. Specifically, we evaluate MET-SVGD on a variety of Gaussian (Figs. 2a, 2d, and 17) and Gaussian mixture models (Figs. 6b and 23) targets, including high-dimensional settings. Across all of these experiments, MET-SVGD consistently and substantially outperforms the baselines (Sec. 4.1 and Appendix H).
>
> In addition, as suggested by the reviewer, we have added a full ablation study for all toy experiments in the main paper to the App. G,
> * Fig. 2a: ablation in Fig. 15
> * Fig. 2b: ablation in Fig. 16
> * Fig. 2c: ablation in Fig. 20
> * Fig. 2d: ablation in Fig. 18
>
> Trends:
> * In setups with smooth targets, adding the step-size bound does not affect performance, as the selected step-size naturally satisfies the upper bound
> * Including the correction term and learning the kernel bandwidth help, but are not always enough to recover the target's entropy
> * Learning the step-size significantly improves entropy estimation. However, in Fig. 16, we show that it is not enough to recover the target's entropy due to the non-smoothness of the landscape
> * Adding MH helps in setups with a non-smooth target distribution, as illustrated in Fig. 16
> ___________________________
> **MET-SVGD algorithm.** We thank the reviewer for this suggestion.
> In response, we moved the algorithm from the appendix to the main paper, and split it into two parts: training (Alg. 1) and inference (Alg. 2).
> We hope this improves readability, and we welcome any further suggestions.
> ___________________________
> **MH rejection rates** are reported in Fig. 28 for Cifar10. We added the MH rejection rates for the RL experiments in Fig. 44 and Fig. 45, respectively.
> * CIFAR-10 (Fig. 25): The rejection rate is initially high (≈0.4–0.8), indicating that early MH proposals fall in low-density regions, resulting in poor samples and the loss divergence as explained in Sec. 4.2.
> * MaxEnt RL (Figs. 44-45): The rejection rates remain low throughout training, showing that the SVGD proposal stays close to the evolving policy while still maintaining sufficient exploration.
> ___________________________
> **Strategies to mitigate high rejection rates.** As shown in Prop. 3.5, the acceptance probability is proportional to $Tr(\nabla \phi)$, which is proportional to the scores, the trace of the Hessian and inversely proportional to the RBF kernel variance (Eq. 2). So, to increase the acceptance probability, we need to either have smoother targets or kernels. Smoother kernels are tricky as enforcing the bandwidth $\sigma$ to be large, will negatively affect the samples quality. When the unormalized target is learnt, regularization or spectral normalization of the layers can be used to control the Lipschitz continuity [1].
>
> [1] Miyato, Takeru, et al. "Spectral normalization for generative adversarial networks." arXiv preprint arXiv:1802.05957 (2018).

---

> > ### Author Response · Authors · 2025-11-29
> >
> > **Number of Particles.** Currently, there is no theoretically optimal rule for choosing $M$ in SVGD-type methods. Shi & Mackey (2024) provide the strongest available result, showing that SVGD with $M$ particles drives the KSD to zero at a rate of $\mathcal{O}\left(\frac{1}{\sqrt{\log\log M}}\right)$. But this bound does not yield a practical formula for selecting $M$: the double–logarithmic dependence is extremely loose, meaning that even astronomically large increases in $M$ produce only negligible theoretical improvement. Thus, the rate gives almost no actionable guidance. Intuitively, the required number of particles $M$ depends on the dimensionality and complexity of the target distribution.  In practice, $M$ is primarily constrained by computational budget, and prior work (including P-SVGD) typically uses tens to a few hundred particles depending on dimensionality. We follow the same convention in our experiments.

---

### Official Review · Reviewer_dBCy · 2025-11-01

**Soundness:** 3
**Presentation:** 1
**Contribution:** 2
**Rating:** 4
**Confidence:** 3

**Summary:**

The authors propose a direct improvement on the work of Messaoud et al. (2024). The proposed methodology of both works aims to solve the following problem: given an unnormalized density $p$ known only up to normalization constant, produce i) a variational approximation $q$ to $p$, and ii) an estimate of the entropy of $p$.The main improvements are 1) formalization of step size conditions that ensure an invertibility result, 2) an on-the-fly approach to tune necessary sampling hyperparameters, 3) a second-order correction to the trace term that improves empirical (non-asymptotic) sampling, and 4) a sampling-within-Metropolis scheme that corrects for deviations along the sampling trajectory formally compared to the heuristic of Messaoud et al.

**Strengths:**

- The paper is portrayed as a direct extension of previous related work. The concreteness of this formulation simplifies understanding of the contribution being made relative to Messaoud et al.
- Regarding the contributions, to my knowledge #1 and #3 are unique results that I have not seen before. #2 and #4 are more algorithmic tweaks, but I still recognize the contribution of changes such as these that improve empirical results
- I appreciate the framing of the approach from a more general perspective, namely the variational inference perspective. This makes the suite of potential problems for which the proposed approach is usable much broader than merely within a reinforcement learning context.

**Weaknesses:**

- I find the paper fairly messy and disorganized. The authors’ contribution does not really begin until the top of page 5, which is quite late. And there are some obvious typos (e.g., line 174, maximizing (not minimizing).
- The framing relative to Messaoud is clear enough as written, but less obvious to me when I go read Messaoud et al., which is primarily framed as a reinforcement learning paper. That paper does not explicitly propose a SVGD gradient as such, but rather how to apply SVGD to the RL problem. This paper could do a much better job motivating why invertibility and entropy are important from this perspective, as this is mostly taken for granted.
- Given the above, I find the comparison and discussion with respect to the rest of the SVGD literature to be lacking. It seems to me the others should be comparing their approach to variants of SVGD, such as \beta-SVGD (Sun and Richtarke, 2022), or at least framing their method with respect to SVGD discretization, which seems poorly understood in the literature relative to the continuous formulation.
- The experiments sometimes raise more questions than answers: for example, the divergence seen in Figure 7 seems to suggest MET-SVGD can be quite sensitive to some of the settings. Am I reading it correctly that the two best configurations (peach and bright pink) do not utilize the MH adjustment?

**Questions:**

- Can you elaborate on the use of normalizing flows for entropy estimation (line 94)? Does the user fit the flow and then estimate the entropy empirically with a Monte Carlo estimate?
- (Line 250) is the velocity $\phi$ known to be Lipschitz? If so, can you provide a reference or citation?
- (Line 391) When $q$ is defined as a mixture in this way, does this affect the ease of using $q$ for downstream applications?
- (Line 399) Convergence of what, precisely? The distribution, the entropy estimator? Convergence in what sense (in distribution, in probability, of real numbers, etc.?

---

> ### Author Response · Authors · 2025-11-29
>
> We thank the reviewer for the positive comments on the novelty of our approach and its broad applicability and address the raised concerns in the following.
>
> **W1. Paper presentation**.
> We understand that the paper may feel dense. However, we respectfully disagree that it's messy and disorganized.
>
> In this revision:
> * We added two algorithms that clearly delineate the training procedure (Alg. 1) and the inference procedure (Alg. 2).
> * We revised Fig. 3 to make it more lightweight. Specifically, we re-ordered the components for improved readability and retained only references to the Algorithms. Besides, the figure now mirrors the structure of the algorithms for better alignment.
> * To further improve readability, in Fig. 3's caption, we prefix our contributions with colored and numbered **C** and refer back to them throughout the paper to maintain a clear flow.
> * We also made sure that the captions are written to be self-contained so that readers can grasp the main ideas from the figures alone.
>
> We acknowledge that the paper requires some background knowledge and careful reading. To maximize clarity within the page limits, we structured the presentation around a small set of guiding figures with expanded captions to highlight the narrative flow:
> * Fig. 1 introduces the problem setting and situates MET-SVGD within the entropy-estimation literature.
> * Fig. 2 outlines the key limitations of P-SVGD that motivate our method.
> * Fig. 3 illustrates the methodological progression SVGD → P-SVGD → MET-SVGD, making our contributions easier to contrast.
>
> We also provide/refer to 2D illustrative experiments for each contribution, and include additional explanatory figures—e.g., Fig. 5 to clarify the correction term.
> ____________________________________
> **Typo (L.174)**: Thanks. We fixed it.
> ____________________________________
> **Positioning wrt. P-SVGD**. We respectfully disagree. While Messaoud et al. (2024) is framed through the lens of a reinforcement-learning application, the underlying technical contribution is explicitly introduced as a new variational inference framework as stated on page 2 of Messaoud et al. (line 6 from the bottom):
>
> > “Beyond RL, the backbone of S2AC is a new variational inference algorithm with a more expressive and scalable distribution characterized by a closed-form entropy estimate. We believe that this variational distribution can have a wider range of exciting applications.”
>
> Our goal in this paper is to address the limitations of this newly proposed family of variational distributions rather than focusing on the RL application.
> We also included MaxEnt RL as one motivating application (L89–L90), but our emphasis is on the broader variational-inference contribution rather than RL.
> ______________________________
> **Comparing to other SVGD methods**
> We appreciate the reviewer’s suggestion and clarify that we already compare against several SVGD variants specifically proposed to scale SVGD to high-dimensional spaces (Fig. 2d and Fig. 17):
>
> * Sliced SVGD (S-SVGD) (Gong et al., ICLR 2021)
> * Grassmann SVGD (GSVGD) (Liu et al., AISTATS 2022) with 1, 2, and 5 projection dimensions
>
> In Fig. 2d, on a high-dimensional Gaussian target, we show that both S-SVGD and GSVGD struggle to scale beyond ~20 dimensions, whereas MET-SVGD maintains high accuracy up to 1000 dimensions.
>
> In this revision, we additionally include β-SVGD (Sun \& Richtárik, 2022), which indeed scales better than S-SVGD and GSVGD (Fig. 17), but still underperforms MET-SVGD as dimensionality increases.
> ________________________________
> **Positioning the paper wrt. SVGD discretizations.**
> SVGD was originally introduced as a discrete sampling algorithm by Liu & Wang (2016):
> * Algorithm 1 presents SVGD explicitly as an iterative particle update,
> * Theorem 3.1 and Lemma 3.2 derive the optimal velocity $\phi$ for the discrete update rule, and
> * Section 3 uses the change-of-variables formula to introduce the family of variational distributions SVGD belong to.
>
> Thus, the foundational formulation of SVGD is itself discrete, and the change-of-variable argument was part of the theory from the outset. However, the explicit form of the SVGD-induced distribution was not derived in Liu & Wang. This was only done later by Messaoud et al. (2024). Our paper builds directly on this line of work: we extend and correct this variational distribution formulation.
>
> Subsequent papers introduced the ODE (continuous-time) interpretation of SVGD, mainly to leverage PDE and gradient-flow tools for proving convergence properties and studying asymptotic behavior (e.g., [1]).
>
> [1]  Qiang Liu. Stein variational gradient descent as gradient flow. NeurIPS, 2017.

---

> ### Author Response · Authors · 2025-11-29
>
> **MH in the EBM experiment**. Yes, the MH run diverges in the EBM experiments.
> We kindly refer the reviewer to the last paragraph in Sec. 4.2, dedicated to explain this result.
> To summarize, this issue is not inherent to MET-SVGD itself; rather, it arises from the combination of MH rejection dynamics and the instability of the contrastive divergence loss variants. Specifically, $L_{\text{ELBO}}$ is not lower-bounded (can be driven to $-\infty$) and thus well-known for being unstable. Its stability depends critically on the quality of samples used to approximate the second expectation. When the underlying energy landscape is highly complex (e.g., high-dimensional and multi-modal), the MH acceptance rate can be low during the initial stages of training (Fig. 28). As a result, the particles fail to move toward the high-density regions of the target distribution. This leads to poor samples and eventually loss divergence.
> For losses based on the reverse KL divergence (entropy estimation and RL experiments), MET-SVGD with MH remains stable and consistently performs well.
> ______________________________
> **Comparison to flow models.** Yes, the standard procedure is to first fit a flow model to the target distribution and then estimate its entropy via Monte Carlo samples.
>
>  We include normalizing flows as baselines in the EBM and RL experiments in Fig. 7 (NF-Glow) and Fig. 8 (SAC-NF), respectively. Across all settings, flow baselines underperform MET-SVGD: they diverge in the EBM task (similarly to MH; due to poor early samples) and exhibit strong mode collapse in the RL task, leading to limited exploration and hence low returns (Fig. 32b shows SAC-NF saturating early during training). This outcome is consistent with known challenges in training flows, i.e., training instability and mode collapse. Crucially, unlike normalizing flows, both P-SVGD and MET-SVGD explicitly leverage the unnormalized target density $\bar{p}$ through $\nabla \bar{p}$ (Eq. 2) which provides better guidance than replying on log-likelihood optimizations and more flexibility than flow models constrained architectures. We have updated the experimental section to include these insights.
>
> Finally, as illustrated in Fig. 11, MET-SVGD can be viewed as a flow model with a full-rank Jacobian and an adaptive number of layers.
> ____________________________
> **Lipschitz continuity of $\phi$.**
> Thank you for raising this point. We clarified the conditions under which
> $\phi$ is Lipschitz continuous. Specifically:
>
> * We revised the wording of Prop. 3.1
>
> * We added the explicit regularity assumptions required for  $\phi$ with an RBF kernel to be Lipschitz in Prop. 3.1, i.e., Lipschitz target density (implied by continuous differentiability).
>
> * We expanded the proof in App. D.2 to show that, under this condition, $\phi$ is Lipschitz continuous.
> ____________________________
> **$q$ as a mixture (Prop. 3.6)**.
>    For better clarity, we changed Prop 3.6 to introduce the step-wise MH-augmented density instead of the final one at the end of sampling. This density is straightforward to implement: the mixture structure is handled implicitly by the recursion and does not require storing or manipulating an exponentially large mixture.
>
> ____________________________
> **Convergence (452-457)**. The MET-SVGD converges strongly (in total variation) to the target distribution as it is transformed into a Metrpolis-Hastings algorithm with irreducible, aperiodic and reversible chains. We clarify in the paper and in App. G1.

---

### Official Review · Reviewer_vDmA · 2025-11-05

**Soundness:** 3
**Presentation:** 1
**Contribution:** 2
**Rating:** 4
**Confidence:** 3

**Summary:**

This paper deals with estimating differential entropy of a distribution whose density is known only up to the normalization constant. Specifically, a method called MET-SVGD is proposed, which is a refinement of the first-order approximation of the particle distribution in Stein Variational Gradient Descent (SVGD) due to Messaoud et al. (2024, Theorem 3.3). The refinement includes 1) conditions on the step-size for invertibility and log-det approximation, 2) optimized SVGD parameters, 3) corrected derivation OF the particle distribution, and 4) divergence control via Metropolis-Hastings. As a results, the MET-SVGD improved upon Messaoud et al. (2024) by 12 folds in high-dimensional differential entropy estimation benchmarks, 80.4% in FID for image generation, and 16% in maximum entropy reinforcement learning.

**Strengths:**

1. Resolves the sensitivity of step sizes in Messaoud et al. (2024), including global invertibility conditions.
2. Adaptively selecting the step sizes and kernel bandwidth by neural networks.
3. Refines the first-order approximation of Messaoud et al. (2024) by second-order approximation, yet avoiding explicit Hessian computation.
4. Finite-sample convergence guarantee by employing the Metropolis-Hastings algorithm.

**Weaknesses:**

1. As the authors point out in L396, MET-SVGD is a Metropolis-Hasting (MH) algorithm with a SVGD-based proposal distribution. Therefore it brings SVGD into the arena of Markov chain Monte Carlo (MCMC), which precisely the SVGD framework intends to avoid.

2. By merging the refined approximation of the SVGD-based particle distribution with MH, it is difficult to discern which component is critical in the empirical performance improvement. It may is plausible that if the method of Messaoud et al. (2024) is incorporated into MH to design a proposal distribution, then the performance improves. On the other hand, with the second-order  refinement alone, will it perform better than Messaoud et al. (2024)? In short, is it the refinement or MH that makes MET-SVGD successful?

3. Proposition 3.5 is not precise. In the proof of Proposition 3.5, the MH acceptance probability is only approximated to the first-order. Convergence of MH usually requires the target density is bounded away from zero. To claim the convergence guarantee, conditions for convergence must be carefully checked.

4. Likewise, in Proposition 3.2 the equality is only approximate.

5. The presentation is the paper is too busy and it is hard to follow. I understand that the authors wanted to put as much as details within the page limit, focusing on the main contributions in an expository manner may work better.

**Questions:**

1. In avoiding explicit Hessian calculation by using Hutchinson's scheme, how the additional random variable $v$ is sampled? and what is the recommended sample size? I think this determines the accuracy of approximation. If done coarsely, then it won't be as good as Messaoud et al. (2024).

2. The method of Messaoud et al. (2024) is called parametric SVGD (P-SVGD). Messaoud et al. (2024) never calls their method P-SVGD. Why is it "parametric"?

---

> ### Author Response · Authors · 2025-11-29
>
> We thank the reviewer for the constructive feedback and address the raised concerns in the following.
> ______________________________
> **W1. MET-SVGD and MCMC.**
> * **MET-SVGD is not a pure MCMC algorithm.** It's a hybrid method that integrates particle-based variational inference (SVGD) with MCMC sampling. Rather than placing SVGD "fully into the MCMC arena", MET-SVGD **bridges** these two approximate inference families combining their strengths while avoiding their main limitations. In particular, MET-SVGD simultaneously benefits from:
> > 1. Fast convergence and sample efficiency,  inherited from SVGD, which helps alleviate the slow mixing behavior in MCMC.
> > 2. Reliable convergence diagnostics via the Stein identity, mitigating the practical difficulty of assessing MCMC convergence.
> > 3. Strong convergence guarantees thanks to the MH correction improving upon SVGD's weak convergence.
>
> * **From an MCMC perspective**, MET-SVGD employs a highly flexible SVGD-based proposal distribution. Under mild regularity assumptions (Villani et al., 2009), this proposal can approximate a broad class of targets. Also, its parameters are optimized end-to-end to adapt to the target complexity; for example, $q^0$ learns to 'zoom' on the target support, which substantially improves the convergence rate [1]. Additionally, the Stein identity enables straightforward convergence assessment.
>
> * **From a variational inference perspective**, the MH step extends the flexibility of the induced variational family of distributions. The resulting distribution is a mixture of P-SVGD transports, offering strictly greater expressiveness than P-SVGD alone (Eq. 7, Updated paper).
>
> * Fig. 9 and Tab. 1 in the Appendix position MET-SVGD within the broader approximate inference landscape and further clarify this hybrid viewpoint.
> ______________________________
> **W2. Comparing the effect of MH and the correction term.**
> The MH step in MET-SVGD (Sec. 3.4) fundamentally relies on a _correct_ expression for the acceptance probability (Prop. 3.5) which, in turn, requires a _correct_ proposal distribution, i.e.,  $q(\tilde{x}|x)$ (Eq. 4) and thus a correct  $q(\tilde{x})$ (see proof in Appendix G.2). The trace correction term is essential for _correctly_ deriving $q(\tilde{x})$ as we explain in Sec. 3.3. Therefore, P-SVGD combined with an MH step without the correction term is mathematically invalid, because its proposal density and acceptance probability are both incorrect. For this reason, the hypothetical comparison suggested by the reviewer (MH + uncorrected P-SVGD) is not well-posed.
> ______________________________
> **W3. Error terms Prop. 3.2 and Prop. 3.5**.
> Thank you for pointing this out. We have added the missing error terms in both Prop. 3.2 $\Big(\mathcal{O}\Big(\epsilon^2\lambda^2_{\text{max}}\big( \nabla_{x}\phi(x)\big)\Big)\Big)$ and Prop. 3.5
> $\Big( \mathcal{O}  \Big(   (\epsilon_{\theta_3}^{l}/\epsilon_{\text{UB}}^{l})^2  \Big)  \Big)$.
>
> The full derivations of the correct higher-order terms are now provided in Appendices D.3 and D.4.
> Importantly, these error terms differ from the simplified $\mathcal{O}(\epsilon^2)$ term used in P-SVGD (Eq. 2).
> We hypothesize that the original authors adopted this simplification because they selected $\epsilon$ to be very small in practice.
> However, as we discuss in Sec. 3.1 (L. 247–253), enforcing such a small step size is sub-optimal and can significantly slow convergence.
> ______________________________
> **W3. Revised Prop. 3.5**. We revised Prop 3.5 to include all necessary assumptions for convergence and elaborated on the proof in App G.
>
> ______________________________
> [1] Sun, Lukang, and Peter Richtárik. "Improved Stein Variational Gradient Descent with Importance Weights." Workshop Optimal Transport and Machine Learning, NeurIPS 2023

---

> ### Author Response · Authors · 2025-11-29
>
> ______________________________
> **W5. Paper presentation**.
> In this revision:
> * We added two algorithms that clearly delineate the training procedure (Alg. 1) and the inference procedure (Alg. 2).
> * We revised Fig. 3 to make it more lightweight. Specifically, we reordered the components for improved readability and retained only references to the Algorithms. Besides, the figure now mirrors the structure of the algorithms for better alignment.
> * To further improve readability, in Fig. 3's caption, we prefix our contributions with colored and numbered **C** and refer back to them throughout the paper to maintain a clear flow.
> * We also made sure that the captions are written to be self-contained so that readers can grasp the main ideas from the figures alone.
>
> We acknowledge that the paper requires some background knowledge and careful reading. To maximize clarity within the page limits, we structured the presentation around a small set of guiding figures with expanded captions to highlight the narrative flow:
> * Fig. 1 introduces the problem setting and situates MET-SVGD within the entropy-estimation literature.
> * Fig. 2 outlines the key limitations of P-SVGD that motivate our method.
> * Fig. 3 illustrates the methodological progression SVGD → P-SVGD → MET-SVGD, making our contributions easier to contrast.
>
> We also provide/refer to 2D illustrative experiments for each contribution, and include additional explanatory figures—e.g., Fig. 5 to clarify the correction term.
>
> ______________________________
>
> **Q1. Hutchinson's estimator.**
> * The random variable $v$ is sampled from any distribution $p_v$ with zero mean and identity covariance. as explained in Sec 3.3. Common choices include Rademacher or standard Gaussian vectors. Under this condition, the Hutchinson estimator is unbiased, and its variance is of the order of $\mathcal{O}(1/V)$ [2], where $V$ is the number of Hutchinson probe vectors.
> * In practice, prior work (e.g., Song et al., 2020) typically uses a single Hutchinson vector per sample $x$. This works well because each element of the minibatch receives an independent draw of $v$, so minibatch averaging implicitly acts as multiple Hutchinson probes of similar samples and significantly reduces variance. This is a standard and stable setting in deep-learning–based trace estimation. We follow the same procedure.
> * Note that Hutchinson's estimator in the MET-SVGD density is scaled by the number of particles $M$. So, the step-wise Hutchinson estimator variance contribution to the SVGD induced distribution is further scaled by $M^2$, i.e., it's of the order of $\mathcal{O}(1/(VM^2))$. This is unlike LD where the step-wise variance is $\mathcal{O}(1/V)$.
> * Thus, adding the trace correction term via the Hutchinson estimator (w. a single $v$ per sample $x$) consistently and significanty improved results across all the experiments.
> * We add this discussion to Appendix C.2 (L1723-1730).
>
> ______________________________
> **Q2. P-SVGD acronym.** Messaoud et al referred to the method as Parameterized SVGD in their abstract (L. 11), as the initial distribution is parametrized via a deepnet and learnt end-to-end via reverse KL-divergence. So, we adopted the P-SVGD acronym even if they didn't introduce it explicitly.
>
>
> ______________________________
> [2] Epperly, Ethan N., Joel A. Tropp, and Robert J. Webber. "Xtrace: Making the most of every sample in stochastic trace estimation." SIAM Journal on Matrix Analysis and Applications 2024

---

### Author Response · Authors · 2025-11-29

We thank the meta-reviewer for their careful assessment of our paper and summarize the main points from the reviews and rebuttal in the following

**Concensus across reviewer.**
All reviewers acknowledge that our work makes significant progress on the under-explored problem of learning variational distributions for unnormalized densities. They also agree that MET-SVGD is a novel, well-motivated, and theoretically principled extension of P-SVGD, delivering strong empirical gains across entropy estimation (Gaussians, GMMs up to 100D), EBM training (3,072D image generation), and MaxEnt RL (up to 17D), consistently outperforming SOTA baselines
_________________________
**Clarifications provided.**
We mainly refined assumptions in the propositions, added two new algorithms detailing the training and inference procedures and revised Fig. 3 to make it more lightweight and aligned with the algorithms.
_________________________
**Key contributions**


* **Entropy Divergence Cause & Fix.** We identify the true cause of entropy divergence in P-SVGD (noise accumulation) and address it via learned step-wise kernel bandwidths and step-sizes.

* **Global Invertibility Guarantee.** We derive a global step-size condition ensuring the SVGD update is invertible, correcting a violated assumption in P-SVGD that leads to erroneous entropy estimates.

* **Unified Step-Size Constraint.** We formalize P-SVGD’s informal constraints into a single principled condition consistent with both invertibility and log-det estimation.

* **Enhanced Entropy Estimation.** We restore the missing Hessian-trace term via Hutchinson’s estimator, substantially improving accuracy in finite-particle settings.

* **Principled Divergence Control.** We replace the empirical particle-truncation heuristic with an MH correction, improving likelihood expressivity and ensuring strong convergence.

* **Adaptive Convergence Criterion.** We introduce a Stein's-identity–based stopping rule that adjusts the number of iterations to the complexity of the target (e.g., in MaxEnt RL).

---

### Meta-Review · Area_Chair_vBDH · 2026-01-05

**Summary:**

The proposed work addresses limitations and drawbacks of previous algorithm on Stein Variational Inference and Stein Variational Gradient Descent and proposes certain improvements.  Among others, these improvements are related to online parameter adaptation to reduce the sensitivity of prior method on SVGD, better approximations on the update laws that reduce the sensitivity and instability attributed to accumulated noise, and the incorporation of Metropolis-Hastings for mitigating and controlling divergence. Although the contributions are multifaceted in this paper, I agree with the reviewers on the poor presentation and organization of the work. This poor organization and presentation is reflected on the low confidence score of the reviewer and the difficulty in reading and understanding the paper.

This is how sometimes important contributions can get lost because of poor organization. I would like to point out specific examples: In Figure 2.a there is no point of having this plot as the qualitative illustration adds nothing substantial to the discussion. This should be removed and free some space.  The caption of figure 3 is repeated in the main text in lines 213 and 244. This caption should be removed, and the figure should move closer to section 3.  This will provide significant space that can be used accordingly to further explain the benefits of the proposed approach.  Finally, when you provide a figure on first page of the paper, this figure should bring upfront data and plots that illustrate the efficacy of the proposed approach. A figure on prior work and the different approaches estimating entropy does not really help the reader to appreciate the science done in the remaining of the paper.  This figure could be completely removed and replaced by text in the introduction.  You could instead bring upfront in page 1 figure 2.2 and 2.3 and show that the proposed methods work better the prior work in approximating GMMs and non-smooth distributions motivating the reader to go deeper into the contain of the paper.

 In summary, my recommendation for this paper is rejection. The paper requires significant and substantial revision.

**Reviewer Concerns:**

Although authors reviews aim to address reviewer comments and criticism the fundamental issue of organization and presentation is not addressed in a satisfactory way. From my perspective the paper needs more work on the organization and presentation of the results.

**Reviewer Scores:**

Reviewer vDmA could have increased its score as most concerns have been addressed.

Reviewer dBCy may have maintain the score because despite the efforts from the reviewers the organization of the paper requires further improvement.

Reviewer TwdB would have maintain the score or further improve it.

Reviewer  Cp7b would have maintain the score.

---

### Decision · Program_Chairs · 2026-01-26

Reject